

# Review of climate simulation by simple climate models

Alejandro Romero-Prieto[1,2,3,*], Camilla Mathison[3,4], and Chris Smith[5,6]

[1] School of Earth and Environment, University of Leeds, Leeds, United Kingdom
[2] Priestley Centre for Climate Futures, University of Leeds, Leeds, United Kingdom
[3] Met Office Hadley Centre, Exeter, United Kingdom
[4] School of Geography, University of Leeds, Leeds, United Kingdom
[5] Department of Water and Climate, Vrije Universiteit Brussel, Brussels, Belgium
[6] Energy, Climate and Environment Program, International Institute for Applied Systems Analysis (IIASA), Laxenburg, Austria
[*] Corresponding author: eearp@leeds.ac.uk

**Abstract.** Simple Climate Models (SCMs) are a key tool in climate research, enabling the rapid exploration of climate responses beyond the reach of more complex models and aiding in the estimation of future climate uncertainty. Over the past two decades, the number and diversity of SCMs have expanded considerably, increasing their use but also complicating efforts to understand differences in model structure and their implications. The reduced-complexity model intercomparison project (RCMIP) has begun to address this challenge by comparing output from a wide range of SCMs. However, the need for a systematic analysis of model structure remains. Here, we complement RCMIP's work by systematically analysing the structure, components, and development histories of the 14 SCMs participating in RCMIP. We begin with a summary of the core principles underpinning SCM-based climate simulation, then review genealogy and design choices of each model. This synthesis provides a comprehensive reference for both developers and users, clarifying the diverse approaches within the SCM landscape and supporting informed use and further development of these models.

## 1 Introduction

Simple Climate Models (SCMs), alternatively termed Reduced-complexity Climate Models (RCMs), are highly-parameterised computationally-efficient climate simulators. This efficiency is primarily achieved in two ways: (i) a reduction of temporal and spatial resolutions, typically operating with global-mean, annual-mean quantities, and (ii) a simplification of simulated processes, often through parameterisation. Consequently, they are positioned at the lowest-complexity level within the climate model hierarchy, beneath Earth system Models of Intermediate Complexity (EMICs), Atmosphere-Ocean General Circulation Models (AOGCMs) and Earth System Models (ESMs). Their efficiency allows SCMs to generate climate projections within seconds while also being relatively easy to understand and use.

This speed makes them a critical tool in climate research, enabling use cases beyond the capabilities of higher-complexity climate models. They have been used in uncertainty estimation (Meinshausen et al., 2009; Nicholls et al., 2021; Smith et al., 2024), scenario creation (Meinshausen et al., 2011b, 2020), ESM emulation (Raper and Cubasch, 1996; Cubasch et al., 2001), and climate projection (Tanaka et al., 2009a; Williams et al., 2017; Goodwin, 2018; Vega-Westhoff et al., 2019). SCMs are also





often used as climate emulators, emulating results from more complex models after being trained on their output. This is a key benefit of SCMs, as it enables the inspection of the vast model space left unexplored by ESMs. Due to the high computational

running costs of ESMs, they can only be executed with a severely-limited set of architectures, boundary conditions, configurations and scenarios, commonly referred to as the "ensemble of opportunity" (Tebaldi and Knutti, 2007). This ensemble of opportunity constitutes only a small fraction of the potential model configurations resulting in realistic climate projections. Furthermore, the heuristic nature of ESM development means that the sampling of the model space is neither systematic nor random, which might introduce significant biases in the exploration of this space. These limitations raise fundamental ques-

tions about the uncertainty of the conclusions drawn from ESMs (Carslaw et al., 2018). SCMs, by virtue of their speed and flexibility, transcend this limitation by using their emulation capabilities to explore the model space beyond the ensemble of opportunity.

The emulation capability of SCMs does not render other models obsolete; complex models like ESMs are still required and highly-useful tools. Firstly, SCMs require simulations provided by more complex models to train their parameterisations

and produce realistic projections. Furthermore, ESMs still offer our most comprehensive understanding of Earth's climate, including many physical properties beyond the scope of SCMs. Moreover, ESMs operate at finer scales, benefitting local and regional analysis, although downscaling approaches to generate regional climate emulators from SCMs have also been explored (Beusch et al., 2020; Mitchell, 2003; Mathison et al., 2025). Ultimately, SCMs constitute a complementary suite of models that can be useful in many scenarios, but they should be developed in conjunction with ESMs to leverage the strengths

of both approaches.

The spectrum of SCMs is wide, ranging from simple linear regression models of global properties to sophisticated schemes operating at regional levels. While this diversity increases the likelihood of finding a suitable SCM for a specific application, it also complicates the selection process for users, particularly for those outside the SCM development community. This problem is not unique to SCMs, with the variety among ESMs also posing significant challenges. However, SCMs lack the extensive

history of model intercomparisons that ESMs have had under the Coupled Model Intercomparison Project (CMIP) umbrella (Meehl et al., 1997, 2000, 2007; Taylor et al., 2012; Eyring et al., 2016), leading the Special Report on Global Warming of 1.5°C (SR1.5, IPCC, 2018) to raise concerns about the validity of SCM simulations.

In recent years, a collective effort has been undertaken by the SCM community to address this question through the systematic evaluation and comparison of SCM output via the RCMIP presented in Nicholls et al. (2020, 2021). Phase one of RCMIP

focused on best estimates for global mean surface air temperature (GSAT) response, while phase two assessed the performance of probabilistic large ensembles to emulate a range of climate metrics, incorporating uncertainty in GSAT projections. While these efforts have significantly increased our understanding of SCM performance, there remains a need for a review on SCM structure and history to aid with the interpretation of model results, as well as model selection. As stated in Nicholls et al. (2020): "An overview of the different models, their structure and relationship to one another (in the form of a genealogy)

would help reduce the confusion and provide clarity about the implications of using one model over another".

Here, we address this need by providing a review of the structure, shared components and development history of SCMs. Additionally, we include information on all available open-source implementations for the analysed models, another gap in




the field identified by Sarofim et al. (2021), hoping to promote transparency and community engagement. Our objective is to provide a clear overview of modern-day SCMs to inform users and developers alike. However, we explicitly consider SCM

performance analysis outside the scope of this review, as this is better accomplished under the RCMIP umbrella.

Establishing criteria for model inclusion in the review was a crucial first step. The decision was made to include all models participating in the RCMIP exercise, as a proxy for the most widely-used and actively-developed modern SCMs. This decision was based on the assumption that SCM users are likely to favour models exhibiting these characteristics, as well as on a desire to mitigate potential biases arising from a more ad-hoc model selection. Although this approach may miss recent additions to

the SCM landscape, the risk is considered low, given that the average first publication date of RCMIP-participating models is 2010, with the latest being 2017, showing relative maturity in participating models.

This is a detailed review tailored to aid understanding of these models, balancing the importance of a concept with its complexity. Accordingly, simpler and more fundamental concepts are reviewed in more depth, while more complex and specific ideas are summarised and appropriately referenced. Mathematical formulations also follow this rationale, being included only

when they promote understanding without hindering legibility. We hope this improves the flow of the review, while creating a useful signposting document for readers seeking more low-level descriptions.

The structure of this review is the following: to contextualise this model review, Sect. 2 offers a brief historical overview of SCM development; Sect. 3 introduces the reader to the basics of SCMs, with some general observations and descriptions of popular model schemes, serving as a reference for subsequent sections; Sect. 4 presents the review of all SCMs participating

in RCMIP, including development history, model description, and some notable uses; Sect. 5 discusses notable commonalities and differences across reviewed models, as well as some limitations of this review; finally Sect. 6 presents the conclusions from this exercise.

## 2 Historical overview

The history of SCMs is closely linked to the broader evolution of climate modelling, as simple analytical models based on

physical principles were initially the only tools available for estimating climate change and assessing potential anthropogenic impacts. The idea of using energy balance considerations to estimate climate change dates back to the late nineteenth century, when the Swedish scientist Svante Arrhenius published his pioneering work quantifying the effects of $CO_2$ on global temperatures (Arrhenius, 1896) - a foundational principle for SCMs. Arrhenius' research built on earlier seminal studies of the Earth's climate, including Fourier's work on the greenhouse effect (Fourier, 1827) and Tyndall's identification of greenhouse gases

(Tyndall, 1861).

In the early twentieth century, efforts to model the global climate often relied on rudimentary energy balance arguments, which, despite their simplicity, are relatively similar to modern approaches (Ångström, 1925). However, the development of these models was significantly limited by the scarcity of meteorological data and the absence of computers, which would not emerge until several decades later. Budyko (1958) provided a detailed review of the state of the field during the first half of the

twentieth century and discussed the challenges faced by researchers at the time.





With the advent of computers and more reliable meteorological data, climate modelling experienced significant advancements during the 1950s and 1960s. Budyko (1969, 1972) and Sellers (1969) introduced one-dimensional thermodynamic models based on heat balance considerations that successfully approximated modern climate. These models not only rekindled interest in the field, but also laid the groundwork for the development of more complex Energy Balance Models (EBMs). At the

same time, technical improvements enabled the creation of climate models that extended beyond what is currently considered within the realm of SCMs, gradually achieving higher levels of complexity and resolution. This progress culminated in the development of the first AOGCM by Manabe and Bryan (1969), marking a pivotal shift towards physically-complex climate models driven by fundamental laws of physics, rather than approximations. The growing diversity of models with varying levels of complexity led Schneider and Dickinson (1974) to define a model hierarchy, categorising climate models from EBMs

to AOGCMs, primarily based on their degrees of freedom. Key publications from this time are: North et al. (1981), providing a survey specifically about EBMs; MacCracken and Luther (1985), offering a broader review of $CO_2$ forcing and climate modelling; and Schneider and Dickinson (1974), reviewing the state of climate models. These publications form a valuable set of resources for understanding the historical context of climate modelling during this period and provide the framework for subsequent developments in the SCM field.

In the 1980s, Wigley and Schlesinger (1985) presented a pure diffusion EBM, which served as a foundational step toward the development of Upwelling-Diffusion Energy Balance Models (UD-EBMs) by Wigley and Raper (1987) and Harvey and Schneider (1985). These models offered an enhanced representation of the oceanic heat sink through advection and diffusion processes, leading to more accurate simulations of surface temperature anomalies and sea-level rise. The model by Harvey and Schneider (1985) significantly advanced our understanding of the Transient Climate Response (TCR), a quantity

that measures the temperature increase after a doubling of pre-industrial carbon concentration in the atmosphere, while the UD-EBM by Wigley and Raper (1987) eventually evolved into the widely influential MAGICC model (described in Sect. 4.9). MAGICC would become a cornerstone in the field of SCMs, participating in all six IPCC assessment reports (IPCC, 1990, 1995, 2001, 2007, 2013, 2021), estimating temperature anomalies, sea-level rise, radiative forcing, and related uncertainties, as well as serving as an emulator of more complex models.

During the 1990s, SCMs began to increase in complexity, incorporating additional climate-relevant processes such as the carbon cycle and non-$CO_2$ Greenhouse Gas (GHG) representations (Wigley and Raper, 1992; Wigley, 1993; Fuglestvedt and Berntsen, 1999). These advancements were critical for making SCMs applicable in policy-relevant scenarios. A useful reference for the state of the SCM field at the turn of the century can be found in IPCC (1997), although it primarily focuses on MAGICC as the only SCM used in IPCC reports at the time. This period also saw the introduction of Impulse Response

Models (IRMs) by Joos and Bruno (1996) to approximate the behaviour of more complex models. This family of methods, and particularly the parameterisations presented in Joos et al. (1996) for the ocean carbon cycle, have proven highly influential in the development of modern SCMs, with several models still employing these emulation techniques, such as MAGICC, FaIR and CICERO-SCM (more details on IRMs are presented in Sect. 3.3.3).

By the first decade of the new millennium, most of the key components of modern SCMs had already been established.

Modelling teams then shifted their focus towards enhancing the flexibility of SCMs, calibrating them to align with the latest





findings from more complex ESMs, and coupling them to other types of models. Notably, some SCMs were coupled to socio-economic modules, giving rise to Integrated Assessment Models (IAMs) (Stehfest et al., 2014; Huppmann et al., 2019; Calvin et al., 2019). During this period, several notable SCMs were introduced, including OSCAR, an SCM with a strong emphasis on the carbon cycle (Gitz and Ciais, 2003; Gitz, 2004), and ACC2, a model capable of running in inverse mode to estimate

parameter uncertainty (Tanaka, 2008). Additionally, Kriegler (2005) published the influential DOECLIM model, which combines a zero-dimensional EBM with a one-dimensional ocean diffusion scheme, and would be later adopted by several modern SCMs: ACC2, Hector and SCM4OPT.

The 2010s witnessed a significant increase in the number of SCMs, with most of the models discussed in this review being introduced during this decade: GREB (Dommenget and Flöter, 2011), EM-GC (Canty et al., 2013), AR5-IR (Joos et al., 2013),

Hector (Hartin et al., 2015), ESMICON (Randers et al., 2016), originally called ESCIMO, WASP (Goodwin, 2016), FaIR (Millar et al., 2017), MCE (Tsutsui, 2017), and SCM4OPT (Su et al., 2017). This proliferation of models was driven by the need to assess a wide range of scenarios and processes, particularly for policy analysis. As a result, there was a demand for a more diverse and flexible family of models that could be calibrated to the outputs of more complex ESMs, as well as coupled with IAMs to evaluate socio-economic scenarios in the context of climate policy. Many of these new models adopted an $n$-layer

formulation for their EBMs, building on the work of Held et al. (2010) with a two-layer model and a three-layer model (Tsutsui, 2017, 2020) (see Sect. 3.3.2 for more details on $n$-layer models). These formulations enabled a clearer distinction between "fast" and "slow" components of climate change, simplifying interpretability and refining model accuracy and applicability.

The rapid increase in the number of SCMs during the 2010s highlighted the need for a systematic inter-SCM comparison, akin to the Coupled Model Intercomparison Project (CMIP, Eyring et al., 2016) initiative for AOGCMs and ESMs. To satisfy

this need and address the concerns raised in the SR1.5 (IPCC, 2018) relating to the accuracy of SCMs, the RCMIP initiative was born (Nicholls et al., 2020, 2021). This SCM intercomparison has become a crucial resource for understanding modern SCMs and evaluating their performance. RCMIP has been instrumental in identifying the strengths and weaknesses of different models, thereby guiding future developments in the field. However, this intercomparison project did not focus on the operational mechanisms or specific schemes employed by SCMs. Addressing this gap is the main objective of this review, which aims to

provide an examination of the mechanisms and methodologies underlying these models.

## 3   Core principles and mechanisms of SCMs

Typically, SCMs follow a simple framework to simulate climate change. This framework, referred to as the emissions-climate change cause-effect chain by Nicholls et al. (2020), consists of three stages:

1. Determine the atmospheric concentrations of climate-relevant chemical species based on emissions and planetary sinks.

2. Calculate the total radiative forcing, that is, the disturbance of the planetary energy balance between incoming and outgoing energy. This may include contributions from GHG species simulated in the previous stage, as well as non-GHG-related contributions, such as albedo changes and aerosols.





3. Estimate temperature change resulting from that forcing, typically using an EBM.

These stages are followed by most SCMs in this review. Consequently, they have been used to structure the description of the models in Sect. 4, which describes how each SCM simulates each stage (when relevant). The current section discusses some generalities about these stages, as well as common approaches taken by SCMs. The objective is two-fold: helping the reader gain some familiarity with SCMs schemes in preparation for the model descriptions presented in Sect. 4, as well as serving as a reference for these descriptions.

## 3.1 GHG concentrations

### 3.1.1 Mass balance model

The evolution of concentrations of atmospheric long-lived non-$CO_2$ species (and sometimes $CO_2$) is typically modelled by a mass balance model, alternatively known as a single-reservoir box model:

$$\frac{dC^x}{dt} = \Pi^x - \Lambda^x = \frac{E^x}{\beta^x} - \frac{C^x}{\tau^x} \tag{1}$$

In this representation, the change in concentration of a given species $x$ ($C^x$) is governed by its production ($\Pi^x$) and loss rates ($\Lambda^x$). The production rate is simply the species emissions ($E^x$), converted to concentration units with a conversion factor $\beta^x$ (this is sometimes omitted in model descriptions). Emissions can be purely anthropogenic in origin, be caused by natural processes or a combination of both, depending on the species, the considered scenario, and the internal structure of the model. The loss rate is typically assumed to be an exponential decay, characterised by a global decay time $\tau^x$. This assumption makes the scheme a particular case of the wider family of impulse-response models, as described in Sect. 3.3.3. The characteristic lifetime of the species may not directly correspond to any single physical process, but combine the effects of multiple sinks removing that species from the atmosphere at similar timescales. In general, the lifetime can be state-dependent, altering its value based on an number of climate properties. This is, for instance, what the FaIR model does (Sect. 4.3) for $CO_2$ and $CH_4$, including carbon and temperature feedbacks for those species lifetimes. However, this is unusual in model representations, with most SCMs adopting a constant lifetime. This is particularly the case for minor GHGs (all GHGs except $CO_2$, $CH_4$ and $N_2O$). A more typical approach to increase the flexibility of the lifetime scheme involves the inclusion of additional sinks acting at different timescales. This is typically incorporated into models through the use of a total lifetime defined as:

$$\frac{1}{\tau_{total}} = \frac{1}{\tau_1} + \frac{1}{\tau_2} + \cdots + \frac{1}{\tau_n} \tag{2}$$

where $n$ is the number of sinks considered. This is particularly relevant for tropospheric methane, represented in most models with a mass balance equation and a total lifetime with contributions from multiple sinks, typically including its reaction with hydroxyl radicals (OH) ($\sim 11.2$ years), stratospheric loss ($\sim 120$ years) and soil uptake ($\sim 150$ years) (Prather et al., 2012; Myhre et al., 2013). Using equation 2 this results in a total lifetime of $\sim 9.6$ years.





This mass balance formulation focuses on the amount of species $x$ that remains airborne after some time $t$, without providing any additional information about the fate or potential impacts after its removal from the atmosphere. Often, this level of detail is enough to satisfy SCM objectives, particularly for chemical species lacking a natural cycle. Consequently, SCMs generally disregard the dynamics of these species after they cease to be relevant for the calculation of radiative forcing. However, this is not the case for $CO_2$, because this species possesses a strong natural cycle with large fluxes across four reservoirs in the Earth's system: atmosphere, ocean, land and biosphere. Understanding the response of this cycle to anthropogenic disturbances, as well as any potential tipping points, is therefore necessary for accurate predictions of future atmospheric concentrations and ultimately, to estimate compatible emission scenarios with climate stabilisation targets (Friedlingstein et al., 2023). Indeed, this is currently an area of intensive research, often discussed in terms of the Zero Emissions Commitment (ZEC), which quantifies the global temperature change in a post-net-zero world (Palazzo Corner et al., 2023). SCM development teams, recognising the need for details on the carbon cycle, have generally either started with (Fuglestvedt and Berntsen, 1999; Gitz and Ciais, 2003; Tanaka, 2008; Hartin et al., 2015; Goodwin, 2016; Su et al., 2017) or transitioned towards (Wigley and Raper, 1992; Tsutsui, 2022) $CO_2$ representations more complex than this mass balance model. By far, the most popular representation for SCM carbon cycles, particularly for the land component, is box-based carbon cycle models, which simulate the flow of carbon through the different Earth's reservoirs.

### 3.1.2 Box-based carbon models

Box-based carbon models consist of a number of conceptual boxes, also known as pools or reservoirs, that abstract away the carbon content of certain parts of the carbon cycle (e.g., vegetation, soil, ocean (layers), atmosphere). These boxes exchange carbon through fluxes which can be modulated by climate properties. For example, the carbon transport from vegetation to the soil can be simulated through a litterfall flux, which can be magnified by higher temperatures. The carbon inventories of the model boxes are determined by the initial stocks and the carbon fluxes. SCMs typically aim to represent all the main fluxes in the Earth's carbon cycle, such as net primary production (NPP), litterfall, decomposition and respiration. See, for instance, the depiction of the carbon cycles of Hector (Fig. 5a) and MAGICC (Fig. 7). More details about the nature and magnitude of these fluxes, as well as the wider Earth's carbon cycle, can be found in Reichle (2023).

SCMs can simulate these fluxes in different ways. Some fluxes may be modelled by relatively simple analytical expressions, accounting for climate feedbacks. This is particularly common for the land component. Other fluxes may be modelled by more complex schemes, such as carbonate schemes to estimate the dissolution of atmospheric carbon into the ocean mixed layer (OML), the uppermost layer of the ocean exhibiting relatively uniform properties due to turbulence homogenisation effects. A common approach, particularly for the ocean component, is the use of IRMs, which are discussed in Sect. 3.3.3.

This seemingly simple representation of carbon reservoirs and fluxes provides modelling teams with remarkable flexibility. It is easy to extend, with the inclusion of new components merely requiring the definition of the carbon content of the new pool and interacting fluxes. It also allows the inclusion of multiple fluxes, each considering different climate feedbacks and being calibrated independently. Furthermore, it is a computationally efficient approach, with the number of parameters and calculations required remaining relatively low. Crucially, box models offer an internally-consistent framework that can keep




track of the flux of carbon in a closed cycle. These qualities are further amplified by the conceptual simplicity of these models, which aids interpretability and visualisation, facilitating their adoption by both users and developers.

The resolution of box models is typically global, with boxes representing the entire Earth's carbon content for that category (e.g., soil, vegetation). However, some models like Hector and OSCAR further partition these boxes into smaller units representing regions or even regional biomes.

## 3.2 Radiative forcing

After computing the concentration of climate-relevant atmospheric species, the next stage in the climate chain is calculating the total radiative forcing. More details will be discussed in Sect. 3.3.1 on energy balance models and how the climate response to forcing complicates the calculation, but for now let us define radiative forcing as the perturbation of the pre-industrial energy balance between incoming and outgoing energy fluxes.

Radiative forcing is a critical quantity, as it determines the extent of warming or cooling that the Earth experiences with respect to the pre-industrial state. Crucially, it is a linear quantity, allowing the total global radiative forcing to be calculated simply as the addition of individual contributions from different sources. This linearity is particularly advantageous for comparing the relative influence of various forcing sources—for example, assessing the impact of methane concentrations in the atmosphere versus changes in cloud properties. Such comparisons would be far less trivial using raw variables like methane concentrations and cloud optical depth. The downside of this property, however, is that measurement of individual forcing contributions is not possible; instead models are required to estimate it, introducing a source of uncertainty.

Potential sources of radiative forcing, also known as forcing agents, are numerous: $CO_2$, $CH_4$, $N_2O$, minor GHGs (CFCs, HCFCs, HFCs, and others - see Table 7.SM.6 in Smith et al. (2021a) for a full list), ozone (tropospheric and stratospheric), aerosols (considering both direct impacts and indirect via cloud interactions), changes in albedo (induced by land use and land cover changes (LULCC), as well as black carbon deposition on snow), irrigation, aviation-induced contrails, and natural phenomena like changes in solar irradiance and volcanic eruptions. Most SCMs will cover a subset of these forcing agents with varying degrees of complexity, as summarised in Table 2.

Different types of radiative forcing exist, depending on which part of the Earth system is required to have reached a thermal equilibrium before the energy imbalance is considered (IPCC, 2021). Traditionally, Stratospherically-Adjusted Radiative Forcing (SARF) was a popular option in SCMs, as this is an easier quantity to compute (Myhre et al., 2013). However, Effective Radiative Forcing (ERF) has progressively emerged as the metric of choice, as its inclusion of all climate adjustments offers a better estimation of the surface temperature response to forcing (Forster et al., 2016). A common approach to calculate ERF is to compute SARF and multiply it by a scaling parameter to account for tropospheric adjustments (Meinshausen et al., 2020; Smith et al., 2024). Generally, SARF should be assumed whenever radiative forcing is mentioned in this document, unless otherwise stated.

A common source for forcing estimations is the reports written by the Intergovernmental Panel on Climate Change (IPCC), which tend to use simplified analytical expressions to describe the impacts of forcing agents. In particular, there have been three studies which have been extensively used in SCM forcing estimations:



– Myhre et al. (1998): established a logarithmic expression for the SARF resulting from elevated $CO_2$ concentrations (Eq. 30), and square-root expressions for $CH_4$ and $N_2O$.

     – Etminan et al. (2016): revised Myhre et al. (1998) expressions to account for $CH_4$ shortwave effects and overlapping in radiation absorption bands between $CO_2$, $CH_4$ and $N_2O$.

     – Meinshausen et al. (2020): re-fit the Etminan et al. (2016) expressions to reduce the error between the curve fit and the
radiative transfer model-derived SARF, and extended the validity range to high $CO_2$ concentrations.

Most SCMs follow one of these studies to estimate forcing from major GHGs, provided they possess a representation of the relevant species concentration. Beyond these expressions, consensus among SCMs diminishes and cross-model variations abound in areas such as the number of considered forcing agents, the estimation schemes for minor GHGs, and the values used for their parameterisations. A common approach, particularly for halogenated compounds, is a linear scaling of the species
concentration (or emissions if it is a short-lived species) by its radiative efficiency (Smith et al., 2021a), thus considerably simplifying the calculation.

### 3.3 Temperature

#### 3.3.1 Energy balance model

Once the total radiative forcing has been calculated, SCMs need a method to translate that quantity into a global surface tem-
270 perature anomaly. This is usually achieved by an EBM (Wigley and Raper, 1987; Schlesinger et al., 1992; Kriegler, 2005; Meinshausen, Wigley and Raper, 2011; Geoffroy et al., 2013; Goodwin, 2018). This type of model relies on a series of assumptions and approximations that will be briefly described in this section. A more comprehensive review of the derivation and validity of these assumptions can be found in Appendix A of Kriegler (2005).

In an equilibrium state, incoming solar energy absorbed by the Earth system ($F^{in}$) must be equal to the outgoing radiated
infrared energy ($F^{out}$). However, the atmosphere absorbs a significant amount of the outgoing radiation leaving the Earth's surface and emits it back down, increasing surface temperature. This is known as the natural greenhouse effect ($G$). Approximating the Earth as a black body, one can estimate the outgoing surface radiation using the Stefan-Boltzmann law: $F^{out} = \sigma T_S^4$ where $T_S$ is the average global surface temperature and $\sigma = 5.67 \cdot 10^{-8} \, Wm^{-2}K^{-4}$ is the Stefan-Boltzmann constant. In equilibrium, the outgoing energy flux must, therefore, equal the incoming radiation plus the energy absorbed by the atmosphere:

$$F^{out} = \sigma T_S^4 = F^{in} + G \qquad (3)$$

If that energy balance is forced with a small perturbation to the amount of incoming or absorbed energy ($\Delta E$) the system will respond with a heat flux at the surface, changing the surface temperature until the system is back at equilibrium. By conducting a Taylor expansion of the outgoing energy around the original equilibrium surface temperature ($T_0$) and retaining only the first order term, one can approximate the temperature response induced by this perturbation as:



$$F^{\text{out}}(T) = \sigma T_S^4 \approx \sigma T_0^4 + 4\sigma T_0^3 \Delta T \Rightarrow \tag{4}$$

$$\Delta F^{\text{out}} = F^{\text{out}}(T) - F_0^{\text{out}} \approx \sigma T_0^4 + 4\sigma T_0^3 \Delta T - \sigma T_0^4 = 4\sigma T_0^3 \Delta T \tag{5}$$

where $\Delta T$ is the difference between the new and original temperature, or temperature anomaly. This linear approximation is valid only for small temperature differences of a few degrees, with non-linear terms becoming relevant for larger anomalies (Bloch-Johnson et al., 2021). Consequently, the heat flux ($dH/dt$) induced by this perturbation in the energy balance during the transient state will be equal to the perturbation ($\Delta E$) minus the surface temperature response:

$$\frac{dH}{dt} = \Delta E - 4\sigma T_0^3 \Delta T \tag{6}$$

where the perturbation can take place either in the incoming radiation or the greenhouse effect: $\Delta E = \Delta F^{\text{in}} + \Delta G$. $\Delta G$ corresponds to the anthropogenic greenhouse effect (assuming the perturbation is human in origin). Typically with energy balance models this perturbation is assumed to be separable into two terms:

- A radiative forcing term, $F$. In SCMs this is usually taken to be the stratospherically-adjusted radiative forcing or the effective radiative forcing (see Sect. 3.2).

- A temperature feedback term comprising all the climate responses to a change in temperature that, in return, have an impact on the energy imbalance. Usually this is assumed to be a linear response to the temperature anomaly ($k \cdot 4\sigma T_0^3 \Delta T$), which only holds for small disturbances.

Under this assumption Eq. 6 can be rewritten as:

$$\frac{dH}{dt} = \Delta E - 4\sigma T_0^3 \Delta T = F + k \cdot 4\sigma T_0^3 \Delta T - 4\sigma T_0^3 \Delta T = F - \lambda \Delta T \tag{7}$$

where $\lambda = 4\sigma T_0^3(1-k)$ is known as the climate feedback parameter and combines the effect of increased blackbody radiation ($4\sigma T_0^3$) and temperature feedbacks caused by the perturbation in the energy flux ($4\sigma T_0^3 k$). In most instances, this is a constant model parameter that can be tuned to emulate results from other, more complex ESMs, although some SCMs allow a time-dependent representation of $\lambda$, like MAGICC (Sect. 4.9.3) and WASP (Sect. 4.5.3).

Typically, Eq. 7 is modified to assume that the heat flux results in an increase in surface temperature, mediated by a global heat capacity $C$, thereby converting it into the following first order differential equation:

$$C\frac{dT}{dt} = F - \lambda T \tag{8}$$





Note that the surface temperature anomaly, $T$, is presented without the delta notation for the sake of simplicity, although it

continues to represent the same anomaly.

The formalism outlined thus far is common to most simple climate models. Where they diverge is in their treatment of the heat flux distribution and effects across the Earth system. While most models include a representation of the heat flux towards the ocean, the largest heat reservoir in the Earth system, how that heat is distributed across the ocean varies. Heat diffusion to the deep ocean is often included, with the occasional addition of heat transport by ocean currents. Representations of the land

heat sink are less frequently included.

### 3.3.2   *n*-layer EBM

Equation 8 provides an approximate representation of the temperature response to the energy imbalance induced by radiative forcing. Assuming a constant climate feedback parameter ($\lambda$), the resulting relationship provides an estimate of the global temperature necessary for the climate system to reach equilibrium with the new radiative forcing.

However, this equilibrium is not achieved instantaneously. The climate system exhibits a large thermal inertia (relative to human timescales), primarily due to the stabilising influence of the deep ocean, which acts as a large heat reservoir. The ocean is estimated to have absorbed around 89 % of the additional heat coming into the Earth system as a result of the historical energy imbalance (von Schuckmann et al., 2023). Consequently, the transient temperature anomaly induced by a certain amount of positive radiative forcing is smaller than the eventual temperature once the system reaches a new equilibrium.

A commonly-used method to incorporate this deep-ocean thermal inertia to SCMs is the two-layer or two-box model of temperature response. This model, referred to as the "Held et. al two-layer model" in Nicholls et al. (2020), is based on the work of Held et al. (2010). Fundamentally, this representation adds an additional box with a large heat capacity to the EBM described by Eq. 8. As a result, the model possesses the original "rapid" box simulating the fast temperature response of the atmosphere, land and ocean boundary layer to the changes in radiative forcing, and an additional second "slow" box coupled

to the first that emulates the slow response of the deep ocean. Mathematically, this can be expressed as follows:

$$
\begin{aligned}
C_F \frac{dT}{dt} &= F - \lambda T - H_o \\
C_o \frac{dT_o}{dt} &= H_o
\end{aligned}
\tag{9}
$$

where $C_F$ and $C_o$ are the heat capacities for the fast and slow boxes, respectively, with $C_o >> C_F$. $H_o$ is the exchanged heat between the two layers, which is assumed to be proportional to the difference in temperature anomaly between the two boxes, characterised by a heat exchange coefficient $\kappa$, $H_o \equiv \kappa(T - T_o)$.

However, an important limitation to this model is its inability to resolve evolving spatial warming patterns that occur as the system approaches equilibrium, as discussed in in Williams et al. (2008) and Winton et al. (2010). These evolving patterns modulate the outgoing energy flux to space, impacting the global temperature response. Since this phenomenon can be related to the ocean heat uptake, Held et al. (2010) introduced an efficacy parameter, $\epsilon$, to account for this effect:



$$C_F \frac{dT}{dt} = F - \lambda T - \epsilon\kappa(T - T_o)$$
$$C_o \frac{dT_o}{dt} = \kappa(T - T_o)$$

(10)

This new formulation is equivalent to Eq. 9 but with heat exchange and deep-ocean exchange coefficients scaled by $\epsilon$, defining $\kappa' \equiv \epsilon\kappa$ and $C_o' \equiv \epsilon C_o$. Winton et al. (2010) reported values for this efficacy parameter greater than one for nearly all models under 1pctco2 experiments. Rohrschneider et al. (2019) provided analytical solutions to this system of differential equations, and compared them with other EBMs, specifically a two-region model - which they demonstrated to be mathematically equivalent to the two-layer model in its temperature response - a one-layer model with a temperature-dependent feedback, and a hybrid of the two.

The two-layer model can be easily generalised to an $n$-layer model by adding new ocean layers beneath the extra layer introduced in Eq. 9, defining a system of n differential equations (Cummins et al., 2020):

$$C_1 \frac{dT_1}{dt} = F - \kappa_1 T_1 - \kappa_2(T_1 - T_2)$$
$$C_2 \frac{dT_2}{dt} = \kappa_2(T_1 - T_2) - \kappa_3(T_2 - T_3)$$
$$\vdots$$
$$C_{n-1} \frac{dT_{n-1}}{dt} = \kappa_{n-1}(T_{n-2} - T_{n-1}) - \epsilon\kappa_n(T_{n-1} - T_n)$$
$$C_n \frac{dT_n}{dt} = \kappa_n(T_{n-1} - T_n)$$

(11)

In this generalised formulation, the top layer corresponds to the "fast" layer in Eq. 10 ($C_1, T_1, \kappa_1, \kappa_2$ and $T_2$ are $C_F, T, \lambda, \kappa$ and $T_o$, respectively). Note that the efficiency parameter, $\epsilon$, is only present in the penultimate layer, as the deepest layer is still the largest reservoir of heat (i.e., it possesses the largest heat capacity) dominating the deep ocean uptake.

A higher number of layers provides the model with more flexibility and an enhanced capability to reproduce non-linearities. Despite this, the number of layers is typically limited to two or three in SCM implementations. A two-layer model, calibrated to Coupled Model Intercomparison Project Phase 5 (CMIP5) model output (Geoffroy et al., 2013a, b), has been employed as the EBM module in several SCMs, including OSCAR, AR5-IR, and FaIR until v2.0. FaIR v2.0 increased the number of layers to three by default based on evidence from Tsutsui (2017, 2020) and Cummins et al. (2020) that suggested three layers are usually sufficient to accurately capture the temperature response of ESMs. Notwithstanding this finding, to further enhance the model's flexibility, FaIR v2.1 allows an arbitrary large number of $n$ layers (as long as it is larger than one), although the number of tuning parameters quickly increases as $n$ becomes larger. It is important to note that $n$-layer EBMs might not be immediately recognizable as such, as they are sometimes presented in an alternative formulation related to a broader family of models, known as IRMs.



### 3.3.3 Impulse response models

As previously discussed, the primary purpose of SCMs is to emulate the behaviour of more complex climate models efficiently. In the preceding sections, methods were explored that aimed to achieve this emulation through the development of
computationally efficient representations of the Earth system maintaining physical intuition. IRMs (Joos and Bruno, 1996) take a different route; they are not derived from fundamental physical principles but through mathematical approximations of the relationships we aim to model.

IRMs tune empirical Impulse-Response Functions (IRFs) to simulate the behaviour (response) of a variable of interest to a perturbation (impulse) in a related property. This tuning is typically based on the output of more complex models, such as
AOGCMs or ESMs. IRFs are also known as "Green's functions" in other fields of physics.

IRFs can fully characterize the dynamical response of a linear system, providing only an approximation for non-linear systems. The quality of the approximation depends on the extent of the system's deviation from linearity. As previously discussed in the derivation of Eq. 8 for the standard EBM, a quasi-linear assumption of the climate system is often employed when working with SCMs, justifying the use of IRFs to approximate climate properties.

For climate-related quantities, IRFs typically take the form of a sum of $n$ decaying exponentials with characteristic timescales $\tau_i$ and magnitudes $A_i$. Thus, the value of a property of interest $x(t)$ at time $t$ can be approximated by evaluating the convolution of the magnitude of the impulse $F(t')$ up to time $t$ with the tuned response to that impulse (i.e., the sum of exponentials):

$$x(t) = \int_0^t F(t') \cdot IRF(t - t')dt' = \int_0^t F(t') \sum_i A_i exp\left(-\frac{t - t'}{\tau_i}\right) dt' \tag{12}$$

Alternatively, taking the time derivative of Eq. 12 yields the equivalent differential equation which is also often employed:

$$\frac{dx(t)}{dt} = \sum_i \left[ A_i F(t) - \frac{x_i(t)}{\tau_i} \right] \tag{13}$$

As an example, the temperature anomaly $T(t)$ after a unit of radiative forcing at $t = 0$ (represented by a dirac delta function $\delta$ centered at time $t' = 0$) can be approximated using an IRM by (Millar et al., 2015):

$$T(t) = \int_0^t \delta(t' = 0) \sum_i A_i exp\left(-\frac{t - t'}{\tau_i}\right) dt' = \sum_i \frac{A_i}{\tau_i} \exp\left(\frac{-t}{\tau_i}\right) \tag{14}$$

Through the tuning of the $\tau_i$ and $A_i$ parameters, Eq. 14 can be calibrated to emulate the temperature response of more
complex climate models. Alternatively, IRFs can model the increase in atmospheric $CO_2$ concentration following an emissions pulse at $t = 0$, as seen in the AR5-IR (Myhre et al., 2013) and FaIR (Millar et al., 2017) models reviewed below and discussed in Sect. 3.1.1, or the transfer of carbon from the ocean mixed layer to the deep ocean (Joos et al., 1996). This last IRM has been





a particularly influential scheme in the SCM field, being used to simulate the ocean carbon cycle in three prominent SCMs: CICERO-SCM, OSCAR and MAGICC. More details about this scheme are presented in Sect. 4.8.1.

The family of $n$-time-constant temperature IRMs described by Eq. 13 and Eq. 14 is particularly interesting because it is mathematically equivalent (Millar et al., 2015; Tsutsui, 2017; Leach et al., 2021) to the family of $n$-layer temperature models described by Eq. 11. In its general IRM form, Eq. 13 can be expressed as:

$$\frac{dT_i}{dt} = \frac{A_i F - T_i}{\tau_i}; \ T = \sum_{i}^{n} T_i \tag{15}$$

which is a diagonalised form of the equation one would obtain if Eq. 11 was expressed in matrix form (Leach et al., 2021;
Geoffroy et al., 2013a). Table 1 in Geoffroy et al. (2013a) provides a list of conversions between the constants in the two formulations for the case $n = 2$. Note that in this formulation, the temperature anomaly at the surface, $T$, is the addition of the contributions from the different temperature components.

Crucially, Geoffroy et al. (2013a) also presented a simple way to relate the parameters in Eq. 15 to two of the most critical and widely discussed quantities in climate science, the Equilibrium Climate Sensitivity (ECS) and the TCR:

$$ECS = F_{2 \times CO_2} \sum_{i=1}^{n} A_i \tag{16}$$

$$TCR = F_{2 \times CO_2} \sum_{i=1}^{n} A_i \left( 1 - \frac{\tau_i}{D} \left( 1 - exp\left( -\frac{D}{\tau_i} \right) \right) \right) \tag{17}$$

where $D = \ln(2)/\ln(1.01) \approx 69.7$ years, is the time required to double the atmospheric $CO_2$ concentration under a 1 % yearly increase scenario. These relations are highly advantageous because, for the case $n = 2$, and given $\tau_1$ and $\tau_2$, they can be inverted to determine $A_1$ and $A_2$ as functions of ECS and TCR. This allows for a straightforward definition of an
EBM consistent with any combination of ECS and TCR values. Even for $n > 2$, Eq. 16 and 17 remain relevant, as they define a hyperplane where any combination of ECS and TCR values can be easily obtained. Characteristic timescales, $\tau_i$ are typically taken following previous studies of $n$-layer models. FaIR v1.0, for instance, based its 2-time IRM on the characteristic timescales of the multi-model mean of a 2-layer model tuned to CMIP5 AOGCMs by Geoffroy et al. (2013a).

## 4 Model description

This section provides descriptions for all SCMs participating in the first and second phases of RCMIP, with the exception of the "Held et al. two-layer model". This model is simply a two-layer EBM, which has already been described in Sect. 3.3.2 regarding $n$-layer EBMs. For the remaining models, an overview of their components and development history is offered first, along with illustrative examples of their application. Following this, each model is described in greater detail, following the emissions-temperature cause-effect chain described in Sect. 3, when applicable. For most models, this translates into three subsections:



**Table 1.** Summary of carbon cycle representations from the SCMs reviewed in this study. This includes the general type of model used (Impulse-response model (IRM) or box model), its resolution, as well as the land-, ocean- and permafrost-specific representations (if any). Inclusion of carbon-relevant processes in these SCMs (fire, precipitation and nitrogen cycle) is also summarised. OML-IRM is shorthand for Ocean-mixed layer IRM, and references a specific IR scheme by Joos et al. (1996).

| | | | Carbon cycle | | | | | |
|---|---|---|---|---|---|---|---|---|
| Model name | Model type | Resolution | Land | Ocean | Permafrost | Fire | Precip. | N cycle |
| ACC2 (V4.3) | Box model | Global | 4 boxes | 4 boxes | - | - | - | - |
| AR5-IR | 4-time IRM | Global | - | - | - | - | - | - |
| CICERO-SCM (V1.1) | IRM | Global | 4-time IRM | OML-IRM | - | - | - | - |
| EMGC | - | - | - | - | - | - | - | - |
| ESMICON | System dynamics box model | Global | Combined land/ocean 6 boxes | | 1 box | Yes | - | - |
| FaIR (V2.1) | State-dependent 4-time IRM | Global | - | - | - | - | - | - |
| GREB | - | - | - | - | - | - | Yes | - |
| Hector (V3.2) | Box model | Global/biomes for land, high-low latitude boxes for ocean | 3 boxes | 4 boxes | 1 frozen pool + 1 thawed pool | - | - | - |
| Held et al. 2-layer model | - | - | - | - | - | - | - | - |
| MAGICC (V7) | Box model | Global | 3 boxes | OML-IR | 50 latitudinal bands | - | - | Yes |
| MCE (V1.2) | Box model - IRM hybrid | Global | 4-time IRM | 4 boxes | - | - | - | - |
| OSCAR (V3.3) | Box model | Regional biomes for land, global for ocean | 3 boxes + 3 wood boxes per biome per region | Box-equivalent of modified OML-IR | 1 frozen pool + 3 thawed pools (2 regions) | Yes | Yes | - |
| SCM4OPT (V3.3) | Box model | Global | 3 boxes | 4 boxes | - | - | - | - |
| WASP (V3) | Box model | Global | 2 boxes | 5 boxes | - | - | - | - |

"GHG concentrations", "radiative forcing", and "temperature", each describing how the SCM simulates the relevant processes. Special attention is given to the carbon cycle in the "GHG concentrations" sections, reflecting its role as the primary greenhouse gas and the frequent inclusion of dedicated carbon-cycle simulation schemes in SCMs. To minimize repetition, the models are generally presented in order of increasing complexity, allowing references to previously discussed components where appropriate. Tables 1-4 summarise details about the carbon cycle representations, included radiative forcing agents, temperature

modules and technical details from these models, while Figs. 1 and 2 depict their development chronology. Finally, it is important to notice that in phase 2 of RCMIP, the "AR5-IR" model referred to a 2-time IRM EBM without any gas cycle representation. In contrast, this review adopts a broader definition of the "AR5-IR" model, referring to a 2-time IRM EBM coupled to a 4-time IRM simulating atmospheric carbon sinks. This expanded interpretation aligns with the designation used by the FaIR SCM publications, where FaIR was originally developed as an extension of this expanded version. Hence, the

adoption of the broader definition is more relevant for this review.

## 4.1 EM-GC

The Empirical Model of Global Climate (EM-GC) is an SCM developed at the University of Maryland College Park, USA. It is based on an empirical linear regression model that estimates the Global Mean Surface Temperature (GMST) anomaly from





**Figure 1.** Chronology of SCM development. The evolution of each model is represented along a timeline, with all published references documenting its development superimposed on the corresponding point in time. Colours indicate the programming language used for each model's implementation, as shown in the legend. Bands around MAGICC's timeline denote a wrapper that allows interfacing with a different programming language (Python) than the model's native implementation.





**Figure 2.** Chronology of SCM development. The evolution of each model is represented along a timeline, with all published references documenting its development superimposed on the corresponding point in time. Colours indicate the programming language used for each model's implementation, as shown in the legend and Table 4. Bands around Hector's timeline denote a wrapper that allows interfacing with different programming languages (Python and R) than the model's native implementation.



**Table 2.** Summary of included sources of radiative forcing in the models reviewed in this study.

* Contributions from these sources are included explicitly as prescribed forcing time series, rather than internally estimated.

† ESMICON follows a system dynamics framework and does not compute radiative forcing estimates. It does, however, include effects of multiple radiative forcing agents.

| | | | | | Radiative forcing agents | | | | | | | |
|---|---|---|---|---|---|---|---|---|---|---|---|---|
| **Model** | **CO$_2$** | **CH$_4$** | **N$_2$O** | **Halogens** | **Trop. O$_3$** | **Strat. O$_3$** | **Strat. H$_2$O** | **Aerosols** | **Volc. aerosols** | **Solar irr.** | **Contrails** | **LULCC** |
| ACC2 (V4.3) | Yes | Yes | Yes | 30 | Yes | Yes | Yes | Yes | Yes* | Yes* | – | – |
| AR5-IR | Yes | – | – | – | – | – | – | – | – | – | – | – |
| CICERO-SCM (V1.1) | Yes | Yes | Yes | 27 | Yes* | Yes | Yes | Yes | Yes* | Yes* | – | Yes* |
| EM-GC | Yes | Yes | Yes | 31 | Yes* | – | Yes | Yes* | Yes* | Yes* | – | Yes* |
| ESMICON | Yes† | Yes† | – | – | – | – | – | – | – | Yes† | – | Yes† |
| FaIR (V2.1) | Yes | Yes | Yes | 40 | Yes | Yes | Yes | Yes | Yes* | Yes* | Yes | Yes* |
| GREB | Yes | – | – | – | – | – | – | – | – | – | – | Yes |
| Hector (V3.2) | Yes | Yes | Yes | 27 | Yes | – | – | Yes | Yes* | – | – | Yes* |
| Held et al. 2-layer model | Yes | – | – | – | – | – | – | – | – | – | – | – |
| MAGICC (V7) | Yes | Yes | Yes | 40 | Yes | Yes | Yes | Yes | Yes* | Yes* | Yes | Yes* |
| MCE (V1.2) | Yes | Yes | Yes | 41 | Yes* | Yes* | Yes* | Yes* | Yes* | Yes* | Yes* | Yes* |
| OSCAR (V3.3) | Yes | Yes | Yes | 37 | Yes | Yes | Yes | Yes | Yes* | Yes* | Yes* | Yes |
| SCM4OPT (V3.3) | Yes | Yes | Yes | 39 | Yes | Yes | Yes | Yes | Yes* | Yes* | – | Yes |
| WASP (V3) | Yes | Yes | Yes | 27 | – | – | – | Yes | Yes* | Yes* | – | – |

various natural and anthropogenic sources of radiative forcing. It does not include a representation of gas cycles, requiring
GHG concentration time series as input. Despite this simplicity in the GHG cycles, EM-GC is one of only two models in this
review accounting explicitly for multiple oceanic sources of periodic inter-annual natural variability, such as El Niño–Southern
Oscillation (ENSO) or the Atlantic Meridional Overturning Circulation (AMOC), the other model being SCM4OPT. While
FaIR v2.1 and WASP implement optional stochasticity to simulate internal variability, they lack the explicit connection to
oceanic variability present in EM-GC. In contrast, SCM4OPT takes a similar approach to EM-GC, and incorporates natural
variability through an ocean index, although it is limited to ENSO.

The EM-GC model has been utilised for various applications, including detrending the impacts of volcano eruptions (Canty
et al., 2013), conducting attribution analysis of global warming (Hope et al., 2020; McBride et al., 2021) and evaluating scenario
likelihoods to achieve climate goals (Hope et al., 2017, 2020; McBride et al., 2021; Farago et al., 2025).

First formulated by Canty et al. (2013), the core of the model has not seen major modifications since its inception. Hope
et al. (2017) compared its output with CMIP5 model results and added the effects of land-cover changes on albedo. Hope et al.
(2020) added a representation of the ocean mixed-layer heat content, improving the model's representation of heat exchange
between atmosphere and ocean through the modulation of the exchange based on the heat differential. McBride et al. (2021)
extended the historical record used for model calibration and updated the model to use Shared Socioeconomic Pathways (SSP)
scenarios (Meinshausen et al., 2020) instead of Representative Concentration Pathways (RCP) scenarios (Meinshausen et al.,
2011b). Finally, Farago et al. (2025) updated the expressions estimating GHG forcing to follow the equations given by the
Sixth Assessment Report (AR6) report (Forster et al., 2021b).



**Table 3.** Summary of temperature simulation schemes from the models reviewed in this study. This includes the type of module used for temperature estimation, the resolution of said module, and whether the model includes a representation of inter-annual variability beyond solar and volcanic forcing, which are often added externally as time series (see Table 2). EBM is shorthand for energy balance model, UD(E) for upwelling-diffusion(-entrainment), and IRM for impulse response model. The DOECLIM scheme combines a 0-D EBM with a 1-D diffusion scheme that simulates heat exchange with the deep ocean (Section 4.7.3).

| Temperature Simulation | | | |
|---|---|---|---|
| **Model name** | **Model type** | **Resolution** | **Variability** |
| ACC2 (V4.3) | DOECLIM | Global | - |
| AR5-IR | 2-time IRM | Global | - |
| CICERO-SCM (V1.1) | UD-EBM | Hemispheric | - |
| EM-GC | Multilinear regression | Global | Ocean indices |
| ESMICON | System dynamics heat model | Global | - |
| FaIR (V2.1) | 3-layer EBM | Global | Optional stochastic noise terms for temperature and forcing |
| GREB | Gridded 3-layer EBM | $3.75° \times 3.75°$ grid | - |
| Hector (V3.2) | DOECLIM | Global | - |
| Held et al. 2-layer model | 2-layer EBM | Global | - |
| MAGICC (V7) | UDE-EBM | Hemispheric | - |
| MCE (V1.2) | 3-time IRM | Global | - |
| OSCAR (V3.3) | 2-layer EBM | Global | - |
| SCM4OPT (V3.3) | DOECLIM | Global | ENSO index |
| WASP (V3) | 6-layer EBM with independent climate feedbacks | Global | Stochastic noise terms for temperature and forcing |

### 4.1.1 Model specification

The core of the model is defined by the following relationship between monthly temperature anomaly ($T$) and sources of radiative forcing:

$$
\begin{aligned}
T = \frac{1+\gamma}{\lambda_P}[F_i^{GHG} + F_i^{AER} + F_i^{LUC} - Q_i^{\text{ocean}}] + C_0 + C_1 \cdot SAOD_{i-6} + C_2 \cdot TSI_{i-1} \\
+ C_3 \cdot ENSO_{i-2} + C_4 \cdot AMOC_i + C_5 \cdot PDO_i + C_6 \cdot IOD_i
\end{aligned}
\tag{18}
$$

The terms between squared brackets include anthropogenic sources of radiative forcing - elevated GHG concentrations ($F_i^{GHG}$), aerosols ($F_i^{AER}$), LULCC changes in albedo ($F_i^{LUC}$) - as well as the ocean heat sink ($Q_i^{\text{ocean}}$). These terms are multiplied by a $(1+\gamma)/\lambda_P$ factor to account for climate feedbacks, where $\lambda_P = 3.2 Wm^{-2°}C^{-1}$ is the Planck feedback parameter, determining the temperature response in the absence of feedbacks, and $\gamma$ is a calibrated factor determining the strength of those feedbacks. The GHG term comprises forcings from $CO_2$, $CH_4$ (including 15% increase from stratospheric



**Table 4.** Summary of technical details about the models reviewed in this study, including available open source implementations, programming languages used to develop them and time resolutions.

\* This model supports variable timesteps, the shown value reflects the resolution most commonly used.

\*\* OSCAR can run internally with sub-annual timesteps, but output is annual.

**Technical details**

| Model name | Open source implementation | Language | Temporal resolution |
|---|---|---|---|
| ACC2 (V4.3) | - | GAMS | 1 year |
| AR5-IR | - | - | 1 year |
| CICERO-SCM (V1.1) | https://github.com/ciceroOslo/ciceroscm | Python | 1 year |
| | Executables: https://github.com/openscm/openscm-runner | Fortran | 1 year |
| EMGC | - | IDL | 1 month |
| ESCIMO | http://www.2052.info/ESCIMO/ | Vensim | 1 year |
| FaIR (V2.1) | https://github.com/OMS-NetZero/FAIR | Python | 1 year |
| GREB | https://github.com/christianstassen/greb-official | Fortran | 12h |
| Hector (V3.2) | https://github.com/JGCRI/hector | C++ / R wrapper | 1 year |
| | https://github.com/openclimatedata/pyhector | Python wrapper | 1 year |
| Held et al., 2-layer model | https://github.com/openscm/openscm-twolayermodel | Python | 1 year |
| MAGICC (V7) | https://gitlab.com/magicc/magicc | Fortran | 1 month |
| | https://github.com/openscm/pymagicc | Python wrapper | 1 month |
| MCE (V1.2) | https://github.com/tsutsui1872/mce | Python | 1 year* |
| OSCAR (V3.3) | https://github.com/tgasser/OSCAR | Python | 1 year** |
| SCM4OPT (V3.3) | https://github.com/sooxm/scm4eco | GAMS | 1/6 year |
| WASP (V3) | https://github.com/WASP-ESM/WASP_Earth_System_Model | C++ | 1 month* |

water vapour), $N_2O$, and 31 halogenated compounds. In early versions of the model, the equations from Myhre et al. (1998) were used to estimate the forcing from these species concentration time series, along with radiative efficiencies from WMO (2018). Forcing related to tropospheric ozone was also added by McBride et al. (2021) to this term directly as an additional forcing time series. Similarly, the aerosol and LULCC terms are included in the model directly as forcing time series data. The

former is taken from a combination of the SSP and RCP databases, while the latter is based on Table AII.1.2 in IPCC (2013). In the latest version of the model (Farago et al., 2025), GHG forcing is estimated following the equations from chapter 7 and annex III of the AR6 report (Forster et al., 2021b).

The ocean heat sink is a prognostic quantity in the model which, since Hope et al. (2020), is calculated by computing the difference between temperature anomaly in the atmosphere and in the OML (the deep ocean is not considered in this model).

This difference is further modulated by an ocean heat uptake efficiency that varies according to the accumulated ocean heat content and past radiative forcing. The ocean heat content is a key metric of the model, as it is one of the two quantities, along with global temperature anomaly, that is used to calibrate it.





The terms outside the brackets represent natural sources of variability: volcanos through stratospheric aerosol optical depth ($SAOD_{i-6}$), total solar irradiance ($TSI_{i-1}$), El Niño–Southern Oscillation ($ENSO_{i-2}$), the Atlantic Meridional Overturning Circulation ($AMOC_i$), the Pacific Decadal Oscillation ($PDO_i$), and the Indian Ocean Dipole ($IOD_i$). These are time series required as input by the model. Detailed explanations of the indices used to characterise each of these processes can be found in the studies cited above. It is worth noting that these sources of natural variability are typically turned off when the model is used to make future climate predictions due to the difficulty in deriving a timeseries that can reasonably predict the behaviour of these natural processes. $C_{0-6}$ are calibration coefficients determined by minimising a cost function defined as the difference between model predictions and historical observations for temperature and ocean heat content. The subscripts $i$ reference the monthly resolution of the model, and the time lags between the sources of radiative forcing and their effects (e.g., the model assumes that the volcano contribution, $SAOD_{i-6}$, takes six months to take effect).

### 4.2 AR5-IR

One of the many contributions of the IPCC Fifth Assessment Report (AR5) (Myhre et al., 2013) was the creation of a minimal set of equations to simulate the concentration, radiative forcing and temperature impact of $CO_2$ in the atmosphere (Myhre et al., 2013). The model extensively employs IRMs (see Sect. 3.3.3), using a 2-time IRF to estimate the temperature response to forcing and a 4-time IRF to simulate the atmospheric carbon sinks, hence the AR5-IR name. This transparent formulation was instrumental in generating the climate projections presented in the report and underpins the conclusions drawn from it.

In this review, the term "AR5-IR" follows the broader usage found in the FaIR literature, encompassing both the temperature and carbon sink components. This contrasts with the definition used in RCMIP Phase 2, where "AR5-IR" referred solely to the energy balance component without representation of the gas cycle. This adoption enables a clearer link to the FaIR SCM, which was initially developed as an extension of this broader "AR5-IR" model , and is reviewed later in section 4.3.

#### 4.2.1 GHG concentrations

The only GHG species included in AR5-IR is carbon dioxide. Its gas cycle is simulated by a 4-time-constant IRM based on the work of Joos et al. (2013), which argued that four time components are enough to emulate the evolution of atmospheric carbon concentrations from ESMs following a 100 GtC pulse. This approach can be interpreted as a distribution of the atmospheric carbon content into four different reservoirs ($R_i$), each governed by a mass balance equation as described in Sect. 3.1.1 (compare to Eq. 13):

$$\frac{dR_i}{dt} = a_i E - \frac{R_i}{\tau_i}, \, for \, i = 1,...,4 \tag{19}$$

where $a_i$ is the proportion of the total anthropogenic emissions ($E$, in ppm per year) allocated to each reservoir. These reservoirs do not correspond to any single physical entity, but rather combine various atmospheric sinks operating at similar timescales.



Values for $a_{1-4}$ and $\tau_{1-4}$ are provided in Myhre et al. (2013), borrowed from Joos et al. (2013). Broadly speaking, these pools account for:

- Indefinite airborne fraction ($a_0 = 0.2173$ and $\tau_0 = $ infinite years - usually implemented as a large number to allow incorporation into an exponential-sum framework, e.g., $10^6$ years in FaIR v1.0 and $10^9$ years in FaIR v2.0).

  - Deep ocean sink ($a_1 = 0.2240$ and $\tau_1 = 394.4$ years)

  - Biospheric and thermocline sinks ($a_2 = 0.2824$ and $\tau_2 = 36.54$ years)

  - Rapid biospheric and ocean mixed-layer sink ($a_3 = 0.2763$ and $\tau_3 = 4.304$ years)

Once the different $R_i$ are calculated, the total atmospheric concentration of $CO_2$ is simply the sum of pre-industrial concentrations ($C_0$) and all considered reservoirs: $C(t) = C_0 + \sum_i R_i$.

#### 4.2.2 Radiative forcing

To compute the resulting radiative forcing from the previously-simulated carbon concentration, AR5-IR multiplies the atmospheric burden (the sum of all $R_i$ pools) by the carbon radiative efficiency ($A$). This factor represents the radiative forcing per additional unit of carbon mass, and is approximated by taking the limit of carbon concentration anomaly as it approaches 0 in the common logarithmic relationship of Myhre et al. (1998). This limit results in a radiative efficiency of carbon of $A = 1.7517 \cdot 10^{-15}\, W m^{-2} kg^{-1}$. The applicability of such scheme is limited to small perturbations, which is why SCMs typically use more complex forcing schemes with wider applicability.

#### 4.2.3 Temperature

The last step in the emissions-climate change chain is to calculate the increase in surface temperature resulting from this radiative forcing. AR5-IR follows Boucher and Reddy (2008) and employs a two-time-constant IRM to produce temperature anomaly estimation, taking $n = 2$ in Eq. 15. Similarly to the equivalent 2-layer model, this temperature response can be interpreted as the addition of two contributions: a fast contribution, including effects from atmosphere, land and the OML, and a slow contribution accounting for deep-ocean heat uptake.

### 4.3 FaIR

The Finite-amplitude Impulse Response model (FaIR) is an SCM primarily developed by researchers at the universities of Oxford and Leeds. Despite its short life, it has gained significant popularity among SCM users and the broader climate modelling community. This is likely due to its relative simplicity, accurate performance and ease of usability, as well as its status as an open-source model. FaIR has been used in IPCC reports, such as the Special Report on 1.5 °C (IPCC, 2018) and the Sixth
Assessment Report (IPCC, 2021), to estimate future increases in radiative forcing. Additionally, it was used in an analysis of the Global Methane Pledge (Forster et al., 2021a) and research on substituting Hydrofluorocarbons (HFCs) in air-conditioning units for propane (Purohit et al., 2022).





The initial version, v1.0 (Millar et al., 2017), extended the AR5-IR model (Myhre et al., 2013) by introducing a new parameter that allows carbon and climate feedbacks to influence the atmospheric carbon sinks. This version was limited to $CO_2$ as

the sole forcing agent. However, FaIR v1.3 (Smith et al., 2018) extended the model to include a comprehensive list of GHG species and radiative forcing agents. In particular, version 1.3 included a representation of 31 GHG species: $CO_2$, $CH_4$, $N_2O$, Kyoto Protocol covered species - HFCs, Perfluorocarbons (PFCs), $SF_6$  - and Montreal protocol covered species - Chlorofluorocarbons (CFCs), Hydrochlorofluorocarbons (HCFCs). It also accounted for non-GHG forcing agents such as tropospheric and stratospheric ozone, stratospheric water vapour, contrails, aerosols (including volcanogenic), black carbon on snow, land

use change and solar irradiance. This version also adopted the use of ERF, allowing the specification of agent-specific efficacies to modify the temperature response per unit of forcing (Hansen et al., 2005).

The increased complexity resulting from these extensions was addressed in v2.0 (Leach et al., 2021). This version significantly simplified the model by introducing a set of six simple equations to determine the behaviour and temperature impact of all GHG and aerosol species. One of these equations generalised the carbon representation from FaIR v1.0 to all GHG species,

while another equation generalised the conversion from atmospheric species concentrations to radiative forcing. Additionally, the number of layers in its EBM was increased from two to three.

The last published version, v2.1 (Smith et al., 2024), introduced stochastic elements to the climate module, increased the flexibility of the methane lifetime and generalised its treatment of aerosol-cloud interactions.

### 4.3.1   GHG concentrations

Initially, FaIR v1.0 extended the IRM used to simulate the carbon cycle in the AR5-IR model (Sect. 4.2.1) with an additional equation to allow climate- and carbon-carbon feedbacks. This scheme was later applied to all other included GHGs ($CH_4$, $N_2O$, and 40 other halogenated gases, as well as aerosols) in FaIR v2.0, albeit in a simplified manner. Specifically, FaIR v1.0 modified the IRM described by Eq. 19 in the AR5-IR model to include a state-dependent gas lifetime through the addition of a scale factor $\alpha$:

$$\frac{dR_i}{dt} = a_i E - \frac{R_i}{\alpha \tau_i}, \, for \, i = 1,...,N \tag{20}$$

and

$$C(t) = C_0 + \sum_{i=1}^{n} R_i(t) \tag{21}$$

which effectively alters the sink strength for that gas species. Similarly to AR5-IR, the number of carbon pools was set to four ($N = 4$), although this is user-definable, while the number of sinks for all other species is kept to one by default. Note

that since v2.0, the state-dependent factor $\alpha$ is also applied to all other species, but other than $CO_2$ and $CH_4$ the default $\alpha = 1$ parameter is not modified.





To determine the appropriate value of this parameter for carbon, FaIR v1.0 (Millar et al., 2017; Smith et al., 2018) used the 100-year integrated impulse-response function (iIRF$_{100}$) derived by Joos et al. (2013). This function multiplies the estimated average airborne fraction by the integration time over a 100-year time span, thereby capturing temporal variations in the

remaining airborne carbon. By equating this to a linear function dependent on temperature ($T$) and land-ocean carbon stock anomalies ($G_u$), the value of $\alpha$ can be determined at each each time step, incorporating temperature and carbon feedbacks into the gas cycle. However, solving this equation is computationally expensive, so from v2.0 (Leach et al., 2021) onwards a simplified exponential solution was adopted, which they present as a reasonable approximation for a "wide range of values":

$$\alpha(t) = g_0 \, exp\left(\frac{r_0 + r_u G_u(t) + r_t T(t) + r_a G_a(t)}{g_1}\right) \tag{22}$$

with

$$G_a(t) = \sum_{i=1}^{n} R_i(t)$$

$$G_u(t) = \sum_{s=t_0}^{t} E(s) - G_a(t)$$

$$g_0 = exp\left(-\frac{\sum_{i=1}^{n} a_i \tau_i (1 - e^{-100/\tau_i})}{g_1}\right)$$

$$g_1 = \sum_{i=1}^{n} a_i \tau_i \left(1 - (1 + 100/\tau_i)e^{-100/\tau_i}\right)$$

where $g_0$ and $g_1$ are new parameters controlling the magnitude and gradient of $\alpha$. The $r_a G_a(t)$ term represents the sensitivity of the gas species to its own atmospheric burden. This has a small effect on $CO_2$ atmospheric lifetime, but it is an important factor in methane lifetime. In v2.1 this formulation was further refined for methane, with a new atmospheric lifetime modulated by the burden of an arbitrarily large number of species:

$$\ln \alpha_{CH_4} = \ln(1 + r_t T(t)) + \sum_i \ln(1 + r_i G_i(t)) \tag{23}$$

where $r_i$ denotes the sensitivity to the abundance of species i, $G_i$. This $G_i$ represents either atmospheric concentrations for GHGs or emission rates for short-lived climate forcers, since their rapid decay prevents any significant accumulation in the atmosphere.

No lifetime sensitivities are assumed for nitrous oxide and halogen gases ($r_u, r_T, r_a = 0$). Lifetime estimates for these species are user-definable. The latest available calibration (Smith et al., 2024) employed the AR6 values (Smith et al., 2021a).



Before V2.1, aerosols were converted from emissions to concentrations by setting $\tau = 1$ and taking a conversion factor between emissions and concentrations of 1. Since V2.1, however, to account for their short lifetimes, the concentration step is bypassed and the forcing is computed using emissions directly.

Finally, while FaIR does not natively simulate permafrost thaw, Steinert and Sanderson (2025) extended v1.6 by coupling it with a simplified permafrost carbon response model.

### 585   4.3.2   Radiative forcing

To calculate the ERF, FaIR v2.0 generalised the expressions offered in Myhre et al. (2013) to a single equation, approximating the concentration-forcing relationships for all Well-mixed Greenhouse Gases (WMGHGs) (or emissions-forcing in the case of aerosols due to their short atmospheric lifetimes) by:

$$F_{det}(t) = \sum_{x}^{\substack{\text{forcing} \\ \text{agents}}} \left[ f_1^x \cdot \ln\left( \frac{C^x(t)}{C_0^x} \right) + f_2^x \cdot (C^x(t) - C_0^x) + f_3^x \cdot \left( \sqrt{C^x(t)} - \sqrt{C_0^x} \right) \right] + F_{ext}(t) \tag{24}$$

Each term is multiplied by an $f_i^x$ factor, allowing the model to account for indirect effects by modulating the direct forcing effects from GHG concentration (i.e., generating ERF estimations). This factor also enables the model to completely switch off a term for a given species. For instance, $CO_2$ forcing is approximated by a logarithmic and squared root term (IPCC, 2001), so $f_2^{CO_2} = 0$. Methane and nitrous oxide contributions are approximated by the square-root term exclusively, $f_{1,2}^{CH_4, N_2O} = 0$. Equally, setting $f_{1,3}^x = 0$ reduces Eq. 24 to the common linear expression often used for minor GHGs, with $f_2^x$ becoming the 595   radiative efficiency. This is used by FaIR to approximate the contribution from halogenated gases and the direct effects of aerosols (scaling with sulfate, organic carbon, and black carbon emissions). $F_{ext}$ covers any exogenous forcing, such as natural forcings (volcanic activity and solar cycles) and albedo effects. These are included in the model directly as forcing time series.

Beyond Eq. 24, FaIR also includes other sources of radiative forcing which may depend on one or multiple species. It parametrises both tropospheric and stratospheric ozone contributions following Thornhill et al. (2021) as a linear function of 600   methane; nitrous oxide and Ozone-Depleting Substances (ODSs) concentrations; as well as nitrate aerosol, carbon monoxide, and Volatile Organic Compounds (VOCs) emissions. Contributions from stratospheric water vapour, black carbon on snow, and aviation contrails are scaled linearly with, respectively, tropospheric methane concentrations, black carbon emissions, and aviation sector $NO_x$ emissions. Finally, indirect aerosol forcing effects due to cloud interactions are approximated as the addition of a logarithmic term from sulfate aerosol emissions and a linear term from organic carbon and black carbon emissions.

FaIR v2.1 increased the model's flexibility by implementing the other three main approaches to radiative forcing: Myhre et al. (1998), Etminan et al. (2016) and Meinshausen et al. (2020). Users can choose which scheme to use, with the model defaulting to Meinshausen et al. (2020) as the most accurate among the four. The expression computing the forcing from aerosol-cloud interaction was also updated following Smith et al. (2021b), generalising it to potentially include the effects from more species.





### 4.3.3 Temperature

FaIR calculates the temperature response using the IRM formulation of an $n$-layer EBM, similar to AR5-IR (see Sect. 3.3.3). However, successive versions of FaIR have increased the complexity of the EBM. V2.0 increased the number of temperature components (or layers) from two to three, following the findings of Tsutsui (2017, 2020) and Cummins et al. (2020), which suggest three layers are better suited to emulate impulse-like forcing scenarios. Subsequently, version 2.1 adopted the model of Cummins et al. (2020), incorporating stochastic terms in the temperature and radiative forcing responses, as well as allowing for an arbitrarily large number of ocean layers greater than two. In the equivalent $n$-layer formulation, the three-layer case can be expressed as:

$$
\begin{aligned}
C_1 \frac{dT_1(t)}{dt} &= F_{\text{tot}}(t) - k_1 T_1(t) - k_2(T_1(t) - T_2(t)) + \xi(t) \\
C_2 \frac{dT_2(t)}{dt} &= k_2(T_1(t) - T_2(t)) - \epsilon k_3(T_2(t) - T_3(t)) \\
C_3 \frac{dT_3(t)}{dt} &= k_3(T_2(t) - T_3(t))
\end{aligned}
\tag{25}
$$

where $C_{1-3}$, $k_{1-3}$ and $T_{1-3}$ denote the heat capacities, heat transfer coefficients with the layer above ($k_1$ being the climate feedback parameter) and temperature of the three layers respectively. $\epsilon$ is the deep ocean efficacy parameter (Held et al., 2010; Geoffroy et al., 2013a) as discussed in Sect. 3.3.2. The only difference with the standard 3-layer EBM, as described in Eq. 11, is the addition of two stochastic disturbances emulating climate's internal variability: one directly affecting the temperature response ($\xi$), and another affecting the total radiative forcing term ($F_{\text{tot}}$), which results from the combination of the deterministic ERF determined earlier ($F_{\text{det}}$) and a red-noise component ($\zeta$) simulating time-correlated variations from the mean ($\zeta$):

$$
F_{\text{tot}} = F_{\text{det}} + \zeta
\tag{26}
$$

$$
\frac{d\zeta}{dt} = -\gamma\zeta + \eta
\tag{27}
$$

where $\gamma$ is a parameter controlling the degree of temporal auto-correlation and $\eta$ represents a white noise addition.

### 4.4 MCE

The Minimal CMIP Emulator (MCE), developed by Dr. Junichi Tsutsui at the Central Research Institute of Electric Power Industry, Japan, combines a three-constant IRM EBM with a carbon cycle component that utilises both a box-based scheme and an IRF scheme. Tsutsui (2020) employed this model to estimate ECS and TCR values using output from CMIP5 and Coupled Model Intercomparison Project Phase 6 (CMIP6) ESMs. Notably, MCE was instrumental in demonstrating that a minimum of three characteristic timescales/boxes in EBMs is required to accurately approximate short-term ESM temperature



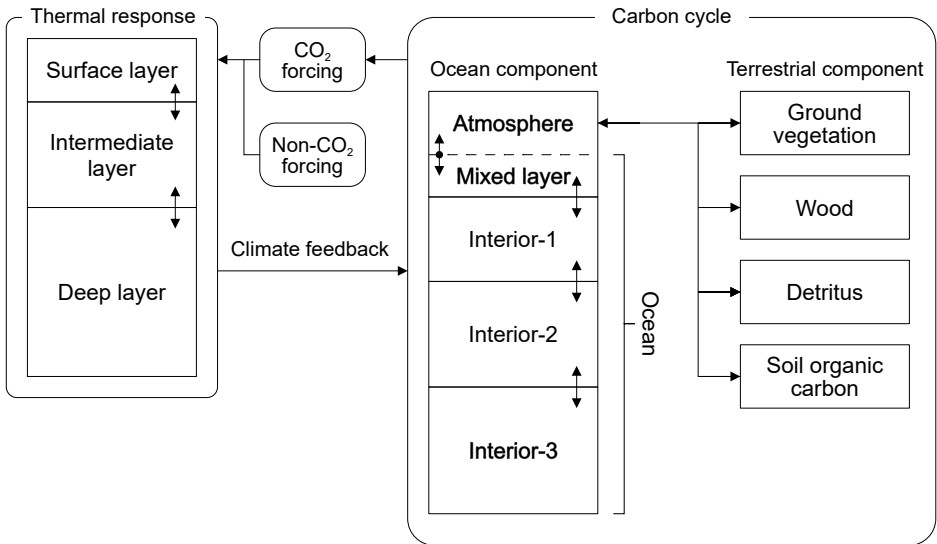

**Figure 3.** Illustration of model structure in MCE v1.2. The SCM consists of a three-layer IRM to calculate the thermal response and a carbon cycle model comprising a four-time IRM for the land and a four box model for the ocean. Figure based on Tsutsui (2022, Fig. 1).

response following instantaneous radiative forcing changes (Tsutsui, 2017), which is particularly relevant for abrupt forcing

scenarios such as volcanic eruptions, geoengeneering, and certain idealised model scenarios (e.g., abrupt $CO_2$ doublings and quadruplings).

Originally introduced by Tsutsui (2017), MCE was initially positioned at the simpler end of the SCM complexity spectrum, driven only by two core equations: a bi-modal forcing expression to convert $CO_2$ concentrations into radiative forcing, and a three-layer IRM to translate that forcing into surface temperature anomalies. Recently, Tsutsui (2022) presented version 1.2

of the model, which incorporates a more sophisticated carbon cycle and non-$CO_2$ forcing calculations. This updated version employs a four-time IRM for the land component based on Joos et al. (1996), and a four-box model for the ocean component based on Hooss et al. (2001). Fig. 3 provides an illustration of this enhanced model structure.

### 4.4.1 GHG concentrations

The new carbon cycle module was introduced in the latest version of the model, v1.2 (Tsutsui, 2022), where further details can

be found. Fundamentally, this module consists of a four-box model for the ocean-atmosphere system and an impulse response scheme for the land component. Sections 3.1.2 and 3.3.3 provide more details on these types of models.

Starting with the ocean-atmosphere component, this is implemented as a four-box scheme (analogous to Eq. 11 with $n = 4$):



$$\frac{dc_0}{dt} = -\frac{\eta_1}{h_s}c_s + \frac{\eta_1}{h_1}c_1 + E - f$$

$$\frac{dc_1}{dt} = \frac{\eta_1}{h_s}c_s - \frac{\eta_1 + \eta_2}{h_1}c_1 + \frac{\eta_2}{h_2}c_2$$

$$\frac{dc_2}{dt} = \frac{\eta_2}{h_1}c_1 - \frac{\eta_2 + \eta_3}{h_2}c_2 + \frac{\eta_3}{h_3}c_3 \qquad (28)$$

$$\frac{dc_3}{dt} = \frac{\eta_3}{h_2}c_2 - \frac{\eta_3}{h_3}c_3$$

where $c_k$ is the excess carbon in layer $k$, $h_k$ is the depth of the layer, $\eta_k$ is the exchange coefficient between layers $k-1$ and

$k$, $E$ represents anthropogenic emissions, and $f$ denotes the carbon uptake by the land component. A peculiarity of this model is that the top layer ($c_0$) represents both the atmospheric and ocean mixed layer carbon pools. Consequently, $c_0$ is partitioned into atmospheric ($c_a$) and oceanic ($c_s$) excess carbon, with the distribution between them being calculated through a complex chemical equilibrium scheme that includes temperature feedbacks. Parameters $h_k$ and $\eta_k$ have been calibrated such that the evolution of $c_0$ tracks an equivalent four-constant IRM for airborne fraction in Hooss et al. (2001). The characteristic timescales

are set to 1.271, 12.17, 59.52 and 236.5 years, calibrated on a three-dimensional ocean carbon cycle model (Hooss et al., 2001).

Terrestrial carbon uptake is governed by an IRM with four characteristic timescales ($\tau_i$), corresponding to four categories of land carbon: vegetation, wood, detritus and soil organic carbon. Carbon input to the terrestrial system originates from an NPP flux, which is modulated by a sigmoid function dependent on atmospheric $CO_2$ concentration to account for fertilisation effects ($\beta_f([CO_2])$). Thus, the land carbon uptake is expressed as:

$$f(t) = \sum_{i=0}^{4}\frac{dc_i}{dt} = \sum_{i=0}^{4}\left(\beta_f(t)\text{NPP}_0\tilde{A}_i\tau_i - \frac{c_i}{\tau_i}\right) \qquad (29)$$

where $\text{NPP}_0$ is the pre-industrial net primary production, $c_i$ is the carbon anomaly of the $i^{\text{th}}$ land carbon category and $\tilde{A}_i$ is the amplitude of the $i$th IRF. Notice that the coefficient $A_i$ in Eq. 13 corresponds to the $\tilde{A}_i\tau_i$ product in Eq. 29. Values for $\tau_i$ and $\tilde{A}_i$ are set to 2.9, 20, 2.2 and 100 years and 0.70211, 0.013414, -0.71846, and 0.0029323 yr[-1], respectively. These values were borrowed from Joos et al. (1996), where the IRM was originally presented.

The model's representation of non-$CO_2$ sources of radiative forcing remains limited. It does not include non-$CO_2$ gas cycles, requiring the use of prescribed concentration time series for $CH_4$, $N_2O$ and halogenated gases to calculate the resulting radiative forcing. This is set to change in a future V1.3 version, where a simple gas cycle model will predict non-$CO_2$ concentrations from emissions (private communication). Similarly, MCE includes radiative impacts of tropospheric and stratospheric ozone; stratospheric water vapour; aerosols, including volcanic aerosols; solar irradiance; contrails; and LULCC albedo as prescribed

forcing time series.

### 4.4.2  Radiative forcing

The radiative forcing from $CO_2$ is calculated using the standard logarithmic expression (Myhre et al., 1998) for concentrations up to twice the pre-industrial:



$$F^{CO_2}(x) = \alpha \ln\left(\frac{C^{CO_2}(t)}{C_0^{CO_2}}\right) \tag{30}$$

where $\alpha$ is a scaling parameter and $x$ is the ratio of $CO_2$ concentration relative to the pre-industrial level. For higher concentrations, up to four times the pre-industrial level, MCE uses:

$$\tilde{F}_{2<x<4}^{CO_2}(x) = (\beta-1)(F^{CO_2}(x) - 2F^{CO_2}(2)) \cdot \left(\frac{2F^{CO_2}(x)}{F^{CO_2}(2)} - 1\right) + \beta F^{CO_2}(x) \tag{31}$$

    where $\beta = \tilde{F}^{CO_2}(4)/F^{CO_2}(4)$ represents a model-dependent scaling factor for the transition from the first to the second doubling of $CO_2$. Both $\alpha$ and $\beta$ have been calibrated to replicate results from CMIP models (Tsutsui, 2020). For $x > 4$, the

quadratic term in Eq. 31 is omitted, and Eq. 30 is reused with an adjustment such that the forcing is continuos at $x = 4$.

    If non-$CO_2$ concentration time series are provided, MCE uses the expressions from Etminan et al. (2016) to calculate the forcing from methane and nitrous oxide, and from Myhre et al. (2013) for halogenated gases.

### 4.4.3 Temperature

To translate the total radiative forcing ($F$) into a temperature anomaly ($T$), the model's default is a three-time IRM (a two-time

IRM is also available) as described in Eq. 12:

$$T(t) = \int\limits_0^t \frac{F(t')}{\lambda} \sum_i^3 \frac{A_i}{\tau_i} exp\left(-\frac{t-t'}{\tau_i}\right) dt', \tag{32}$$

    where $\tau_i$ and $A_i$ are the characteristic times and amplitudes of the $i^{\text{th}}$ decaying exponential used in the IRM, and $\lambda$ is the climate feedback parameter. The three characteristic times $\tau_i$ are approximately 1, 10 and >100 years. Notably, similar to Eq. 29, the amplitudes from Eq. 12 have been slightly redefined in Eq. 32, so $A_i$ in Eq. 12 correspond to $A_i/\lambda\tau_i$ in Eq. 32.

## 4.5 WASP

The Warming, Acidification and Sea-level Projector (WASP) is an SCM developed by Dr. Philip Goodwin at the University of Southampton. The model is characterised by its box-model framework, which is applied both to its carbon cycle and its EBM, with a particular focus on the oceanic component of the climate system. WASP has played a fundamental role in various studies, including an examination of the surface warming after cessation of carbon emissions (Williams et al., 2017), the generation

of climate projections based on a history-matching model calibration (Goodwin et al., 2018) and a cost analysis of adaptation strategies to sea-level rise for different scenarios (Brown et al., 2021).

    WASP V1 was initially presented by Goodwin (2016) as an 8-box model simulating the carbon flows through the atmosphere-land-ocean system and the heat exchange between atmosphere and ocean, as depicted in Fig. 4. A comprehensive mathematical



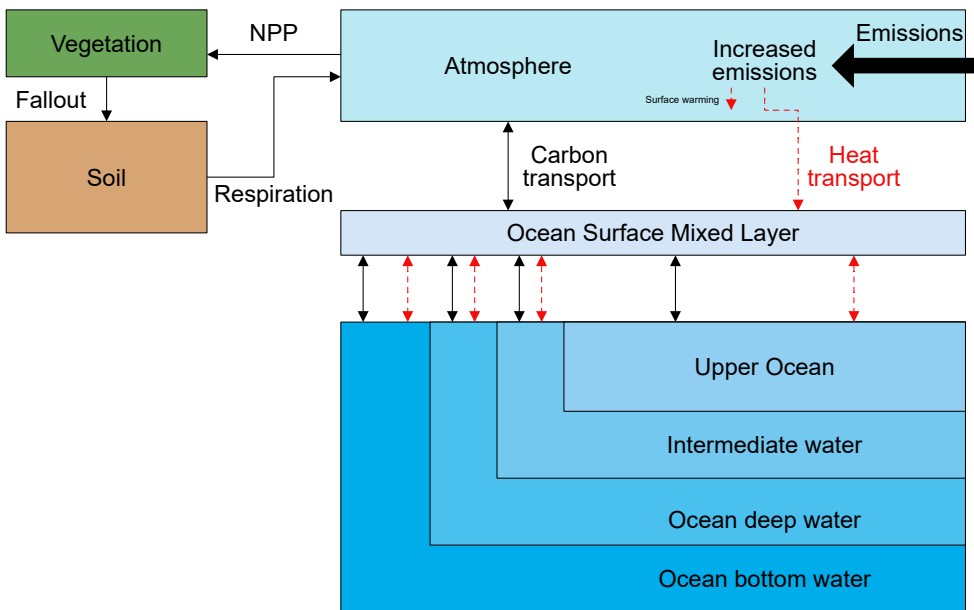

**Figure 4.** Diagram of the carbon and heat flows in the WASP SCM. Image based on Goodwin (2016, Fig. 2).

specification of the model is provided in its appendix. Subsequently, Goodwin et al. (2017) augmented the model to include

a representation of global sea-level change, including thermosteric and isostatic contributions. The following year, WASP V2 (Goodwin, 2018) further refined the model by incorporating time-evolving climate feedback parameters in its EBM, and by including volcanic and solar forcings. Goodwin et al. (2018) introduced stochasticity to the model temperature response. Finally, WASP V3 (Goodwin and Cael, 2021) increased the granularity of the model's forcing representation, separating the effects from different forcing agents, updated the model to use SSP scenarios instead of RCPs, and included stochasticity in

the model's forcing representation.

### 4.5.1   GHG concentrations

The only species whose atmospheric concentration is computed prognostically in WASP is carbon dioxide. All other species are included indirectly, either through radiative forcing data in V1 and V2, or through prescribed concentration data in V3. WASP's carbon cycle is an eight-box scheme, with five of those boxes dedicated to the ocean. For all boxes, carbon stocks are

tracked using the carbon anomaly with respect to pre-industrial stocks.

     The key mechanism determining the distribution of anthropogenic carbon across the system in this SCM is ocean carbon undersaturation (Goodwin et al., 2015). This measure indicates the amount of carbon the ocean needs to absorb to reach equilibrium with the atmosphere, and is calculated using a relatively complex scheme depending on atmospheric carbon content, cumulative airborne emissions, an equivalent carbon emissions term accounting for the ocean temperature-$CO_2$ solubility feed-





back (Goodwin and Lenton, 2009) and the buffered carbon inventory (Goodwin et al., 2007). The further the ocean is from equilibrium, the more carbon it absorbs, with the evolution of ocean layers towards an equilibrium with the OML and the atmosphere modelled via decaying exponentials. How rapidly each layer approaches equilibrium is dictated by the volume of each box and the characteristic restoring e-folding timescale of each ocean layer. This model also uses the amount of carbon in the atmosphere and the carbon undersaturation in the OML to simulate ocean acidification, using the carbonate chemistry

solver by Follows et al. (2006) to estimate the pH change.

Comparatively, the land carbon cycle is simpler than its ocean counterpart. NPP transfers carbon from the atmosphere to a vegetation box, which eventually decays to a soil box through a litterfall flux and returns to the atmosphere via a heterotrophic respiration flux. The magnitude of NPP is influenced by temperature (linear dependence) and $CO_2$ fertilisation effects (log dependence). Similarly, the effects of surface warming on heterotrohic respiration are incorporated through a linear influence

on soil residence time.

### 4.5.2 Radiative forcing

Originally, WASP included three sources of radiative forcing. First, atmospheric carbon dioxide, whose radiative forcing is estimated via the logarithm of the increase since pre-industrial concentrations. Second, Kyoto-protocol agents (WMGHGs and CFCs), taking a forcing timeseries from RCP scenarios as input. Third, non-Kyoto-protocol agents (mainly aerosols),

scaled proportionally to Kyoto-protocol agents. This division is still employed if the model is used to run any RCP scenario. However, since its update to SSP scenarios in Goodwin and Cael (2021), WASP's representation of radiative forcing takes non-$CO_2$ species concentrations as input and distinguishes between the following agents: $CO_2$, calculated following the log-arithmic expression in Myhre et al. (2013); $CH_4$ and $N_2O$, following Etminan et al. (2016); 27 halogenated species, using radiative efficiencies from Smith et al. (2018); and aerosols, including both direct and indirect contributions from black carbon,

organic carbon, sulphates, nitrous oxides, ammonia and VOCs, borrowing the scheme from FaIR V1.3 (Smith et al., 2018). Additionally, since Goodwin (2018), the model includes the effects of solar and volcanic radiative forcings via forcing time series.

To represent the internal variability in Earth's energy imbalance, Goodwin and Cael (2021) included a noise term in the model's radiative forcing with parameters tuned to emulate the monthly and annual root-mean-square energy imbalance in

Trenberth et al. (2014).

### 4.5.3 Temperature

WASP follows the common energy balance model approach (see Eq. 7), with the increase in surface temperature being pro-portional to the total radiative forcing minus the heat absorbed by the Earth system, which in this model is limited to the ocean. To determine this heat uptake, a similar approach to the carbon cycle is employed: total radiative forcing pushes the

heat content of the atmosphere away from equilibrium with the OML, inducing a heat flux into this mixed layer and the four deeper ocean layers. This disequilibrium is quantified by computing the eventual heat content of the OML required to balance the instantaneous total radiative forcing through a linear equivalence relationship. The ocean heat uptake at each time step is





then determined based on the difference between the current heat content of the OML and its equilibrium heat content. Similar to the carbon cycle, the flux of heat from the OML to deeper ocean layers is modelled by decaying exponentials that push the system towards a new equilibrium. The characteristic timescales for each ocean box are the same for both the heat and carbon schemes.

Goodwin (2018) further refined the model's EBM by incorporating $i$ forcing-agent-specific climate feedbacks that can vary independently on a set of $j$ feedback processes ($\lambda_{i,j}$), instead of a single climate feedback applicable to all forcing agents ($\lambda$). In practice, this translated to a reformulation of Eq. 7 into:

$$\Delta T = \left(1 - \frac{N(t)}{F(t)}\right) \sum_i \left(\frac{F_i}{\lambda_{planck} + \sum_j \lambda_{i,j}(t)}\right) \tag{33}$$

where $N$ is the heat flux towards the surface ($dH/dt$ in Eq. 7), $\lambda_{planck}$ is the Planck climate sensitivity (Caldwell et al., 2016), and $F$ and $F_i$ are the total and agent-specific radiative forcings. Values for the new parameters were constrained by constructing a large ensemble of simulations using ranges of climate feedbacks from CMIP5 models and applying observational constraints.

The EBM described so far produces a deterministic temperature response. However, since Goodwin et al. (2018), this response has been further modified by a stochastic source of variability. Specifically, a noise term was introduced for both surface air temperature and sea surface temperature, which was calibrated to reproduce the magnitude and auto-correlation properties observed during the historical period.

## 4.6 GREB

As the sole model in this review with an explicit grid division of the Earth's surface, the Globally Resolved Energy Balance (GREB) serves as a bridge between globally-averaged SCMs and general circulation models (not ESMs, as GREB does not include a carbon cycle). While other SCMs, such as OSCAR and Hector, use box models to represent biomes, and most models use an EBM with multiple layers, GREB is the only SCM in this review employing an explicit grid to simulate Earth's climate. Additionally, GREB has been designed primarily to enhance the physical understanding of processes driving the mean climate state, particularly for university teaching. As a result, it resolves a somewhat atypical list of processes: solar and thermal radiation, hydrological cycle, sensible heat, atmospheric circulation simulating advective and diffusive transport, sea ice, and heat absorption by the subsurface ocean (deep ocean is not considered).

The GREB model has been used in several studies: Dommenget et al. (2019) used it to create a climate scenario database and to enhance the understanding of processes driving the mean climate state; Latif et al. (2023) investigated the impacts of wind-induced latent heat flux changes on the sea surface temperature of the Pacific ocean using GREB; and Xie and Dommenget (2023) explored climate-ice sheet feedbacks with GREB-ISM (Xie et al., 2022), a coupling of GREB with an ice-sheet model.

The initial version of GREB was presented by Dommenget and Flöter (2011), with its hydrological cycle being further refined by Stassen et al. (2019) to improve representations of precipitation, evaporation and horizontal transport of water vapour.





### 4.6.1 Model description

Lacking a representation of any gas cycles beyond water vapour dynamics (see hydrological cycle below), GREB estimates global temperature anomaly through a 3-layer EBM (atmosphere, OML, subsurface ocean). Its resolution is unusually high for an SCM, operating at 12h timesteps in a $3.75° \times 3.75°$ cell grid. For each cell, the surface temperature anomaly is governed by:

$$C\frac{dT_{surf}}{dt} = F_{solar} + F_{thermal} + F_{latent} + F_{sense} + F_{ocean} + F_{ice} + F_{correct} \tag{34}$$

These are all the forcing agents included in the model, which also determine the remaining prognostic variables: atmosphere temperature anomaly, subsurface temperature anomaly, humidity of surface layer, and thickness of ice cover since GREB-ISM (Xie et al., 2022). Processes associated with these agents are described below, except $F_{correct}$, which is an empirical heat flux to correct for model error. The cell heat capacity, $C$, varies depending on the nature of the cell (ice-free ocean, frozen ocean or land). Note this is the same equation describing the typical EBM (see Eq. 8) with the only difference being that the temperature

response ($\lambda T$) is part of the forcing terms in Eq. 34. Table 1 of Dommenget and Flöter (2011) offers a list of all prognostic and diagnostic GREB's variables, as well as the required boundary conditions.

These forcing terms are determined by several parameterised processes, which are illustrated in Fig. 2 from Dommenget and Flöter (2011). These are:

– Solar radiation: the absorbed incoming solar radiation ($F_{solar}$) is determined by the 24-hour average of radiation reaching
the surface, modulated by the day of the year and surface as well as cloud albedo effects.

– Thermal radiation: the net thermal radiation ($F_{thermal}$) is the difference between the black body emission from the Earth's surface and the downward thermal radiation from the atmosphere. The latter depends on atmospheric temperature, $CO_2$ concentration (only GHG included in the model, required as input), vertical integrated atmospheric water vapour concentration, and cloud cover. Although this is the forcing contribution closest to other SCM forcing, the atypical nature
of the included processes results in slightly atypical parametrisations for an SCM.

– Hydrological cycle: although parameterised, GREB is the only SCM to include a hydrological cycle. The scheme used to simulate it is relatively complex, resolving three main processes: evaporation, precipitation and moisture transport. Evaporation is simulated via a bulk formula approach, which, in turn, determines the latent heat release to the surface layer ($F_{latent}$). Precipitation is governed by the upward motion of air and its humidity (a prognostic variable of the
model), while moisture transport is modelled through advection and diffusion processes, following mean winds. Although present in Dommenget and Flöter (2011), these processes were further refined in Stassen et al. (2019), which is the best resource for more details about GREB's hydrological cycle.

– Sensible heat: the amount of sensible heat exchanged between the surface and the atmosphere ($F_{sense}$) is parameterised via the difference between surface and atmospheric temperatures. Atmospheric temperature is a prognostic property of





the model that depends on sensible heat exchange with the surface, thermal atmospheric radiation, latent heat released by water condensation in the atmosphere, and atmospheric circulation.

– Atmospheric circulation: the model includes a seasonal mean atmospheric circulation independent from forcing. Horizontal transport of heat and humidity is determined via diffusion and advection parameterisations and is further modulated by topographical effects.

– Subsurface ocean: the heat exchange between the subsurface and the OML ($F_{ocean}$) in GREB is determined by the difference in temperature between the two ocean layers, which controls the amount of turbulent mixing and deeper-water entrainment into the surface layer. Notice that GREB does not consider the impacts of the abyssal ocean, with the maximum depth of the subsurface ocean layer being only three times the OML depth (between 100 and 1000m), hence the reference to a subsurface ocean, rather than a deep ocean. The subsurface ocean temperature ($T_{ocean}$) is another

prognostic property of the model, which depends on the amount of heat absorbed since the beginning of the simulation. Finally, it is worth noting that GREB also includes an additional empirical ocean heat flux to counteract model drifts in the ocean temperature.

– Ice sheets: Xie et al. (2022) coupled the previously described model with an ice sheet model, creating GREB-ISM. This enhanced version of GREB incorporates three types of ice surfaces: land ice, floating ice (ice shelves), and ice over

ocean. The thickness of these ice layers is a prognostic variable within the coupled model, evolving in response to the climatology provided by the original GREB model. In turn, the ice module introduces an additional heat flux to Eq. 34 ($F_{ice}$), representing the impact of ice on heat exchange with the atmosphere. Furthermore, the presence of dynamic ice sheets influences both albedo and topography of the climate module. Fig. 2 in Xie et al. (2022) offers an illustration of the coupling between the ice sheet model and GREB, outlining the exchange of properties between the two components

that conform GREB-ISM.

## 4.7 Hector

Hector is a box-based SCM developed at the Pacific Northwest National Laboratory, emphasizing modularity and clearly defined interfaces to support flexible integration and development. (Hartin et al., 2015). (Hartin et al., 2015). Beyond the open-source versions presented in Table 4, this model also includes an interactive online version at https://jgcri.shinyapps.io/

HectorUI/.

The model has been instrumental in multiple studies, including the analysis of gas cycles and climate responses from SCMs (Schwarber et al., 2019); examination of the effects of climate sensitivity on sea-level change (Vega-Westhoff et al., 2019); and emulation of ESM output for different RCP scenarios (Dorheim et al., 2020). Additionally, it has served as the default climate module in the Global Change Analysis Model IAM (GCAM, Joint Global Change Research Institute, 2023) since 2015

(GCAM-4.3, Calvin et al., 2019), an open-source multisector model with representations of the economy, energy, agriculture, and water supply in 32 geopolitical regions across the globe.




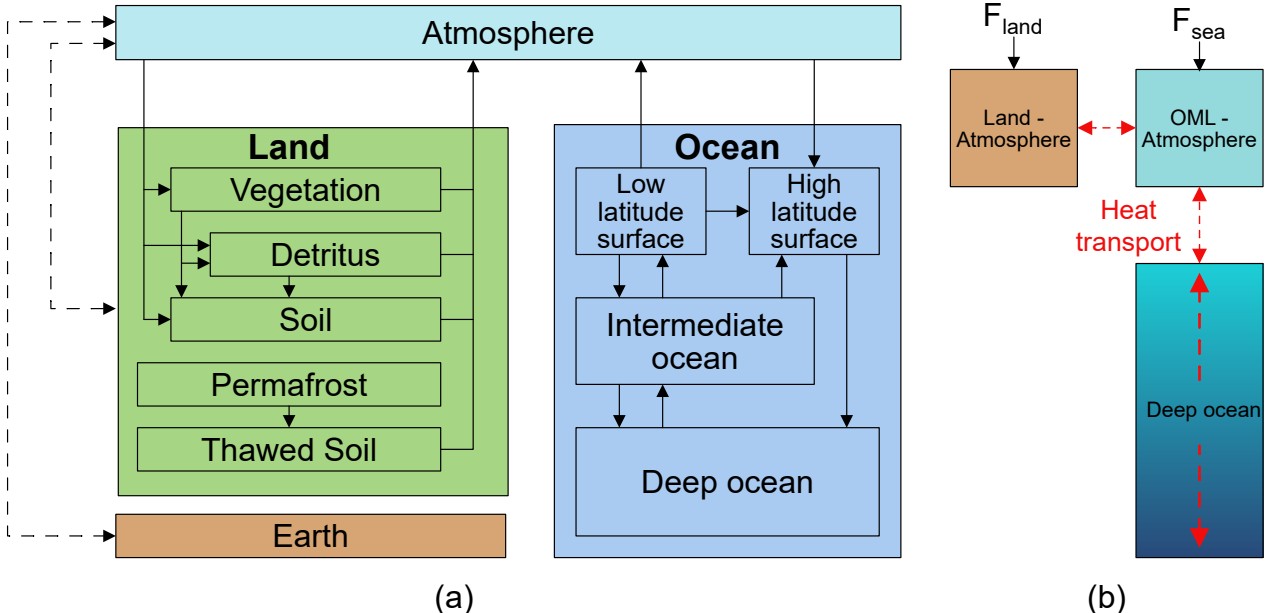

**Figure 5.** Hector's diagrams. (a) The carbon cycle. Solid arrows indicate flows simulated internally by the system, while dashed arrows denote externally supplied fluxes. Image based on Dorheim et al. (2024b, Fig. 1). (b) Hector's temperature module: the DOECLIM scheme. Radiative forcings over land ($F_{land}$) and ocean ($F_{ocean}$) are applied to the combined land-atmosphere and ocean-mixed-layer-atmosphere boxes. The latter is also coupled to a 1-D diffusion ocean, with constant diffusivity across the water column.

Originally introduced by Hartin et al. (2015) as a simple global climate-carbon model incorporating atmospheric, terrestrial and oceanic carbon pools, Hector saw its first major update with version 1.1 (Hartin et al., 2016), which implemented a new carbonate scheme for the upper ocean. Version 2.0 (Vega-Westhoff et al., 2019) brought significant enhancements, including the integration of the DOECLIM EBM (Kriegler, 2005), and a new sea-level component based on Wong et al. (2017) accounting for five different contributions: thermal expansion, glaciers and small ice caps, the Greenland ice sheet, the Antarctic ice sheet, and changes in land water storage. The latest iteration, Hector V3.2, described in Dorheim et al. (2024b), features a new permafrost module (Woodard et al., 2021), some minor changes to the carbon cycle and radiative forcing components, and the novel capability to track the movement of carbon through different pools (Pressburger et al., 2023).

### 4.7.1 GHG concentrations

Hector includes a representation of the gas cycle for $CO_2$, $N_2O$, $CH_4$ and 27 halocarbons. Atmospheric concentrations for all these species except $CO_2$ are governed by the mass balance equation detailed in Sect. 3.1.1, incorporating anthropogenic emissions and sinks. Carbon concentrations are determined by the box carbon cycle model described below. Additionally, Hector estimates concentrations of tropospheric ozone based on methane concentrations and emissions of nitrogen oxides, carbon monoxides, and Volatile Organic Compounds (NMVOCs). For full details, see section 4 in Hartin et al. (2015).



The carbon cycle in Hector has remained largely unchanged since V1. There are four categories of carbon reservoirs: a well-mixed atmosphere, land, ocean and Earth. The latter represents long-term carbon storage, including fossil fuels and carbon capture, which, since V3, can be specified independently. A diagram of the model is provided in Fig. 5a. An interesting peculiarity of Hector is that the model performs a spin up stage before each run to ensure that its carbon cycle is in equilibrium.

The land component consists of five categories of carbon: vegetation, soil, detritus, permafrost and thawed soil. By default, the model operates with global carbon pools, but users can specify an arbitrarily large number of "biomes", each with its carbon pools and parameter values. These parameters include a warming factor to convert the global temperature anomaly into a biome-specific temperature anomaly. If multiple biomes are configured, the processes described below apply independently for each biome.

Net primary production transfers carbon from the atmosphere to the land component, distributing it across three carbon pools: vegetation, detritus and soil. This is modulated by a logarithmic $CO_2$ fertilisation factor and a land-use change (LUC) factor, which accounts for vegetation loss and gain (added in V3). A fraction of the carbon in the vegetation pool flows to the detritus pool as litter, and a fraction of both vegetation and detritus flows to the soil pool. Both detritus and soil pools lose carbon to the atmosphere through heterotrophic respiration fluxes, modelled by first-order decay equations modulated by 870 temperature and carbon pool size. Carbon residence times for detritus and soil pools are four and fifty years, respectively. As of V3, Hector allows for two independent gross LUC flows (rather than one net flow), one for carbon loss and one for carbon uptake, impacting all three carbon pools. Additionally, a permafrost module was added in V3, which is controlled by land temperature, and releases $CO_2$ and $CH_4$ into the atmosphere through an intermediate "thawed soil" pool.

The ocean component of Hector's carbon cycle is based on the work of Lenton (2000) and Knox and McElroy (1984). 875 As depicted in Fig. 5a, the ocean carbon cycle is divided into four boxes: two surface boxes for low and high latitudes, an intermediate box and a deep ocean box. The exchange of carbon between the atmosphere and the ocean surface boxes is governed by a linear function of the differential in carbon partial pressures between the two reservoirs, further influenced by temperature and salinity. This carbonate scheme resolves the following prognostic variables: carbon partial pressure ($pCO_2$), pH, concentrations of $HCO_3^-$ and $CO_3^{2-}$, and saturation states of aragonite ($\Omega_{AR}$) and calcite ($\Omega_{ca}$). Typically, the higher-880 latitude box, representing subpolar gyres, acts as a carbon sink, while the lower-latitude box outgasses carbon. Once dissolved into the surface boxes, the carbon flows to the intermediate and deep ocean (if not outgassed back to the atmosphere) through advection and mass exchange, simulating a simple thermohaline circulation.

### 4.7.2 Radiative forcing

At each time step, Hector computes the total radiative forcing relative to the year 1750, including 39 sources (see Table S1 885 in Dorheim et al. (2024b)): concentrations of carbon dioxide, methane, nitrous oxide, stratospheric water vapour, tropospheric ozone, and 27 halogenated compounds; emissions of black carbon, ammonia, organic carbon, sulphur dioxide; as well as simulating aerosol cloud interactions and taking externally defined time series for forcing related to LULCC albedo, volcanic activity and miscellaneous sources. The last contribution is zero by default, but allows the user to specify an additional forcing time series. Since V3, Hector uses the forcing equations from AR6 (IPCC, 2021) to estimate the forcing for most species (see





supplement in Dorheim et al. (2024b)), except for tropospheric $O_3$ and stratospheric $H_2O$, which still follow the formulation in Hartin et al. (2015) using radiative efficiency factors.

### 4.7.3 Temperature

Since V2, Hector has used the Diffusion Ocean Energy balance CLIMate (DOECLIM) model to estimate temperature anomaly. Originally formulated by Kriegler (2005), DOECLIM combines a zero-dimensional EBM with a one-dimensional ocean heat
diffusion scheme, as depicted in Fig. 5b. Near-surface air temperature anomaly is calculated via a linear relationship with the total radiative forcing as described in Sect. 3.3.1, using a scaling climate feedback parameter $\lambda$. The DOECLIM scheme distinguishes between forcing over land ($F_{land}$) and ocean ($F_{ocean}$), although Hector assumes both these forcings are equal to the global forcing. Heat transfer to the Earth's system is then simulated through a standard two-box scheme coupled to a 1-D diffusion ocean with uniform diffusivity across the water column. The two boxes correspond to the combination of the
upper land layer and the atmosphere over land, and the ocean mixed layer and the atmosphere over ocean. These two boxes are also allowed to exchange heat between them based on their temperature gradient. The model's transient behaviour is primarily determined by the heat exchange with the deep ocean, due to its much larger heat capacity. The model's estimation of global temperature is the area-weighted average of these land and ocean box temperatures.

## 4.8 CICERO-SCM

The CICERO-SCM is a simple climate model developed at the Centre for International Climate Research at Oslo (CICERO), Norway. It has been used in a range of studies, including estimations of historical national and regional contributions to climate change (den Elzen et al., 2005; Höhne et al., 2011; Skeie et al., 2017), an exploration of the impact of transportation and shipping sectors on global temperature (Skeie et al., 2009; Tronstad Lund et al., 2012) and the evaluation of mitigation strategies (Torvanger et al., 2012; Myhre et al., 2011).

The model was first formulated by Fuglestvedt and Berntsen (1999), where the main components of the model were presented: emission-concentration gas cycles following mass balance principles; an IRM to simulate the carbon cycle, as described in Alfsen and Berntsen (1999) and based on Joos et al. (1996); radiative forcing formulae largely following IPCC (1995, 1997); and a semi-hemispheric UD-EBM following Schlesinger et al. (1992). Since then, the model has not experienced significant changes except for an update to the radiative forcing formulae for $CO_2$, $CH_4$ and $N_2O$ based on Etminan et al. (2016), incorpo-
rating the effects of overlapping absorption bands between the species. Despite this gradual development, the model has been re-calibrated periodically as new sources of AOGCM and ESM data became available. For an up-to-date and detailed reference of the model, see Sandstad et al. (2024a), although the model differences with the original publication are small.

### 4.8.1 GHG concentrations

Atmospheric $CO_2$ concentrations in CICERO-SCM are determined using a carbon cycle module following IRM principles
(see Sect. 3.3.3). The atmosphere-ocean carbon exchange is simulated using the mixed-layer IRM described by Joos et al.



(1996), while terrestrial carbon uptake is modelled through an IRM that reduces the "effective" emissions seen by the ocean component.

The scheme by Joos et al. (1996) simulates ocean carbon uptake in two stages. First, it calculates the difference in carbon partial pressures between atmosphere and the OML to estimate the carbon uptake by the OML ($f_o(t)$), where the ocean partial pressure is approximated via a parametrisation modulated by mean global concentration of Dissolved Organic Carbon (DOC ($\delta m(t)$). Second, it calculates this global concentration of DOC in the OML using an IRM to approximate the transport to the deep ocean. The IRM is represented by the convolution of the historical carbon uptake by the OML with an IRF representing deep ocean uptake ($R(t)$), akin to Eq. 12:

$$\delta m(t) = \frac{c}{d} \int_{t_0}^{t} f_o(u)R(t-u)du \tag{35}$$

where $c$ is a coefficient for unit conversion and $d$ represents the depth of the mixed layer. The original calibration of this scheme by Joos et al. (1996) was conducted using data from two models: the HILDA model (Siegenthaler and Joos, 1992) and the Princeton 3D model (Sarmiento et al., 1992). While CICERO-SCM employs the HILDA calibration, MAGICC - another SCM utilising this scheme, reviewed in Sect. 4.9 - employs the Princeton calibration.

For the terrestrial component, CICERO-SCM also adopts an IRM approach, distinguishing itself from most other models in this review. Specifically, land carbon uptake is modelled through an "effective" NPP flux, which modifies the flux from anthropogenic emissions ($E$):

$$E_{\text{eff}}(t) = E(t) - NPP_{\text{eff}}(t) \tag{36}$$

The effective NPP term incorporates a $CO_2$-dependent term to account for carbon fertilisation effects, along with a convolution term to represent carbon returning to the atmosphere through overturning of terrestrial carbon (Joos and Bruno, 1996):

$$NPP_{\text{eff}}(t) = NPP(t) - \int_{-\infty}^{t} NPP(t')r_b(t-t')dt' \tag{37}$$

$$NPP(t) = NPP_0 \cdot \beta \cdot \ln\left(\frac{CO_2(t)}{278ppm}\right) \tag{38}$$

It is this effective emissions term, $E_{\text{eff}}$, that the ocean component uses to calculate the atmosphere-ocean carbon exchange, thus accounting for the terrestrial component of the carbon cycle. A full description of this scheme can be found in Sandstad et al. (2024a).

Concentrations for all remaining non-$CO_2$ species ($CH_4$, $N_2O$, and 27 halogenated species) in CICERO-SCM are governed by mass balance equations as described in Sect. 3.1.1. The model includes a time-varying characteristic lifetime ($\tau^{CH_4}$) for methane, considering three contributions: OH chemistry, stratospheric sink and soil sink, as per Eq. 2.




### 4.8.2 Radiative forcing

In terms of radiative forcing, CICERO-SCM uses the expressions found in Etminan et al. (2016) to estimate the radiative
forcing induced by elevated atmospheric concentrations of $CO_2$, $CH_4$ and $N_2O$, accounting for stratospheric adjustments and
the overlap between absorption bands. Additionally, efficiency factors for each species can be used to account for tropospheric
adjustments, producing an estimation for effective radiative forcings. Forcing from 27 other WMGHGs ($SF_6$, CFCs, HFCs,
HCFCs) is calculated through the usual linear efficiency approach, scaling with the concentration anomaly since pre-industrial.
Other prognostic sources of forcing included in the model are tropospheric and stratospheric ozone, stratospheric water vapour
and aerosols. The contribution to total forcing of tropospheric ozone scales based on its concentrations, which are estimated
based on methane concentrations and emissions of $NO_x$, CO, and NMVOCs. In the case of stratospheric ozone concentration,
it decreases based on the concentration of chlorine- and bromine-containing species three years prior to the evaluation step, to
account for atmospheric transport. Forcing related to stratospheric water vapour scales linearly with methane concentrations,
while aerosol forcing scales linearly with emissions of sulfates, fossil fuels, biofuels, black carbon, organic carbon and biomass
burning. Finally, albedo changes and natural forcing agents (volcanic aerosols and solar irradiance) can be added through
prescribed forcing timeseries.

As described below, CICERO-SCM possesses a hemispheric EBM, which requires a hemispheric partition of radiative
forcing. This is only relevant for three forcing agents whose forcings are not split equally: tropospheric $O_3$, albedo changes
and aerosols. The first is weighted by 1.45 for the Northern Hemisphere and 0.55 for the Southern Hemisphere, following
Skeie et al. (2020), while the other two are split following the results from Smith et al. (2020).

### 4.8.3 Temperature

To convert the total radiative forcing into a temperature response, the CICERO-SCM implements a semi-hemispheric UD-EBM
following Schlesinger et al. (1992). As illustrated in Fig. 6, this UD-EBM consists of four main parts for each hemisphere:
atmosphere, the OML (default depth 107m), deep-ocean layers (39 layers), and a sinking column of water negligible in area
compared to the ocean. The purpose of this water column is to represent thermohaline circulation, simulating polar deep water
formation which later upwells through the ocean layers. Using the standard EBM relationships described in Sect. 3.3.1), total
hemispheric radiative forcing is translated into temperature anomaly and additional heat content in the atmosphere. This heat
flows into the OML and eventually into the deep ocean through thermal diffusion across ocean layers, as well as via the
sinking water column mentioned earlier. Water in this column starts at the OML, flowing downwards to the ocean bottom,
then upwelling through the ocean layers back to the OML, restarting the cycle. The upwelling velocity in CICERO-SCM
depends on the global temperature anomaly, linearly decreasing as the temperature anomaly increases, following Raper et al.
(2001). While the land is not integrated as a heat-exchanging component, the model does incorporate the difference in ocean
extension between hemispheres. Atmospheric interhemispheric heat exchange is included in the model, but it is rarely used. A
full mathematical description of the UD-EBM in CICERO-SCM can be found in Sandstad et al. (2024a).





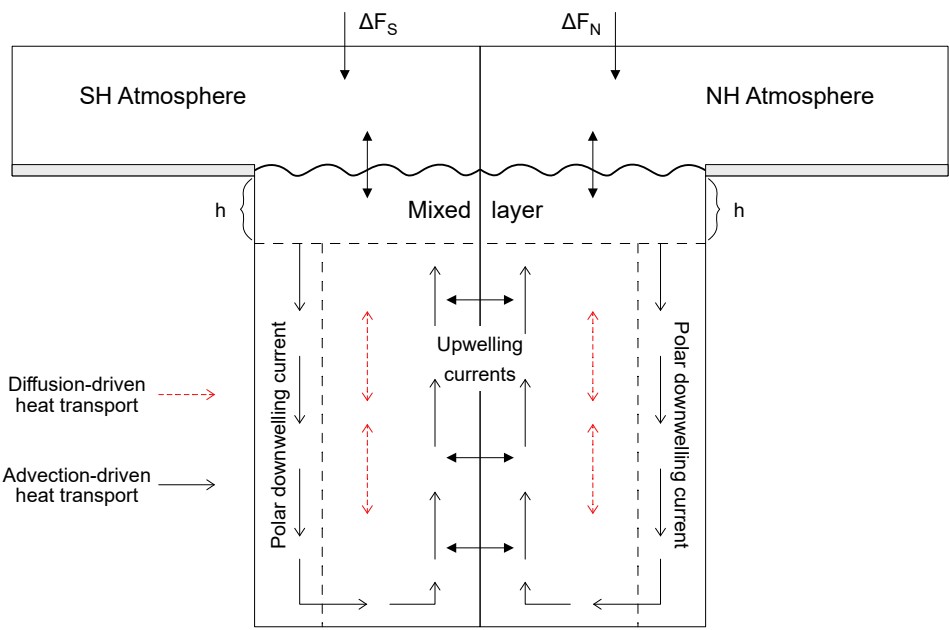

**Figure 6.** Illustration of the hemispheric upwelling-diffusion energy balance model (UP-EBM) used in CICERO-SCM. The model simulates heat transport across the ocean through two processes: diffusion (red arrows) and advection (black arrows). Figure based on Sandstad et al. (2024a, Fig. 6).

This is a similar scheme to the EBM used in MAGICC, as described in Wigley and Raper (1987), with two main differences: entrainment (or alternatively, depth-dependent area profile, as discussed in 4.9.3) was never added to the CICERO-SCM, and heat exchange with the land is disregarded. An advantage of UD-EBM schemes like these is the improved ability to estimate both ocean temperature anomaly and ocean heat content, as a representation of the different ocean layers and their temperatures exists, and a more flexible inclusion of forcing with possible hemispheric variations.

## 4.9 MAGICC

The Model for the Assessment of Greenhouse Gas Induced Climate Change (MAGICC) stands as the most prominent and long-established SCM, with nearly four decades of development. It comprises several components: a box-model representation of the terrestrial carbon cycle, an ocean carbon cycle based on a OML impulse response, a comprehensive list of forcing agents, and an Upwelling-Diffusion-Entrainment Energy Balance Model (UDE-EBM) for simulating energy balance dynamics. The model has been extensively employed in IPCC reports for generating climate projections under different forcing scenarios, as demonstrated in the Second Assessment Report (SAR, IPCC, 1995) and for scenario design in the AR6 (IPCC, 2021). Furthermore, MAGICC has been instrumental in emulating the behaviour of more complex AOGCMs and ESMs, as evidenced in the Third Assessment Report (TAR, Cubasch et al., 2001). Beyond IPCC reports, MAGICC is widely used within the IAM





community, being the climate module in IAMs such as IMAGE (Stehfest et al., 2014) at PBL Netherlands, MESSAGEix
(Huppmann et al., 2019) at the International Institute for Applied Systems Analysis in Vienna, and GCAM IAM (Calvin et al.,
2019) at the Pacific Northwest National Laboratory until version GCAM-4.3, after which the Hector SCM was adopted as the
default option.

The seminal publication for MAGICC was presented by Wigley and Raper (1987), who built upon the upwelling-diffusion
scheme for ocean heat transport proposed by Hoffert et al. (1980) to explore projections of future sea level rise. This scheme
became the core EBM around which MAGICC evolved. The transition from a simple EBM to a comprehensive SCM, incor-
porating both a carbon cycle and a representation of non-$CO_2$ species, occurred with Wigley and Raper (1992). This iteration
of MAGICC included a representation of methane, nitrous oxide, 23 halocarbons species, and sulphate aerosols in the forc-
ing calculations. The new carbon cycle module comprised a four-box terrestrial component (Wigley, 1993) and an IRM for
the ocean component (Wigley, 1991). It was also around this time that the model began to be referred to as MAGICC in the
scientific literature (Hulme et al., 1995).

With these advancements, MAGICC began to serve as an emulator for more complex AOGCMs, beginning with the Hamburg
model (Cubasch et al., 1995; Raper et al., 2001) and expanding to include multiple AOGCMs in the IPCC's TAR (IPCC, 2001).
TAR used MAGICC version 4.1 (Raper et al., 2001) while the Fourth Assessment Report (AR4, IPCC, 2007), employed version
4.2, though version 5.3 was later made compatible with AR4. The primary differences between the versions used in these two
ARs lay in the parameter values, which were recalibrated to align with the AR4 and the Coupled Climate–Carbon Cycle
Model Intercomparison Project (C4MIP, Friedlingstein et al., 2006) findings. Additionally, an updated sea level rise module
was incorporated, following Wigley and Raper (2005). For a comprehensive overview of the model changes between these
reports, refer to Appendix 2 in Wigley et al. (2009).

Meinshausen et al. (2011a) introduced MAGICC6, a version of the model extensively used throughout the 2010s, which
offers the latest comprehensive description of the model. Consequently, this publication stands as the best entry point for users
seeking to understand the modern iteration of MAGICC without delving into its historical development. For this reason, it is the
version used for the model descriptions offered below, with references to later enhancements where relevant. Enhancements in
MAGICC6 included the introduction of time-dependent climate sensitivities, greater flexibility in simulating $CO_2$ fertilisation
effects, inclusion of the OML IRM developed by Joos et al. (1996), and advanced ocean heat dynamics featuring depth-variable
area profile, entrainment, and warming-dependent thermal diffusivity. Additionally, MAGICC6 offered increased flexibility in
radiative forcing efficacies, including the ability to account for spatial patterns. It also upgraded the model's implementation
from Fortran 77 to Fortran 95.

Nearly forty years after its inception, the most recent major iteration of this SCM is MAGICC7, with a detailed discussion
of the updated model provided by Meinshausen et al. (2020). MAGICC7 introduced a permafrost module based on the work
of Schneider von Deimling et al. (2012), and enhanced the representation of GHG cycles, including improved modeling of the
Brewer-Dobson circulation (Butchart and Scaife, 2001), the evolution of hydroxyl (OH) concentrations, and an expansion in the
number of included halogenated gases to 43 species. As of 2025, MAGICCv7.5.3 is the version powering the online MAGICC
simulator available at https://live.magicc.org. From version 7.6 onwards, the model code has been made open source, with




previous versions available from the authors upon registration at https://magicc.org/download. Limited information on these

later versions is available, with a brief description of MAGICC v7.4.1 provided in the supplementary material of Nicholls et al. (2021). This material mentions the inclusion of a new state-dependent climate feedback factor, a nitrate aerosol forcing scheme, and parameterisations to simulate heat uptake by the land and cryosphere.

Among the latest enhancement we find MAGICC's latest sea level rise module, which was presented by Nauels et al. (2017). It includes an emulation of all major contributions: thermal expansion, glacier and ice sheets melting, and land water storage

change. Additionally, since Tang et al. (2025), MAGICC has a nitrogen cycle, which it uses to limit terrestrial carbon uptake.

### 4.9.1 GHG concentrations

MAGICC offers a comprehensive treatment of the various agents currently believed to influence the climate. MAGICC6 calculates concentrations for 30 species (expanded to 43 in MAGICC7) by simulating the gas cycles for $CO_2$, $CH_4$, $N_2O$, and other 28 halogenated gases (40 in MAGICC7). Generally, the standard approach of combining emissions with sinks and using

decaying exponentials with characteristic lifetimes is employed in this model to simulate gas cycles (see Sect. 3.1.1). However, MAGICC's treatment of gas cycles has accrued modifications over the years, making it relatively complex. Generally, these modifications involve additional parameterisations enabling the modification of species lifetimes over time, to account for phenomena such as interaction with OH radicals, increased Brewer-Dobson circulation (Butchart and Scaife, 2001), and interactions between species. These representations were further refined in MAGICC7. Another peculiarity of this model is

that it resolves hemispheric emissions and concentrations for non-well-mixed GHGs, a consequence of the hemispheric EBM it possesses. Due to the complexity and number of enhancements a full description is outside the scope of this review and the reader is directed to Meinshausen et al. (2011a, 2020) for full details.

The carbon cycle module consists of an ocean component following the implementation of Joos et al. (1996) calibrated to the Princeton 3D model results (see Sect. 4.8.1) and a three-box terrestrial component as outlined in Wigley (1993) and described

in Meinshausen et al. (2011a). These boxes represent global vegetation, litter and soil carbon reservoirs, each exchanging carbon with the atmosphere. A diagram illustrating the various carbon pools and associated fluxes is provided in Fig. 7. These fluxes are:

- NPP flux: This flux transfers carbon from the atmosphere to the vegetation (35%), litter (60%), and soil (5%) pools. This distribution of NPP across different pools, along with the similar distribution of the litter flux below, is atypical in SCM

carbon cycles which usually implement this flux taking carbon exclusively to the vegetation pool. The objective is to account for the long time steps that SCM are usually run with (usually one year). By including effects not only in the target pool (vegetation), but also in subsequent pools in the carbon cycle, MAGICC aims to create a carbon cycle model that is more sensitive to NPP and litter flux changes.

- Litter flux: This flux moves carbon from the vegetation pool to the litter (98%) and soil (2%) pools.

- Decomposition: This process transfers carbon from the litter pool to the soil pool.





- – Deforestation: This flux accounts for land-use change, transferring carbon from all three land pools (vegetation, litter, soil) back to the atmosphere.

- – Respiration: This flux represents carbon losses due to plant respiration and the decomposition of organic matter in litter and soil, returning carbon from the vegetation, litter, and soil pools to the atmosphere.

The litter, deforestation and respiration flows in MAGICC are proportional to the carbon content in their respective source pools and are governed by associated turnover times. These turnover times are constant, which implies that following a land-use change event, the carbon in the various pools will asymptotically return to their original levels, assuming no further changes occur. This behaviour effectively simulates a regrowth process after deforestation. This simplification clearly does not capture real-world dynamics, where land-use changes usually result in persistent alterations to carbon stocks. To address this, an

additional parameter was included in the model to account for the fraction of deforested land that does not recover. The scheme was further modified by Tang et al. (2025), where a direct impact on NPP by the extent of the deforestation was implemented, thereby impacting long-term equilibrium in carbon stocks.

The return to original carbon content in the carbon pools is contingent on stable climatic conditions, as the model incorporates carbon- and climate-carbon feedbacks. Specifically, the model resolves two key processes: $CO_2$ fertilisation of NPP

and temperature-induced changes in NPP, respiration and decomposition fluxes. $CO_2$ fertilisation can be simulated using the usual logarithmic formulation, a hyperbolic formulation, a sigmodial formulation since Tang et al. (2025), or a linear combination of the three. Temperature effects are modelled as an exponential modulation in the aforementioned flows in response to the temperature anomaly, with the added possibility of a sigmodial modulation since Tang et al. (2025). For a more detailed explanation, refer to the Appendix A1.1 in Meinshausen et al. (2011c) and Tang et al. (2025).

The terrestrial carbon cycle described here was further enhanced by Tang et al. (2025) to include the limiting effects of nitrogen that have been observed in more complex models (Arora et al., 2020). MAGICC is, therefore, the first and only SCM in this review to include a nitrogen cycle and emulate its impact on the carbon cycle. This is a relatively complex scheme, and only a brief summary is offered here. It mirrors the structure of the MAGICC's carbon cycle, comprising four global nitrogen pools: vegetation, litter, soil, and mineral. The flow of nitrogen through the different pools is depicted in Fig. 7 and behaves as

follows: similarly to the NPP flux in the carbon cycle, two fluxes transport nitrogen from the inorganic pools into the organic pools (vegetation, litter and soil), plant uptake from the mineral pool and biological nitrogen fixation from the atmosphere. Then, vegetation loses nitrogen to the litter and soil pools through a litter production flux. Litter loses nitrogen to the soil and mineral pools through litter decomposition. Soil loses carbon to the mineral pool through soil respiration. Additionally, the organic pools can lose carbon to the atmosphere through an anthropogenic land use emission flux. Finally, the mineral pool

can gain nitrogen through fertiliser application and through atmospheric deposition (from the atmosphere pool), and lose it through a mineral loss flux. Similarly to the carbon cycle, most of these fluxes are governed by first order decay functions with characteristic turnover times.

This nitrogen cycle is coupled to the carbon cycle mainly through the NPP flux, which is modulated by the possible plant uptake of nitrogen. This uptake is computed as a function of the size of NPP (to account for declining carbon:nitrogen ratios),





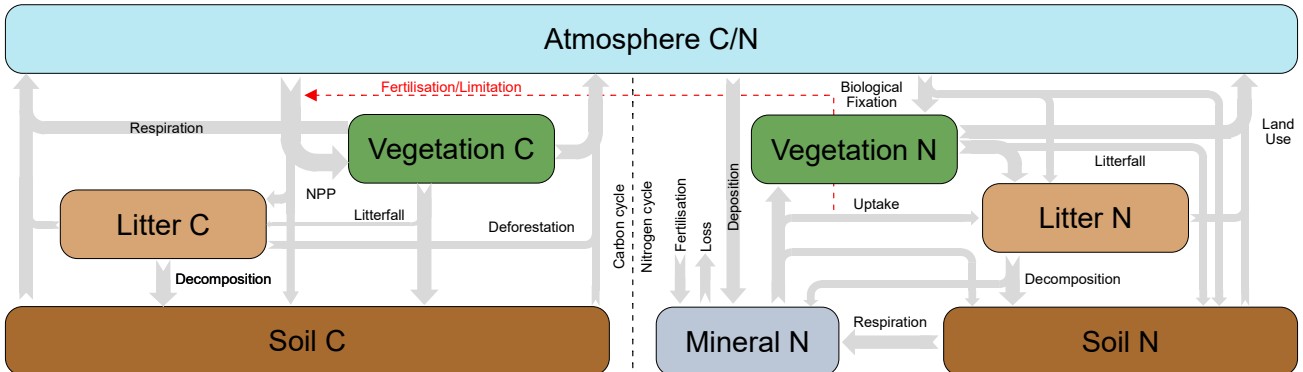

**Figure 7.** Diagram of the terrestrial carbon and nitrogen cycles in the MAGICC SCM. The limitation of carbon uptake by vegetation is mainly a function of the nitrogen uptake flux. Image based on Tang et al. (2025, Fig. 1).

nitrogen availability and temperature. The nitrogen availability, in turn, is approximated using the fluxes in the nitrogen cycle described above.

Since MAGICC7, (Meinshausen et al., 2020), the carbon cycle also possesses a permafrost module based on the work of Schneider von Deimling et al. (2012). This module divides the permafrost stocks into a number of zonal (latitudinal) bands - 50 by default - each with different carbon content and thawing thresholds. When the local temperature excedes the thawing threshold, $CO_2$ (and potentially $CH_4$) is released into the atmosphere. The quantity of gases released depends on the type of soil (mineral or peatland) and the temperature. This is a relatively complex module, and readers are referred to Schneider von Deimling et al. (2012) for further details.

### 4.9.2 Radiative forcing

For WMGHGs, MAGICC uses standard methods from the literature. MAGICC6 employed the usual logarithmic relation to calculate $CO_2$ forcing (Myhre et al., 1998), and followed IPCC (2001) for the combined $CH_4$ and $N_2O$ forcing. These forcings were further refined in MAGICC7, adopting the scheme from Etminan et al. (2016) to compute forcing for these three species. For halogenated gases a radiative efficiency approach is taken, multiplying this factor by concentrations. MAGICC accounts for the direct and indirect contributions of tropospheric aerosols on radiative forcing directly from their emissions, given their short atmospheric lifetimes. The direct contribution is estimated as a linear relationship between concentrations and forcing, while the indirect effects are modelled by using optical thickness timeseries for the relevant species: sulfates, nitrates, black carbon and organic carbon. In addition, the model includes the contributions of both tropospheric and stratospheric ozone, which are simulated through simple relationships based on ozone concentrations. Similarly, forcing from stratospheric water vapour due to methane-induced enhancement is estimated linearly (default 15%) with the pure methane forcing (without absorption band overlaps). Natural forcings (volcanic aerosols and solar irradiance), as well as LULCC albedo effects on forcing, can be added as prescribed time series. Full details can be found in Meinshausen et al. (2011a) with the MAGICC7 modifications described





in Meinshausen et al. (2020). Finally, MAGICC also includes a contrail scheme that estimates forcing effects from aviation emissions supplied by the user. However, this is seldom used and mentioned in the literature.

Each individual forcing contribution is assigned an efficacy value to account for indirect effects, leading to an estimation of ERF, which is subsequently used to estimate temperature anomalies using the model's EBM. These efficacies are allowed to vary over time and space, with potential different values for each of the hemispheric ocean and land boxes. The hemispheric partition is a consequence of the EBM employed by the model, although only three species have hemispheric differences in their forcing contributions: tropospheric ozone, halogenated gases, and aerosols. Hemispheric differences of tropospheric ozone and aerosols are a consequence of the different hemispheric emissions and concentration for these species, while the difference for halogenated gases is dependent on the species lifetime, following Hansen et al. (2005).

### 4.9.3  Temperature

MAGICC's EBM traces back to the origins of the model, with a hemispheric UD-EBM initially formulated by Wigley and Raper (1987), based on the work of Hoffert et al. (1980). This hemispheric separation can be useful for spatially inhomogeneous forcings associated with human activities such as aerosols and tropospheric ozone. A similar scheme would be adopted by the CICERO-SCM years later, as described in Sect. 4.8.3 where a summary on the basic principles on how it works can be found. There are some differences however. Unlike CICERO-SCM, MAGICC considers heat exchange between land and ocean. In fact, since MAGICC6, varying heat-exchange coefficients between land and ocean have been employed as a mechanism to alter the temporal profile of the model's effective climate sensitivity. Additionally, the model allows for time-dependent feedback parameters to modify its climate sensitivity over time.

Originally, most of the parameters associated with this scheme were fixed, but they were gradually relaxed overtime. For instance, upwelling and downwelling rates were allowed to evolve in time (Raper et al., 2001). Starting with MAGICC6, a warming-dependent gradient of thermal diffusivity was introduced to account for the increased stratification of the ocean as temperatures rise. This version also saw the inclusion of the "entrainment" component of the module. This entrainment mechanism became necessary with the introduction of a depth-dependent area profile in the ocean column, as illustrated in Fig 8b. To satisfy conservation of mass with a vertically constant upwelling rate, water had to be added, leading to the incorporation of water entrainment from the sinking column into each of the ocean layers (the default being 50 layers).

### 4.10  SCM4OPT

The Simple Climate Model for Optimization (SCM4OPT) is an IAM developed by Dr. Xuanming Su at Japan's National Institute for Environmental Studies and Agency for Marine-Earth Science and Technology. Like other IAMs, the model includes interactions between society and the environment, but this review focuses exclusively on its climate component. Further details on its socio-economic components can be found in the referenced literature. SCM4OPT has been applied in various contexts, including the exploration of costs associated with climate mitigation and adaptation strategies (Su et al., 2017, 2018), the assessment of anthropogenic and natural contributions to global warming (Su et al., 2022, 2024), and an assessment of the likelihood of triggering certain climate tipping points (Iseri et al., 2018).





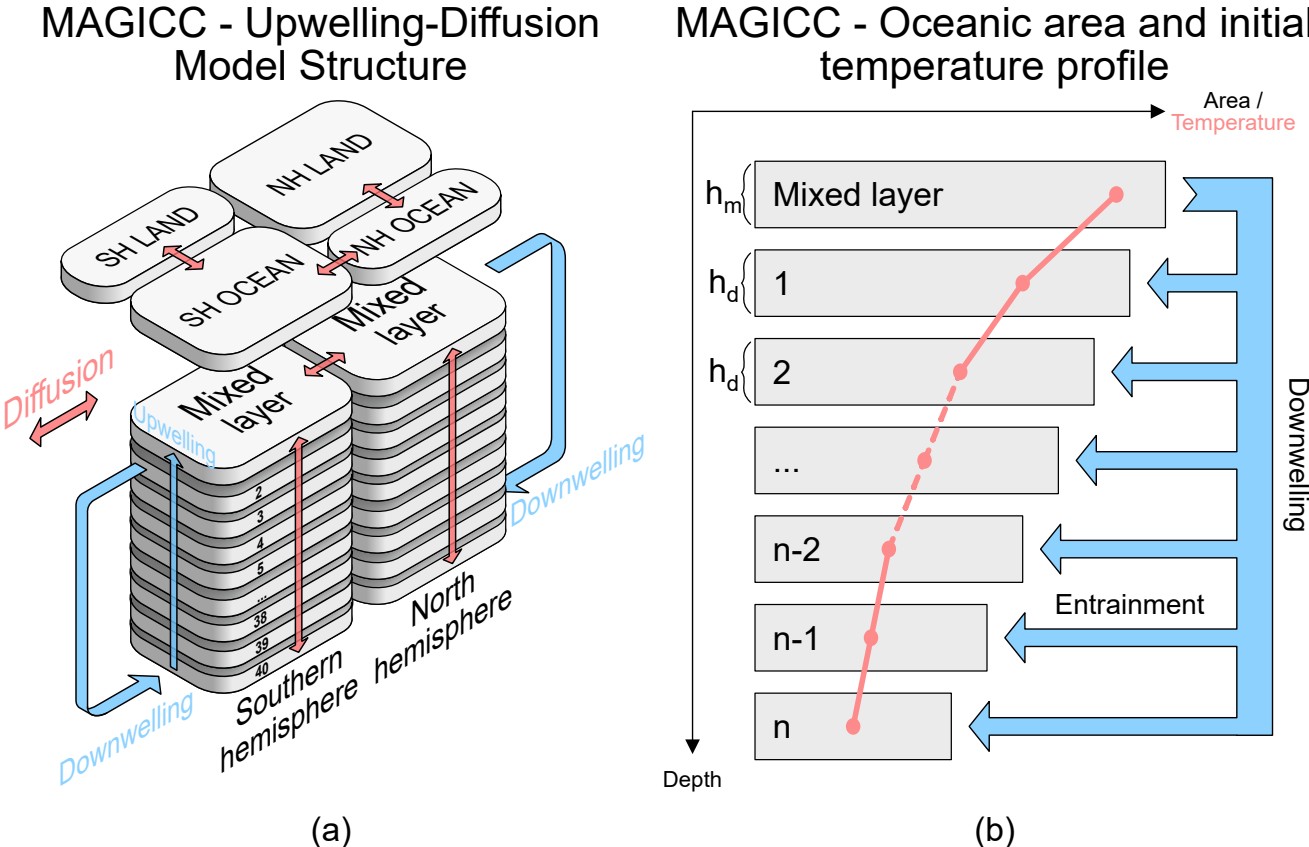

**Figure 8.** Diagrams for the hemispheric upwelling-diffusion-entrainment ocean heat transport module in MAGICC. Forcing is applied to the surface ocean and land, modifying temperature and available heat. This heat is exported to the deep ocean through two mechanisms: diffusion and advection. The model also includes a depth-dependent layer size, which requires water entrainment to maintain constant water upwelling velocity. Diagrams based on Meinshausen et al. (2011a, Figs. A1 and A2).

SCM4OPT is a relatively recent model, first introduced by Su et al. (2017) as a modification of the DICE-2013R IAM (Nordhaus, 2014), incorporating process representations borrowed from MAGICC6. The same SCM4OPT model was also used by Fujimori et al. (2016), seemingly a year earlier, due a publication order issue. Version 2 (for which no comprehensive source is available, but it was used by Nicholls et al. (2021)), introduced modifications to the ocean carbon cycle, updating it to follow Hector v1.1 (Hartin et al., 2016), and adopted a 0-D climate and 1-D ocean heat diffusion EBM, DOECLIM (Kriegler, 2005), as implemented in Hector V2.0. This version also involved a reparametrisation and recalibration of the model based on AR5 data (IPCC, 2013) and OSCAR V2.2 parameterisations (Gasser et al., 2017). Version 3 (Su, 2021; Su et al., 2022), brought further updates, particularly to the carbon cycle, which was recalibrated using CMIP6 model outputs. The latest





published iteration, Version 3.3 (Su et al., 2024), introduced a new parameterisation of $CH_4$ forcing based on the work of Etminan et al. (2016), along with the use of an ENSO-associated index to account for natural variability originating from the ocean in historical simulations.

### 4.10.1 GHG concentrations

The carbon cycle in SCM4OPT integrates various components that have been discussed elsewhere in this review. The land component, for instance, employs the same three-box model used in MAGICC6 (Meinshausen et al., 2011c), which includes carbon pools for the vegetation, detritus and soil, as described in Sect. 4.9.1. The key distinction in SCM4OPT compared to MAGICC lies in its representation of forest regrowth, which is modelled as a variable with a linear dependence on the relaxation times of each carbon pool, rather than through a parameterisation of permanent deforestation.

Similarly, the ocean carbon cycle in SCM4OPT initially followed the approach of Meinshausen et al. (2011c), implementing the OML IRM of Joos et al. (1996). However, in SCM4OPT version 2, this scheme was replaced with a four-box ocean carbon cycle model, as implemented in Hector v1.1 (Hartin et al., 2016), and described in Sect. 4.7.1, where more details can be found. A full description of both the land and ocean carbon cycles in SCM4OPT is provided in the supplementary materials of Su et al. (2017, 2022).

Over time, the number of species whose concentrations are determined from emissions in this model has expanded. The latest Version 3.3 (Su et al., 2024) included a representation of $CO_2$, $CH_4$, $N_2O$, 39 halogenated gases, tropospheric and stratospheric ozone, and aerosols (comprising $SO_4$, black carbon, nitrates, and primary and secondary organic aerosols). SCM4OPT adopts the conventional approach of defining mass balance models with emissions, sinks, and lifetimes (see Sect. 3.1.1) for most species - $CH_4$, $N_2O$ and halogenated gases - with some ad-hoc expressions for aerosols and ozone. The aerosol concentration scheme is based on a different SCM, OSCAR (Gasser et al., 2017).

### 4.10.2 Radiative forcing

Similarly to the representation of WMGHGs, the number of radiative forcing agents included in SCM4OPT has increased over time. Version 3.3 (Su et al., 2024) included radiative forcing effects from $CO_2$, $CH_4$, $N_2O$, 39 halogenated gases, aerosols (direct) including mineral dust, clouds, stratospheric and tropospheric ozone, stratospheric water vapour, land albedo, black carbon on snow, as well as natural forcings (solar and volcanic). The forcing emulation generally follows expressions from the literature. Contributions from $CO_2$, $CH_4$ and $N_2O$ are calculated following the IPCC (2001) formulation, although Etminan et al. (2016) was used in V3.3 alongside this formulation to estimate the $CH_4$ contribution in a Monte Carlo simulation; direct effects of aerosols, clouds, tropospheric ozone and land albedo effects calculation follows OSCAR's (Gasser et al., 2017); the contribution from stratospheric ozone is based on the equivalent effective stratospheric chlorine concentration, following Newman et al. (2007); impacts from halogenated gases are estimated through a radiative efficiency approach to species concentrations, after MAGICC6; and the contribution from black carbon on snow is linearly scaled with black carbon emissions. Natural forcings can be included through prescribed forcing time series. The latest comprehensive reference for the





forcing calculation in SCM4OPT is provided in the supplementary materials of Su et al. (2022), with a subsequent modification

to account for the effects of overlapping absorption bands of $CO_2$, $CH_4$ and $N_2O$ (Etminan et al., 2016).

### 4.10.3 Temperature

The conversion of total radiative forcing into a temperature response was initially performed using a two-box EBM as in the original DICE model. However, in SCM4OPT v2.0, this component was replaced by the Diffusion Ocean Energy balance CLIMate (DOECLIM) model (Kriegler, 2005). As discussed previously in Sect. 4.7 on Hector, where more details can be

found, DOECLIM couples a 0-D energy balance model with a 1-D ocean heat diffusion scheme. In V3.3, an observationally-constrained statistical model was introduced to correct for temperature biases stemming from ocean variability using an ENSO index. This is similar to EM-GC's use of ocean indices to account for natural variability; however SCM4OPT is restricted to ENSO alone. As a result, it faces the same limitation: it can only be applied in historical simulations, where the ENSO index is available, since there is currently no reliable method for projecting the index into the future.

## 4.11 ACC2

The Aggregated Carbon Cycle, Atmospheric Chemistry, and Climate model (ACC2) comprises three primary climate modules: a box-based global carbon cycle module, an atmospheric chemistry module calculating concentrations for other GHGs, and a climate module determining the total radiative forcing from different sources and translating it into temperature anomaly via the DOECLIM scheme (Kriegler, 2005). A diagram of these three modules is provided in Fig. 9. A distinctive feature of

ACC2 is its ability to run in inverse mode, allowing it to produce best estimates for its model parameters based on historical climate data. This capability has enabled the model to produce estimations for the global warming potentials of $CH_4$ and $N_2O$ (Tanaka et al., 2009a) and for climate sensitivity (Tanaka et al., 2009b; Tanaka and Raddatz, 2011), as well as to assess the behaviour of different emission metrics under various stabilisation scenarios (Tanaka et al., 2013; Tanaka and O'Neill, 2018; Tanaka et al., 2021; FAO, 2023; Mastropierro et al., 2025). Additionally, ACC2 has been used to estimate transfer payments

to lower-income countries under a global carbon price scenario (Landis and Bernauer, 2012), quantify the climate impact of permafrost thaw (Yokohata et al., 2020), explore carbon cycle feedbacks (Melnikova et al., 2023), and investigate the roles of atmospheric methane removals and enhanced weathering on mitigation pathways (Gaucher et al., 2025a, b)

ACC2 was born as a significant expansion of the Integrated Assessment of Climate Protection Strategies (ICLIPS) Climate Model (ICM, Bruckner et al., 2003), which in turn evolved from the nonlinear impulse-response model of the coupled carbon

cycle-climate system (NICCS, Hooss et al., 2001) and the structural integrated assessment model (SIAM, Hasselmann et al., 1997). The original ACC2 publications (Tanaka et al., 2007; Tanaka, 2008) provide a comprehensive guide to the model, and remain a valuable resource for understanding its details, as no major changes have occurred to the Earth system modules since then. Tanaka et al. (2013) introduced a fourth module that estimates GHG emission costs using Marginal Abatement Cost (MAC) functions based on Johansson (2011). This addition transformed the coupled system into an IAM, although the climate

component has continued to be used independently. The latest version of ACC2, V4.3 (Tanaka and O'Neill, 2018), further modified the $CO_2$ MAC function to incorporate the effects of negative emissions. Subsequently, the climate component has





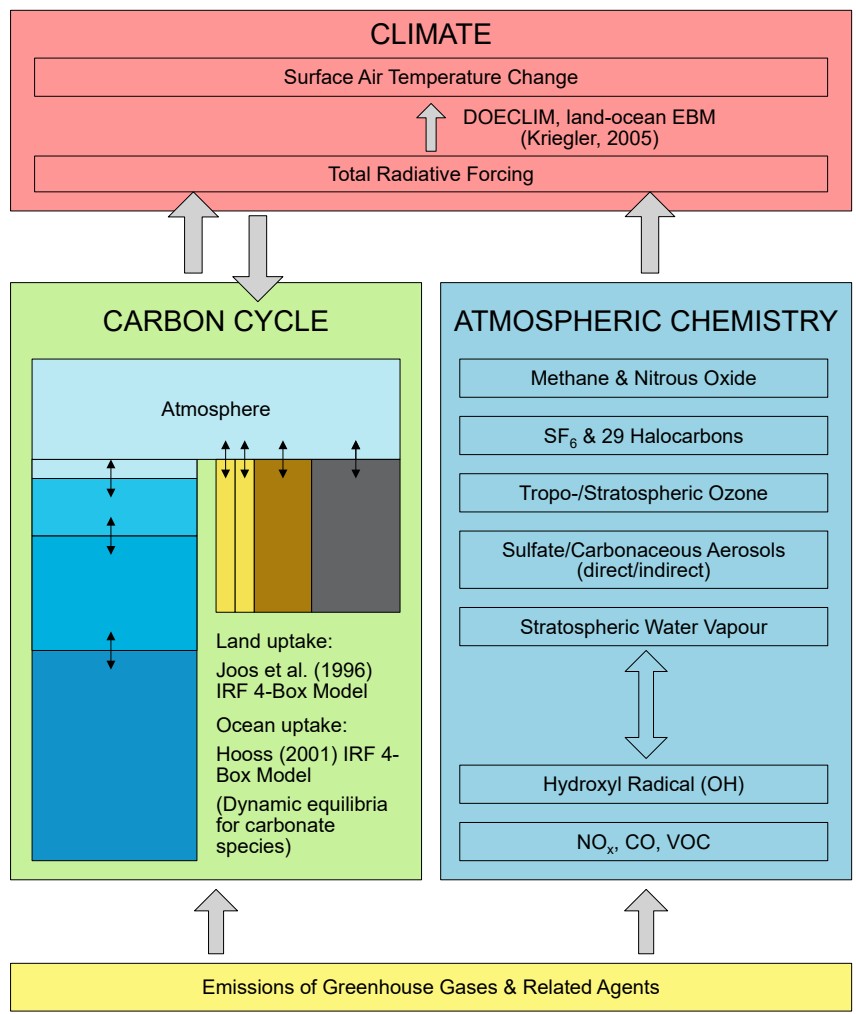

**Figure 9.** Diagram of the three primary modules constituting the climate component of the ACC2 model. Image based on Tanaka (2008, Fig. 1).

been coupled to the GET model (Azar et al., 2013; Johansson et al., 2020) to create an IAM that includes two-way interactions between climate and economy (Gaucher et al., 2025b), and to a more sophisticated mitigation module based on MAC curves emulating ten different IAMs (Xiong et al., 2025). These additions are not assessed, as only the climate components are within
scope of this review.





### 4.11.1 GHG concentrations

The carbon cycle in ACC2 consists of four-box representations for both the terrestrial and oceanic components, which are coupled together through the OML-atmosphere box. Details can be found in Tanaka et al. (2007) and Tanaka (2008), both offering the same model description.

For the ocean component, the model simulates progressive carbon uptake using a combined OML-atmosphere layer and three deep ocean layers. The combination of atmosphere and mixed-ocean layer is justified by the short time required for these carbon reservoirs to reach equilibrium, which is significantly shorter than the one-year timestep used in the model. Carbon emissions from fossil fuels and land-use change are added to this layer. A relatively complex carbonate chemistry scheme then determines the amount of Dissolved Inorganic Carbon (DIC) in the OML, as well as its pH. The model also accounts for

the effects of rising OML temperatures on carbonate uptake, with a recommended model emulation range of up to four times the pre-industrial level of atmospheric concentrations. Carbon is subsequently exported to the deeper ocean layers through diffusion. The model parameters were calibrated using the IRF for ocean carbon uptake from Hooss (2001) (R01 experiment), which emulates the response of the HAmburg Model of the Ocean Carbon Cycle version 3i (HAMOCC 3i) to a small increase in atmospheric carbon.

Similarly, the terrestrial carbon cycle is represented by four boxes that roughly correspond to vegetation, detritus, wood and soil organic carbon. These boxes interact exclusively with the OML-atmosphere layer rather than with each other. This atypical structure is a consequence of the IRM used in the model for heterotrophic respiration, which is a diagonalisation of the Bern-CC model (Joos et al., 1996). The diagonalisation decouples the carbon reservoirs from each other, allowing a single interaction with the OML-atmosphere layer. The downside of this diagonalisation is that the resulting boxes are no longer a

representation of physical biospheric reservoirs. This is similar to the relationship between n-time-constant temperature IRMs and n-layer temperature models discussed in Sect. 3.3.3. Each box has two processes: (i) a carbon gain term associated with NPP, modulated by the standard logarithmic expression to account for $CO_2$ fertilisation effects; and (ii) a carbon loss term representing the turnover of carbon within the box, characterised by a decay time that simulates the effects of respiration. The carbon loss term also incorporates temperature effects through an exponential factor based on the land surface temperature

anomaly.

For all other GHGs, ACC2 employs the standard mass balance treatment to simulate their life cycle (see Sect. 3.1.1). This includes $CH_4$ (with a characteristic lifetime accounting for different sink times through Eq. 2), $N_2O$, and 30 halogenated species. ACC2 also includes a representation of OH and tropospheric ozone concentrations, which follow the OxComp Workshop results (Joos et al., 2001; IPCC, 2001), and affect GHG concentrations. Table 2.1 in Tanaka et al. (2007) presents a useful

summary of GHG concentration calculations in this model.

### 4.11.2 Radiative forcing

ACC2 includes a comprehensive list of radiative forcing agents: $CO_2$, $CH_4$, $N_2O$, 30 species of halogenated gases (29 halo-carbons and $SF_6$), tropospheric and stratospheric ozone, stratospheric water vapour and aerosols (sulfate and carbonaceous,



plus indirect effects). The model generally adheres to standard treatments for these species, primarily based on IPCC (2001)
and WMO (2003). For instance, it follows IPCC (2001) to calculate the contribution from $CH_4$ and $N_2O$, accounting for over-
lapping effects, as well as using its logarithmic expression for the $CO_2$ forcing. Contributions from halogenated species are
scaled based on their radiative efficiencies (IPCC, 2005; WMO, 2003). Aerosol forcing is parametrised as a function of $SO_2$,
organic carbon and black carbon, based on the work of Joos et al. (2001) and IPCC (1997). Natural forcings, volcanic aerosols
and solar irradiance, are included as prescribed forcing time series, using estimations from Ammann et al. (2003) and Krivova
et al. (2007), respectively. Since the model uses DOECLIM as its EBM, the radiative forcing must be separated between its
land and ocean components. All forcing contributions are roughly split in half, except the carbonaceous aerosol and the tropo-
spheric ozone contributions, which are divided following Harvey (2000). A full description of non-$CO_2$ and forcing estimation
is provided in section 2.2 of Tanaka et al. (2007), with a useful summary in Table 2.1 of the same publication.

### 4.11.3   Temperature

ACC2 supports two different EBMs to compute the temperature response to a given total radiative forcing. One option is a box-
based formulation of the forcing describing the temperature response to forcing from Hooss (2001), which is recommended
only for inverse calculations aiming to estimate the climate sensitivity. The default and recommended EBM for general model
use, particularly when a computationally lighter model is necessary, is the DOECLIM scheme presented in Kriegler (2005)
and discussed in Sect. 4.7. This consists of a zero-dimensional EBM with a one-dimensional diffusion scheme to represent the
heat exchange with the deep ocean. For more details about this scheme see, in increasingly level of detail, Sect. 4.7.3 in this
text, section 2.3 in Tanaka et al. (2007), and the original publication of Kriegler (2005).

### 4.12   OSCAR

The "Occupation des Sols et cycle global du CARbone" (OSCAR) model is an SCM that primarily focuses on the carbon
cycle and the disturbances caused by LULCC. Over time, the model has evolved well past the capabilities and role suggested
by its name. Indeed, OSCAR's book-keeping approach to carbon tracking, which is applied across various model regions and
biomes, arguably places the SCM as the most complex and flexible model for carbon cycle simulation among those reviewed
in this text. This intricacy aligns with one the model's core design principles (Gasser et al., 2017): "adding as many modules
and processes to the module as possible, favouring number of processes over process complexity". Due to this complexity, only
an overview of the model is provided in this review. For detailed information, readers are referred to Gasser et al. (2017) for
developments up to V2.2, which is the most up-to-date comprehensive description of the entire model, and Gasser et al. (2020)
for the latest details on its land carbon cycle.

OSCAR has been utilised in multiple high-impact studies, including the attribution of emissions to emitting and absorbing
regions (Ciais et al., 2013), an analysis of China's contribution to global radiative forcing (Li et al., 2016), an investigation of
the implications of permafrost thawing to the global carbon budget (Gasser et al., 2018), and an evaluation of the effects of
climate change on future bioenergy production (Xu et al., 2022). Its complexity and flexibility have also made it a key tool in
the production of the annual Global Carbon Budget (GCB) reports (Friedlingstein et al., 2024).





This model was first introduced by Gitz and Ciais (2003) as a carbon cycle model designed to simulate carbon stocks and flows across regional biomes. Shortly after, Gitz (2004) enhanced the model to include a simple climate response and climate-carbon feedbacks, such as temperature impacts on NPP and soil respiration. A decade later, OSCAR V2 was presented by Gasser and Ciais (2013), with the main difference being the transition from Scilab to Python 2 as the programming language. V2.1 followed shortly, incorporating representations for non-CO2 species and multiple climate responses calibrated against CMIP5 models. This version is comprehensively described by Gasser (2014), though in French, with partial English descriptions provided by Cherubini et al. (2014) and Li et al. (2016).

The most significant increase in OSCAR's complexity occurred with V2.2 (Gasser et al., 2017), which introduced numerous additions and modifications to OSCAR modules. These included enhancements to both the ocean and land carbon cycles, the development of ozone (both stratospheric and tropospheric) and aerosol modules, and the creation of albedo and wildfire modules. V2.3 (Gasser et al., 2018) introduced a permafrost module. V3 completely rewrote the model in Python 3, improving its code quality and solver scheme while leaving all physical equations and parameters unchanged. Since then, only minor changes have been implemented. V3.1 included some small modifications to the carbon cycle representation, as described in Gasser et al. (2020), where a description of the changes between versions 2.2 and 3.1 can be found (Appendix 3). A brief description of subsequent changes is offered in the model's CHANGELOG. V3.2 implemented a new formulation of $CO_2$ partial pressure and modified the parameters controlling deep ocean carbon transport following Strassmann and Joos (2018). V3.3 updated long-lived GHG forcing expressions, as well as temperature response parameters, to follow the IPCC AR6 Chapter 7 (Forster et al., 2021b).

An unusual feature of OSCAR is its emulation of precipitation change. This process is not typically included in SCMs, GREB being the only other SCM in this review to include it. The global anomaly in precipitation is modelled as a linear dependence on surface temperature and forcing. For each model region, OSCAR then translates this global anomaly into local values using a pattern-scaling approach (Santer et al., 1990; Mitchell, 2003; Mathison et al., 2025) with linear weights calibrated to the CMIP5 relationships between global and regional values. A similar approach is also used to derive local temperature anomalies from the global amount determined from the EBM.

### 4.12.1 GHG concentrations

Generally speaking, this model operates with anomalies with respect to the pre-industrial state, rather than absolute quantities. This is the case for OSCAR's carbon cycle, which was broadly established in its current form by V2.2 (Gasser et al., 2017), with V3.1 (Gasser et al., 2020) adding a few minor fluxes and recalibrating its pre-industrial steady state to output from the GCB 2018 (Le Quéré et al., 2018). Its primary strength resides in its land component, which includes a complex representation of LULCC disturbances across model boxes that represent different regions and "biomes" within regions. The ocean component is based on the work of Joos et al. (1996) with several modifications.

In OSCAR, land is divided into multiple boxes representing a number of regions, each containing several biomes. These boxes provide an average characterisation for each biome within each region. Users can choose the number of biomes and, since V3, of regions; however, typical configurations include around ten regions and five biomes. For instance, Gasser et al.





(2020) used ten regions following Houghton and Nassikas (2017), and five biomes: forests, other natural lands (grasslands, shrublands, bare soil), croplands, pastures and urban lands. This can vary, however. For instance, the latest GCBs (Friedlingstein et al., 2022, 2023, 2024) include 210 regions. Each biome in the model contains three carbon pools: vegetation, litter and soil. The flow of carbon through these boxes is governed by the following fluxes:

– NPP: carbon flux from atmosphere to vegetation, modulated by temperature (linear relationship), precipitation (linear) and $CO_2$ concentration (logarithmic or hyperbolic).

– Litterfall: carbon flux from vegetation to both litter and soil (introduced in V3.1) pools, with a magnitude scaling linearly with vegetation carbon stock.

– Litter respiration: carbon flux from the litter pool to the atmosphere, dependent on temperature (exponential or Gaussian
relationship) and precipitation (linear).

– Decomposition: carbon flux from litter to soil, proportional to litter respiration.

– Soil respiration: carbon flux from the soil pool to the atmosphere, with similar dependence on local properties as the litter respiration flux.

– Fire: carbon flux from the vegetation pool to the atmosphere, proportional to the vegetation carbon stock and influenced
by $CO_2$ levels, local temperature and precipitation, following a linear relationship.

– Harvest (introduced in V3.1): carbon flux from the vegetation pool to the atmosphere, representing emissions from harvested crop products. It only applies to the crop biome.

– Grazing (introduced in V3.1): carbon flux from the vegetation pool to the atmosphere, representing emissions from pasture grazing. It only applies to the pasture biome.

Through an elaborate book-keeping approach, OSCAR tracks carbon across each region-biome pair, and emulates the movement of carbon associated with LULCC disturbances. These disturbances are currently limited to anthropogenic activities (no dynamic vegetation). For each biome and region, three wood product pools are defined that receive carbon after a LULCC disturbance. These pools are characterised by distinct turnover times that determine the rate at which carbon is subsequently released into the atmosphere. Broadly speaking, these pools represent fuel wood ($\sim$ 1 year), pulp-based products ($\sim$ a few years)
and hardwood-based products ($\sim$ dozens of years). The allocation of carbon to each wood pool, as well as the disturbance-induced movement of carbon between vegetation, litter and soil carbon pools across different biomes, depend on the specific nature of the LULCC disturbance. The model includes three types of disturbances: land-cover change, wood harvest and shifting cultivation. Given the large number of potential combinations (five biomes, three carbon pools and three wood pools) the detailed mechanics of these transitions can be complex, and interested readers are referred to the appendix of Gasser et al.
(2020) for further information.





OSCAR's ocean component of the carbon cycle builds on the mixed-layer pulse response function proposed by Joos et al. (1996), discussed in Sect. 4.8.1. However, several modifications have been implemented to improve upon the original Joos model:

- Following Harman et al. (2011), OSCAR replaced the convolution of the atmosphere-ocean flux in Joos et al. (1996) (Eq. 35), with an equivalent box model. These boxes represent different turnover times of carbon mixing between the mixed layer and the deep ocean, rather than different ocean basins.

- The empirical carbonate chemistry equation used to calculate ocean carbon partial pressure (Eq. 6b in Joos and Bruno (1996)) was augmented with a dependency on sea surface temperature, extending applicability of the original formulation.

- The depth of the mixed layer ($d$ in Eq. 35) was updated with a dependence on sea surface temperature to simulate ocean stratification.

Since V2.3 (Gasser et al., 2018), OSCAR also includes a representation of permafrost carbon stocks and potential thawing. This addition introduced a frozen carbon pool for two regions (Eurasia and North America). Thawing is driven by local temperature, following an empirical S-shaped function. Similarly to the LULCC module, thawed carbon is not immediately emitted to the atmosphere, but is instead distributed among thawed carbon pools (default number is three), each with its characteristic respiration turnover time. Respiration from these thawed pools is influenced by local temperatures, following the same relationships as standard soil respiration functions, but with different parameter values.

Beyond $CO_2$, OSCAR includes a comprehensive list of GHGs: methane, nitrous oxide and 37 halogenated compounds. It uses the usual one-box approach with emissions and sinks (see Sect. 3.1.1), along with Eq. 2 to combine different sink contributions. Additionally, OSCAR implements time-dependent lifetimes with relatively complex dependencies on temperature and other atmospheric species. Due to this complexity, only a brief summary is offered here, with full details available in Gasser et al. (2017). A notable peculiarity of OSCAR is its implementation of stratospheric concentrations for methane, nitrous oxide and halogenated compounds, which follows a time-lagged linearisation of the corresponding tropospheric concentration based on Newman et al. (2007).

The methane representation accounts for four sinks: OH tropospheric oxidation, stratospheric loss, soil uptake and OML uptake. These oxidation processes are modelled through a complex scheme depending on several quantities: temperature, atmospheric methane concentrations, stratospheric ozone concentrations (related to OH production), and emission of three ozone precursors ($NO_x$, CO, NMVOCs), although not all quantities are relevant for all processes. It also includes an estimate of wetland methane emissions, calculating both changes in wetland area and in emissions per unit of area. Wetland area depends on atmospheric $CO_2$, local temperature, and local precipitation, while emissions per unit of area scale based on total heterotrophic respiration since v3.1 Gasser et al. (2020).

Similar schemes are used to emulate the concentrations of $N_2O$ and halogenated species. For $N_2O$, a single atmospheric sink – the stratospheric sink – is considered. In contrast, halogenated species are subject to three atmospheric sinks: tropospheric





OH oxidation, stratospheric oxidation, and surface oxidation comprising both land and ocean contributions. The lifetimes associated with these these sinks for both $N_2O$ and halogens can vary based on several climate factors, such as the stratospheric concentrations of the corresponding species, the equivalent effective stratospheric chlorine and global temperature, with the specific dependencies varying based on the particular sink. For species where some of these processes are negligible or irrelevant, an infinite lifetime is defined.

#### 4.12.2   Radiative forcing

OSCAR includes a comprehensive list of forcing agents: carbon dioxide, methane, nitrous oxide, 37 halogenated species, tropospheric and stratospheric ozone, aerosols (direct and indirect), stratospheric water vapour, albedo change, aviation contrails, volcanic emissions and solar irradiance. The last three are included through prescribed forcing time series, while the rest are computed prognostically by the model.

The calculation of radiative forcing in OSCAR is, generally, simpler than its treatments of gas cycles. In the latest available version, V3.3, OSCAR was updated to use the AR6 expressions (Smith et al., 2021a) to estimate the ERF of long-lived GHGs. However, since this version has not yet been described in the literature (aside from a brief mention in the model's CHANGELOG), and earlier versions remain widely used, we provide a brief overview of the pre-V3.3 forcing calculations, with the caveat that post-V3.3 versions will incorporate AR6 parameterisations.

In earlier versions, $CO_2$ forcing is estimated via the standard logarithmic formula (Myhre et al., 1998), whereas $CH_4$ and $N_2O$ forcings are derived from a modified square root expression accounting for absorption band overlaps, following Myhre et al. (1998, 2013). This square root expression (without overlap effects) is also used to estimate forcing related to water vapour in the stratosphere, based on lagged stratospheric methane concentrations (changed to a simpler linear scaling with methane concentrations in V3.3).

Contributions from halogenated compounds, tropospheric ozone and stratospheric ozone are assumed to scale linearly with the relevant atmospheric species. Ozone contributions are calculated directly from its atmospheric burden, which is based on methane concentrations, ozone precursors ($NO_x$, CO, VOC), stratospheric concentrations of chlorine, bromine, and nitrous oxide, as well as global temperature. Ozone contributions, like aerosols', are regionalised with region-specific weights, although these are logically-different regions from those in the biospheric module. Instead, OSCAR uses values from the four Hemispheric Transport of Air Pollution (HTAP) regions (Fiore et al., 2009; Yu et al., 2013) to derive the values for these weights. In terms of aerosol forcing, OSCAR considers both direct and indirect contributions. The direct contribution follows the linear radiative efficiency approach and considers five sources of anthropogenic aerosols: sulfate aerosols, primary organic aerosols, black carbon, nitrate aerosols, and secondary organic aerosols. Each of these contributions depends on two precursors and global temperature. The indirect effects are scaled linearly on black carbon emissions and logarithmically with the five sources mentioned earlier. Finally, the land surface albedo is based on the LULCC module, which estimates regional changes of land cover. OSCAR takes this estimations, as well as estimations of black carbon deposition on snow and averaged yearly albedo, and calculates the forcing contribution via the associated radiative efficiency. Three forcing agents, aviation contrails, volcanic





aerosols, and solar irradiance, are taken directly as forcing time series from IPCC (2013). For the last two, OSCAR uses forcing efficacies to compute the associated ERF.

### 4.12.3   Temperature

OSCAR employs a two-box formulation of an IRM as EBM (see Sect. 3.3.1). The two constituent boxes are the global surface and the deep ocean. An exchange coefficient governs the heat exchange between the two, while the climate sensitivity parameter translates the total radiative forcing into a heat flux. This formulation also includes two inertia factors to modulate the time lag in the temperature response of the two layers. Thus, an estimation for the global temperature anomaly is produced, which is then scaled linearly to the different regions in the model. Additionally, OSCAR also computes an estimation for the ocean heat

content, based on the total forcing, the ocean temperature, and the climate sensitivity.

## 4.13   ESMICON

The Earth System Model Integrating Cycle of Nature (ESMICON), originally named Earth System Climate Interpretable Model (ESCIMO), is a system dynamics model designed to simulate the multiple feedback processes present in the climate system. System dynamics is a modelling framework that focuses on stocks, flows and feedback loops to represent the behaviour of

complex systems (Naugle et al., 2024; Sterman, 2018). ESMICON stands out as the only SCM using this framework, making it a unique contribution to the field. A consequence of this choice is the fact that ESMICON is one of only two models in this review (along with EM-GC) that does not employ an EBM to estimate global surface temperature increases. First introduced by Randers et al. (2016), ESMICON covers three broad areas: global carbon flows, global energy flows and global albedo change. The model simulates a relatively extensive list of processes, with a total of 50 non-linear differential equations.

ESMICON has been used to analyse the effects of various policy interventions (Randers et al., 2016), such as stratospheric aerosol injection and tropical deforestation cessation. It has also been used to explore the potential for permafrost thawing to continue warming the planet after net zero is achieved (Randers and Goluke, 2020). Additionally, ESMICON serves as the climate submodule in the Earth3 model, a socioeconomic-biophysical model developed to analyse the challenges of achieving the three environmental global Sustainable Development Goals (SDG) while simultaneously pursuing the remaining 14 SDGs

(Randers et al., 2019).

### 4.13.1   GHG concentrations

ESMICON possesses a variable called "concentration of greenhouse gases in the atmosphere" which is made up of two contributions: carbon and methane. Furthermore, methane is only contemplated as a product of permafrost thawing, and it is expressed in $CO_2$ equivalent units, like the GHG concentration variable. Therefore, ESMICON only resolves the ecosytem

cycle of $CO_2$. Its system dynamics framework is similar to a box-based model (see Sect. 3.1.2), being based on inventories and fluxes.





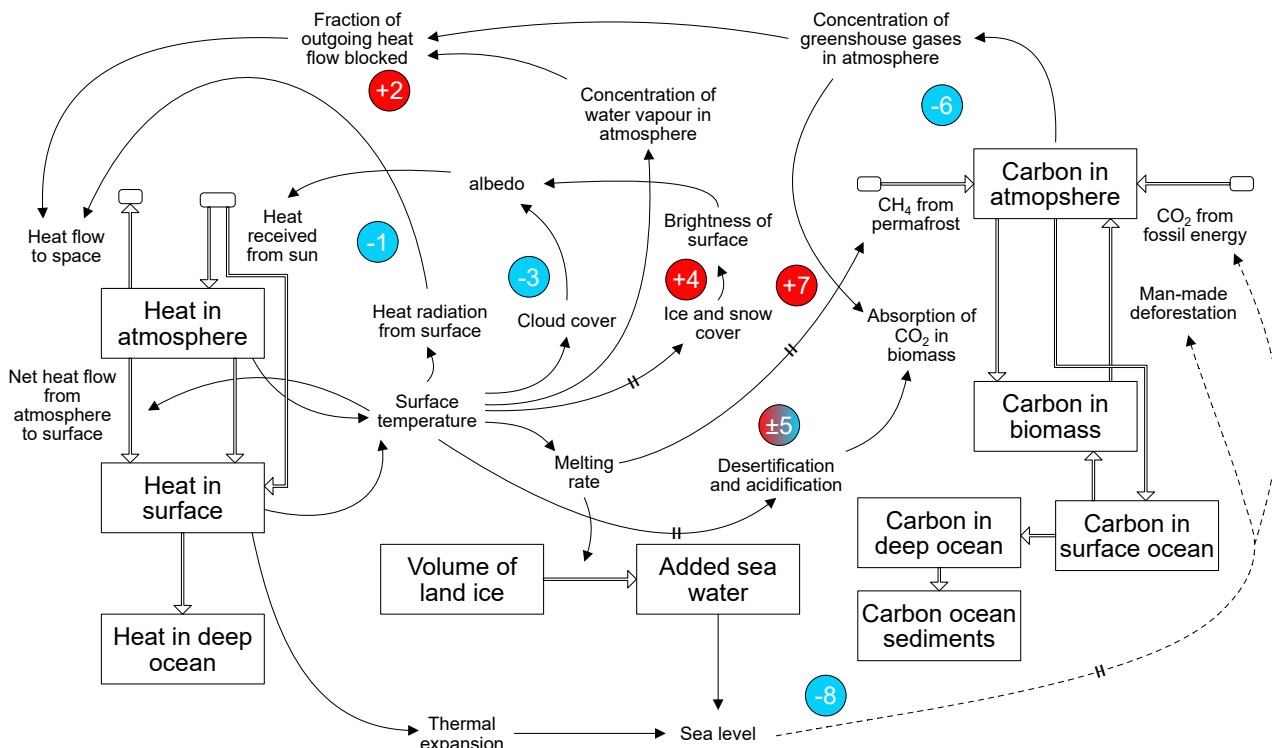

**Figure 10.** Main feedback loops in the ESMICON SCM. Numbers identify the eight main loops, along with the feedback polarity: negative (blue colour and minus sign), positive (red colour and plus sign) or mixed (red-blue gradient and plus-minus sign). Image based on Randers et al. (2016, Fig. 2). Single arrows denote information flows, while double arrows denote material flows of carbon and heat. Two parallel lines crossing the arrow represent a long delay.

The carbon cycle in ESMICON consists of seven pools: fossil reserves, atmosphere, biomass, permafrost, surface ocean, deep ocean and sediments. Fig. 10 shows a diagram of these pools in its right side. As a module following system dynamics principles, the carbon cycle in ESMICON is governed by a series of feedback processes (Randers et al., 2016):

– Anthropogenic emissions: carbon is emitted into the atmosphere as a result of fossil fuel combustion. This is required input by the model.

– Ocean surface layer diffusion: carbon is transferred from the atmosphere to the ocean's surface layer via a chemical diffusion, modulated by the concentration difference between these two carbon pools.

– Deep ocean transfer: carbon is transferred from the ocean's surface layer to the deep ocean, influenced by the long-term mean speed of downwelling and upwelling waters.





- Sedimentation: carbon is deposited at the ocean floor, transferring from the deep ocean pool to the sediments pool. This flow is proportional to the carbon content in the deep ocean.

- NPP: the net gain of carbon in the biomass pool, drawn from the atmospheric pool. It increases with higher atmospheric $CO_2$ concentrations and decreases with higher temperatures.

- Ocean biomass gain: carbon is absorbed by the biomass pool from the ocean pool. This gain increases with higher $CO_2$ concentration and decreases with higher ocean temperatures.

- Fire: carbon is released from the biomass pool into the atmosphere due to fire. It is proportional to GMST anomaly.

- Permafrost thawing: carbon is released from the permafrost pool as it thaws, with the release rate being proportional to the GMST anomaly.

**4.13.2   Temperature**

ESMICON does not use the concept of radiative forcing to determine the temperature anomaly. Instead, it tracks the distribution of heat across the system, and estimates GMST anomaly based on the heat inventories from the atmosphere and surface. The changes in temperature and atmospheric $CO_2$ concentration impact various feedback loops driving the model's dynamics, and, ultimately, the heat distribution. The eight primary feedback loops, as identified in Fig. 10, are:

- Higher GMST leads to increased outgoing radiation to space (negative feedback).

- Higher GMST leads to higher atmospheric water vapour concentrations, which increases radiation retention by the atmosphere (positive feedback).

- Higher GMST leads to increased low-cloud coverage, increasing Earth's albedo (negative feedback).

- Higher GMST leads to increasing ice and snow melt, reducing Earth's albedo (positive feedback).

- Higher GMST leads to higher biomass growth, reducing atmospheric $CO_2$ concentration. This balancing feedback results from the opposing concomitant effects of $CO_2$ fertilization, which promotes growth, and desertification and ocean acidification, which hinder it. This loop can turn into a positive feedback if the latter effects dominate.

- Higher $CO_2$ concentrations lead to increased $CO_2$ absorption by terrestrial and maritime components. This is mainly a negative feedback loop, as $CO_2$ absorption by vegetation and ocean increases with higher atmospheric $CO_2$ concentra-
tions. However, these effects can be counteracted by reduced increase in plant uptake when carbon ceases to be a limiting factor in plant growth and by reduced carbon uptake in a more acidic ocean surface layer.

- Higher GMST leads to permafrost thawing, releasing more carbon into the atmosphere (positive feedback).

- Higher GMST leads to increases in the sea level, promoting a reduction in anthropogenic emissions (negative feedback). This loop is often not activated. Randers et al. (2016), for instance, disabled it for their model analysis.



Fig. 10 provides a diagram of these eight primary feedbacks loops, which explain most of the dynamics in the ESMICON SCM. More processes are included in the model (e.g., low and high cloud reflection and radiation, convection, evaporation, volcanic aerosols, sunspots, desertification), with a comprehensive list available in the original publication. Notably, these feedback loops are often non-linear, with many processes in ESMICON exhibiting time delays and saturation or depletion effects.

## 5   Discussion

This review examines the suite of SCMs participating in RCMIP, detailing their components and development history. The aim was to provide clarity on the current landscape of SCMs by identifying the processes represented by each model, their respective implementations, and the commonalities shared across different models.

    Given that atmospheric concentration of carbon dioxide is the primary driver of climate change, nearly all reviewed models internally simulate $CO_2$ dynamics, barring EM-GC and GREB. Two different approaches are typically employed for this representation: box models and IRMs, with some models using both – applying one to the land component and the other to the ocean component (see Table 1). A particularly noteworthy scheme in this area is the OML-IRM devised by Joos et al. (1996), which couples a parameterisation of carbon uptake by the OML with an IRM emulating carbon export to the deep ocean. This scheme is shared by three widely used SCMs: MAGICC, OSCAR and CICERO-SCM. Following developments in more complex ESMs, SCMs have increasingly integrated representations of additional climate processes. Notably, several models (ESMICON, Hector, MAGICC and OSCAR) now include permafrost thaw dynamics, typically modelled by introducing additional carbon reservoirs that activate under elevated temperatures, releasing carbon (and, in some cases, methane). Similarly, SCMs have begun incorporating processes previously exclusive to ESMs, such as wildfire (ESMICON, OSCAR), precipitation variability (GREB, OSCAR), and nitrogen cycle interactions (MAGICC). However, given the inherent simplicity of SCMs, these representations rely on substantial simplifications and parameterisations compared to ESMs. Additionally, some models go beyond the standard global representation of carbon stocks, incorporating spacially resolved elements such as biome-specific (Hector and OSCAR) or region-specific (OSCAR) carbon pools, which evolve independently with distinct fluxes and parameters.

    Beyond carbon dioxide, the representation of gas cycles for other GHG species varies across models. Several models (AR5-IR, EM-GC, ESMICON, GREB, MCE, WASP) do not resolve any non-$CO_2$ gas cycles, although EM-GC and MCE allow for the inclusion of prescribed concentrations time series for some of these species. In contrast, the remaining models (ACC2, CICERO-SCM, FaIR, Hector, MAGICC, OSCAR, and SCM4OPT) all include a mass balance representation of gas cycles for the other two major GHGs, $CH_4$ and $N_2O$, accounting for global sources and sinks. This treatment is also extended to halogenated species, although the number of included gases varies across models. Methane lifetime is typically computed by considering multiple atmospheric sinks via Eq. 2. Furthermore, more complex models (ACC2, MAGICC, OSCAR) incorporate additional parameterisations for key atmospheric processes, including species interactions with hydroxyl (OH) radicals and Brewer-Dobson circulation effects (MAGICC).



The calculation of radiative forcing varies across models in both number of included forcing agents and the parameterisations used for their estimations. AR5-IR exclusively calculates forcing from carbon dioxide, whereas pre-V3 WASP only distinguishes between non-$CO_2$ Kyoto protocol and non-Kyoto protocol agents, which must be provided as external forcing time series. All other models, except ESMICON, GREB, and WASP V3, estimate forcing contributions from methane, nitrous oxide, and a varying number of halogenated compounds based on their internally simulated concentrations and emissions. Beyond these core GHG contributions, models typically include a subset of additional forcing agents, either parametrising their effects or directly ingesting prescribed forcing time series, as summarised in Table 2. Parameterisations generally follow analytical expressions established in the literature, with the formulations of Myhre et al. (1998) and Etminan et al. (2016) being particularly widely adopted for $CO_2$, $CH_4$ and $N_2O$ estimates. Minor GHG forcing contributions are often computed using linear scaling based on radiative efficiencies, typically following the latest IPCC assessment data. ESMICON and GREB diverge from conventional approaches due to their distinct design philosophies, with GREB being the only gridded model in the review, and ESMICON following a system dynamics philosophy. This leads to the inclusion of non-standard forcing contributions such as latent heat cooling, turbulent heat exchange, and albedo changes driven by cloud and land-cover changes.

The conversion of total radiative forcing to temperature anomaly is typically accomplished through EBMs. With the exception of two models – EMGC, which employs a linear regression model, and ESMICON, which relies on heat inventories – all reviewed SCMs use EBMs. These are implemented in one of two equivalent formulations (see Sect. 3.3.3): layer models (ACC2, CICERO-SCM, GREB, Hector, MAGICC, OSCAR, SCM4OPT, WASP) or IRMs (AR5-IR, FaIR, MCE). Two notable schemes are shared by multiple SCMs: DOECLIM (Kriegler, 2005) and UD-EBMs (Hoffert et al., 1980; Wigley and Raper, 1987). DOECLIM, used by ACC2, Hector and SCM4OPT, combines a zero-dimensional EBM with a one-dimensional ocean heat diffusion model. Meanwhile, UD-EBMs, shared by CICERO-SCM and MAGICC, simulate ocean heat transport through both diffusion and advection, emulating the effects of ocean circulation. Most of these EBMs lack any representation of internal variability, with the exception of EM-GC, FaIR, SCM4OPT, and WASP. EM-GC and SCM4OPT employ parameterisations for ocean-driven variability, but rely on historical time series data, limiting their applicability to future projections. In contrast, FaIR and WASP introduce stochastic noise terms in their expressions estimating forcing (FaIR) and temperature anomaly (FaIR and WASP), allowing variability to influence both historical simulations and future projections.

This study has focused on reviewing model structure and process representation, assuming that model calibration is performed employing appropriate data sets at the time of calibration. While differences in calibration methodologies and data can introduce significant variations in model output – particularly since models can and often are calibrated at different times using different datasets – a comprehensive review of model calibration was deemed out of scope for this study. This decision was based on two main considerations: (i) a comprehensive calibration review would substantially increase the length and complexity of this analysis, and (ii) calibration approaches are subject to frequent revisions, making any such review likely to soon become obsolete. Nevertheless, some details on model calibration have been included when relevant, with additional information available in the comprehensive list of model references cited in this text.



Similarly, this study does not address technical implementation details beyond what is offered in Table 4. While such details can be important for SCM developers, they are often inadequately documented in model publications and may be of limited relevance to most users. Consequently, a decision was made exclude technical implementations aspects from this review.

Finally, we reiterate that this review does not provide an exhaustive account of every SCM in the literature. While ambitious in scope, a limit on the number of models included was necessary to maintain focus and conciseness. As a result, only participating models in RCMIP phases 1 (Nicholls et al., 2020) and 2 (Nicholls et al., 2021) were reviewed. This selection was intended as a reasonable proxy for the most widely used and actively developed models, aiming to meet the needs of most users and developers. Future RCMIP phases may include a different set of models, creating opportunities for further reviews that cover additional models and updates.

## 6 Conclusions

This study provides a review of the fundamental principles underlying SCMs and the mechanisms by which they generate climate projections. A detailed description of all models participating in the RCMIP exercise has been presented, structured around the three key stages of the emissions-climate change cause-effect chain – GHG concentrations, radiative forcing and temperature anomaly – where relevant. By providing clarity on how these models represent various climate processes and identifying their key differences, we aim to enhance understanding among both developers and users while also informing about the implications of selecting one model over another.

*Code availability.* This review paper does not present new data or model code. However, it discusses several existing climate models, many of which have publicly available code. Links to the open-source versions of these models are provided in Table 4. Archived copies in persistent repositories, where code availability and licensing allows, are detailed below.

The ACC2 model code is available upon request from the original authors.

The AR5-IR model code is available upon request from the original authors, although a copy of the version used in RCMIP is available in the OpenSCM repository: https://github.com/openscm/openscm/blob/ar5ir-notebooks/notebooks/ar5ir_rcmip.ipynb (last access: 5 June 2025, Nicholls). A copy of this repository is archived at https://zenodo.org/records/15600556 (Romero-Prieto, 2025).

CICERO-SCM's latest reviewed code is available at https://doi.org/10.5281/zenodo.10548720 (Sandstad et al., 2024b).

The EM-GC model code is available upon request from the original authors.

The ESMICON model along with its documentation can be downloaded from http://www.2052.info/escimo/ (last access: 5 June 2025).

FaIR's latest reviewed code is available at https://doi.org/10.5281/zenodo.10566813 (Smith, 2024)

GREB's latest reviewed code is available at https://doi.org/10.5281/zenodo.2232282 (christianstassen, 2018).

Hector's latest reviewed code is available at https://doi.org/10.5281/zenodo.10698028 (Dorheim et al., 2024a).

The Held et al. two layer model implementation used in the RCMIP study is available in the OpenSCM repository at https://github.com/openscm/openscm/blob/ar5ir-notebooks/notebooks/held_two_layer_rcmip.ipynb (last access: 5 June 2025, Nicholls). A copy of this repository is archived at https://zenodo.org/records/15600556 (Romero-Prieto, 2025).

MAGICC's latest reviewed code is available at https://zenodo.org/records/15600556 (Romero-Prieto, 2025).



MCE's latest reviewed code is available at https://zenodo.org/records/5574895 (tsutsui1872, 2021).

OSCAR's latest reviwed code is available at https://zenodo.org/records/15600556 (Romero-Prieto, 2025).

SCM4OPT's latest reviewed code (part of the CB-IAM model) is available at https://doi.org/10.5281/zenodo.11928479 (Su, 2024).

WASP's latest reviewed code is available at https://zenodo.org/records/4639491 (WASP Earth System Model, 2021).

*Author contributions.* Alejandro Romero Prieto led the conceptualization, analysing, and writing of the manuscript. Chris Smith and Camilla Mathison contributed to conceptualization, writing (reviewing and editing), and supervision.

*Competing interests.* The authors declare that they have no conflict of interest.

*Acknowledgements.* The authors would like to thank the developers of simple climate models who graciously provided feedback on their models: Dietmar Dommenget, Kalyn Dorheim, Thomas Gasser, Philip Goodwin, Tanaka Katsumasa, Zebedee Nicholls, Ross Salawitch, Marit Sandstad, Xuanming Su, and Junichi Tsutsui. Alejandro Romero Prieto also thanks Piers Forster for his guidance throughout the process, and Chris Wells for his feedback on the text.

*Financial support.* This work was supported by the Leeds-York-Hull Natural Environment Research Council (NERC) Doctoral Training Partnership (DTP) Panorama under grant NE/S007458/1. Chris Smith was supported by the European Union's Horizon 2.5 Climate Energy and Mobility programme under grant agreement No. 101081661 (WorldTrans). Camilla Mathison was supported by the Joint UK BEIS/Defra Met Office Hadley Centre Climate Programme (GA01101)





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
