# Peer review of "Review of climate simulation by simple climate models"

_EGUsphere, 2025_

## Author Comment (AC1)

**Response to reviewers**

Thank you to the reviewer for your detailed review of our manuscript. We have responded to each comment in full and outlined the changes we will make to the manuscript to address your comments in this document. Our responses are in black font in response to review comments in blue, and where we quote new text, this is in italic.

**Reviewer comments**

I personally enjoyed reading this comprehensive and ambitious review on SCMs. I believe it will serve as a valuable cornerstone for the field, particularly for newcomers from different disciplines. I especially appreciated the synthesis presented in Tables 1–4 and Figures 1–2, as well as the historical perspective provided on both the general development of SCMs and the evolution of individual models. The authors have reviewed an impressive amount of material with great thoroughness, and I really appreciate this effort. The balance between high-level discussion and technical detail is particularly well thought of – the level of detail is sufficient to provide clear context without overwhelming the reader. I only have a few clarifications/suggestions that I'm outlining here below:

Thanks for your kind words and comments. The intention of this manuscript, which evolved from a literature review made by the lead author as part of his PhD, was precisely to serve as a guide to newcomers to the field as well as SCM users/developers. Hearing that this objective may have been fulfilled is very encouraging. We respond to each comment in turn here and in the revised manuscript.

1. L37: "Downscalling" might be a typo

Thank you, it was indeed a typo. Now corrected in the text.

2. L98: "rather than approximations"; this is a bit nitpicky, but I would argue that also SCMs are driven by fundamental laws of physics (e.g. mass balance, energy balance), except for perhaps purely regression-based models. The difference is that loss rates for chemical species and radiative transfer for heat are highly parameterized in SCMs (first-order kinetics, linear forcing-feedback for heat) but better resolved in ESMs.

We agree. We have slightly reframed the text to focus instead on the methodological differences: ESMs aiming to resolve processes explicitly (as much as feasible) while SCMs aim to produce accurate parametrisations that emulate the effects of those processes. The text now reads:

*"This progress culminated in the development of the first AOGCM by Manabe and Bryan (1969), marking a pivotal shift towards physically-complex climate models aiming to explicitly resolve processes, rather than relying on highly-parametrised approximations."*

3. L305: I would stress here that lambda is arguably the most important parameter in this formulation, and that is a strong control on climate sensitivity (the central quantity in climate science). I think that is mentioned later, but I wonder if the casual reader would think at this stage that lambda is basically constrained by $4\sigma T_0^3 \sim 5$ W m-2 K-1, which is actually pretty far from the real feedback (~1 W m-2 K-1 for a climate sensitivity of ~3.5K). In your formulation, that means that tuning k is a really important step for any SCM. In other words,

there are lot of things that happen in SCMs but perhaps the most consequential parameter to set in an EBM is lambda (and I would probably call out here that k would need to take into account all the important climate feedbacks such as water vapor, lapse rate, clouds, albedo... and that all of that complexity is swept under the number assigned to lambda).

We agree with the review that this is a critical quantity in any SCM, and that it would be beneficial to point this out explicitly in the text. We have modified the paper to read as follows:

"... While seemingly simple, Eq. 7 is arguably the single most important parametrisation in most SCMs, as it controls the model's temperature response for a given forcing. The k parameter (or, indirectly, the λ parameter) abstracts away the great complexity of the climate system and its numerous feedbacks effects (e.g., albedo change, aerosol interactions, etc.). In most instances, this is a constant model parameter that can be tuned to emulate results from other, more complex ESMs. However, more complex formulations are possible, like MAGICC's (Sect. 4.9.3) time-varying, and WASP's (Sect. 4.5.3) forcing-agent-specific time-varying λ parameters."

4. L352: not sure if I follow the "reproduce non-linearities" part. Even with n-layers Equation (11) is still a linear system of ODEs. I would also call out here that adding n boxes for the ocean does not alter the equilibrium solution, which is still F/k1 for all temperatures, because at equilibrium T1=T2=...=Tn. In other words, having n boxes allow for extra flexibility in setting the time scales of response (the "transient"), but not the final surface temperature equilibrium, which is set by forcing F and feedback k1 (using the notation of Equation 11)

Thank you for raising this, particularly the insightful point about the difference between transient and equilibrium dynamics. We agree that Eq. 11 still describes a linear model, regardless of the number of layers. The point we wanted to raise here was the enhanced ability to describe complex dynamics in the transient response, which we have now amended the text with, along with the remark regarding the immutability of the equilibrium solution:

*"A higher number of layers increases the number of distinct timescales in the system (see $\tau_i$ in Eqs. 12 and 13), and therefore increases the complexity the model is able to display in its transient response. The equilibrium state, however, remains independent from the number of layers, as Eq. 11 reduces to $T_1 = F/\kappa_1$ when $T_1 = T_2 = \dots = T_n$."*

5. Table 4 and beyond: there are a lot of github links in Table 4, and other URLs across the paper that point to model websites. I see why this is very useful for the reader interested in the code, but I worry that in the relative near future (e.g. >5 years) some of those links might be broken since they are not "permanent" repositories. Since the paper is typically a stand alone contribution that lives in eternity (hopefully :)), I wonder if the temporary http links should be added as a separate supplement that says "current links" or something along those lines. On the other hand I do see the value of having the links there in the near-term; I just wanted to flag this as a potential issue thinking longer-term.

Thank you for our comment. While we agree that these links will eventually all be broken, we think that the usefulness from a user/reader perspective to have them easily findable outweighs the eventual redundancy of having a list of broken links. However, to partially mitigate this eventual redundancy, we have added a mention in the table caption to the Zenodo archive associated with the paper that stores a permanent copy of all open-source models reviewed in the text: *"A*

*permanent archive of the publicly-available models reviewed in this work can be found in the* Code availability *section".*

6.  I really enjoyed the summary info provided by Table 1-4 and Figure 1-2, and I appreciate the authors' efforts to summarize a large amount of literature so effectively. I wonder if maybe having an additional column/table with (1) a typical use case where the SCM has been used and (2) a specific aspect of the model that distinguishes it from others could be useful for the reader to identify straight away what model they might be interested in. Or maybe a combination of (1) and (2). For example for OSCAR, that "peculiar" thing that distinguishes it from other models would be strong focus on LULCC and carbon cycle, and if a reader is interested in studying LULCC effects they would immediately know that OSCAR is probably the first thing to look at. For MAGICC that would be the long history and the strong use in IPCC. For GREB that would be the focus on "understanding" the climate system and university teaching. Just some food for thought – Not sure if that would play out well with all the models so feel free to disregard if not.

We agree that a brief summary concerning the nature and focus of the different models reviewed in the paper would be valuable for the casual reader that does not necessarily want to read the detailed descriptions of all the models. However, the problem with assigning a "good-for" label for each model is that it may be clear for certain SCMs (e.g., OSCAR for the carbon cycle, GREB for grid-based process resolving), but there are others without clear areas of focus. This is why originally the paper did not include such a discussion. However, prompted by this comment (and a similar one from RC1), we wrote a synthesis in the discussion section that divided SCMs into "generalist" models, without any strong focus, and "specialist" SCMs that do possess such a focus. This allowed us to point out such specialist focus when it clearly exists, while avoiding assigning it when it is not present or it is not as clear. Given the nature of the review, which only examines the structure of the models without delving into the performance/accuracy, we think this is a suitable method to present this information to the user community. We didn't put this into a table format because we felt that would convey too much weight to arguably one of the most subjective sections of the review, which may not necessarily align with the opinions of model developers. We have tried to be as free from value judgements as possible in this paper, and we preferred to handle this opinion in text form in the discussion section to allow for a more nuanced treatment. The paper now includes the following text in the "Discussion" section (which precedes the more detailed model summary already present in the previous iteration of the text):

"*The aim of this review was to provide clarity on the current SCM landscape by identifying the processes represented by each model, their respective implementations, and the commonalities shared across different models. This was achieved by reviewing the suite of SCMs participating in RCMIP, detailing their components and development history. Ultimately, we hope this texts serves as a valuable guide for the difficult task of SCM selection. While other considerations such as accuracy, calibration or usability are important when selecting a model, clarifying which processes a model resolves is a critical first step to assess model suitability for a particular use. Consequently, we offer here a brief summary of the model descriptions presented in section 4, first classifying SCMs in two broad families based on design philosophy, and then summarising the commonalities and differences of the models included in this review. Finally, we conclude with a brief discussion of the limitations of our review.*

*When selecting an SCM, it is important to recognise that different models were developed with different intended applications and design philosophies. Broadly, SCMs can be grouped into what might be termed "specialist" and "generalist" models. Specialist models are developed around*

*clearly defined objectives or processes. Examples include AR5-IR, which provides a minimal framework to estimate warming from CO2 concentrations; EMGC, which explicitly represents natural variability; ESMICON, designed within a system dynamics framework; GREB, whose spatio-temporal resolution and process representation are designed to support a physical understanding of climate and its teaching; OSCAR, which focuses on the carbon cycle and LULCC disturbances; and WASP, with an emphasis on ocean dynamics. In contrast, generalist SCMs can be viewed as models developed without prioritising any particular component or objective beyond providing climate simulations. Note, however, that the design philosophy should be evaluated across the whole model lifetime, as this distinction is likely violated in the short-term (e.g., FaIR was initially developed as an extension of AR5-IR consisting of a small set of equations to produce warming estimates, arguably qualifying as an "specialist" SCM, but has evolved considerably since then). The most representative example of a generalist SCM is MAGICC, as the oldest SCM in this review with an extended history of usage to generate climate projections. Other models in this category arguably include ACC2, CICERO-SCM, Hector, FaIR, MCE, and SCM4OPT. Reflecting their broad scope, several generalist models (ACC2, Hector, MAGICC, SCM4OPT) have been coupled as climate modules within IAMs, while others (FaIR, MAGICC) have been widely used across multiple IPCC assessment reports. This specialist-generalist split can be a useful first step to evaluate which SCM to use, particularly if a given specialist design philosophy aligns with the requirements of the user. However, a more detailed understanding of the processes and methods used by different SCMs is likely still required after this first step, which we offer below."*

7. L785: I would point out that GREB has the sensible/latent fluxes as explicit terms because they are looking at the surface energy balance, in contrast with all other models that are looking at the energy balance from the top-of-atmosphere (hence they only have radiative fluxes F and \lambda*T and no turbulent fluxes). In other words, the "flavor" of GREB's EBM is quite different from the typical TOA EBMs that most SCMs use, despite still balancing energy fluxes.

Thank you for your interesting remark, which we had overlooked in the text. We agree that this is relevant for the reader wishing to understand the how GREB works and how it differs from other models, so we have added the following text:

"This EBM possesses two main peculiarities that set it apart from other SCMs in this review. First, this EBM focuses on the surface energy balance, as opposed to other SCMs that focus on the top-of-atmosphere energy balance. This allows GREB to include explicitly turbulent heat fluxes like the sensible and latent heat fluxes"

8. L832: I think Hartin (2015) is cited twice, might be a typo

Thank you. Now corrected in the text.

9. L1015: by "this publication" do you mean Meinshausen's or your own article? Maybe that should be clarified since it could be interpreted as both.

We meant the Meinshausen publication. Now modified to "that publication" instead of "this publication" for clarity.

10. L1472: Not sure if I follow this part (not familiar with ESMICON) – "tracking the distribution of heat across the system" and "heat inventories" sound like an energy balance model to me (i.e., calculate fluxes in and out of predefined boxes...). I would also add that EBMs also

include all the listed feedbacks (increased OLR, increase water vapor, clouds, etc.. they just do it through a lambda parameter). Can this idea of "heat inventory" be clarified if it is completely different than traditional EBMs? In this framework, how is a CO2 perturbation added if not through extra energy (radiative forcing)?

Thank you for raising this, as the text may have not been clear enough on this regard. Although using a different framework and vocabulary, we agree that the description of the temperature module in ESMICON effectively works as an EBM. The key distinction between ESMICON and other SCMs is the absence of a concept of "radiative forcing", with the various processes such as GHG-induced warming or ice-volume having a direct impact on several feedback loops that ultimately affect the heat distribution in the system. We have updated the text to make this clearer:

"While not using explicitly an EBM framework to determine temperature anomalies, ESMICON uses instead an analogous system dynamics framework to determine those anomalies based on the distribution of heat across the system. In particular, it uses the heat inventories from the atmosphere and surface, along with the associated heat capacities, to estimate GMST. The key distinction between the more common EBM and ESMICON's temperature module is the absence of an explicit1500 radiative forcing concept and the parametrisation of feedbacks through a $\lambda$ parameter, as in Eq. 8. Instead, state variables in the system, such as CO2 concentrations and ice volume, impact directly various feedback loops driving the model's dynamics, and, ultimately, the heat distribution."

**Other changes**

Following a request from the authors, in section 4.9.1 (~L1110) we modified the phrasing regarding MAGICC's vegetation regrowth parameter. The text now reads:

"To better approximate real-world dynamics where land-use changes usually result in persistent alterations to carbon stocks, the MAGICC carbon cycle module uses a1095 regrowth fraction parameter to adjust the turnover time and thereby allow for partial regrowth (Meinshausen et al., 2011a)"

We also changed incorrectly attributed citations to

Meinshausen, M., Wigley, T. M. L., and Raper, S. C. B.: Emulating atmosphere-ocean and carbon cycle models with a simpler model, MAG-2050 ICC6 – Part 2: Applications, Atmospheric Chemistry and Physics, 11, 1457–1471, https://doi.org/10.5194/acp-11-1457-2011, publisher: Copernicus GmbH, 2011c

To the relevant publication:

Meinshausen, M., Raper, S. C. B., and Wigley, T. M. L.: Emulating coupled atmosphere-ocean and carbon cycle models with a simpler model, MAGICC6 – Part 1: Model description and calibration, Atmospheric Chemistry and Physics, 11, 1417–1456, https://doi.org/10.5194/acp-2045 11-1417-2011, 2011a.

---

## Author Comment (AC2)

**Response to reviewers**

Thank you to the reviewer for your detailed review of our manuscript. We have responded to each comment in full and outlined the changes we will make to the manuscript to address your comments in this document. Our responses are in black font in response to review comments in blue, and where we quote new text, this is in italic.

**Reviewer general comments**

Simple Climate Models (SCMs) make up a critical component of the climate model hierarchy and have been used for decades for climate assessment. Although many SCMs with varying levels of complexity exist, there have been few efforts to clarify differences in model structure. In this work, the authors review the 14 SCMs participating in the reduced-complexity model intercomparison project (RCMIP), organizing them by increasing complexity and creating a guide for developers and users.

Overall, this is a strong manuscript with a clear contribution in providing a comprehensive overview of some of the most widely used SCMs. The methodological choices of both which SCMs to include and what components to focus on (e.g. choosing structure over performance) are clear and sufficient. I have one major comment about the stated goals/conclusions of the manuscript. The rest of my questions/suggestions to improve the quality of the manuscript are relatively minor.

Thanks for your comments, we respond to each comment in turn here and in the revised manuscript.

**Specific Comments:**

The paper aims to support informed use of SCMs, e.g. 1571-1572: "… while also informing about the implications of selecting one model over another." While the detailed descriptions achieve this, the discussion could be strengthened by adding a paragraph that informs practical guidance for model selection. For example, you could briefly summarize which models are best suited for specific applications based on their features. This type of synthesis would make the paper's thorough exploration of SCMs more accessible to the user community.

We agree that a paragraph briefly summarising the nature and focus of the different models reviewed in the paper would be valuable for the casual reader that does not necessarily want to read the detailed descriptions of all the models. However, the problem with assigning a "good-for" label for each model is that it may be clear for certain SCMs (e.g., OSCAR for the carbon cycle, GREB for grid-based process resolving), but there are others without clear areas of focus. This is why originally the paper did not include such a discussion. However, prompted by this comment (and a similar one from RC2), we wrote a synthesis in the discussion section that divided SCMs into "generalist" models, without any strong focus, and "specialist" SCMs that do posses such as focus. This allowed us to point out such specialist focus when it clearly exists, while avoiding assigning it when it is not present or it is not as clear. Given the nature of the review, which only examines the structure of the models without delving into the performance/accuracy, we think this is a suitable method to present this information to the user community. The discussion now includes the following text (which precedes the more detailed model summary already present in the previous iteration of the text):

"*The aim of this review was to provide clarity on the current SCM landscape by identifying the processes represented by each model, their respective implementations, and the commonalities shared across different models. This was achieved by reviewing the suite of SCMs participating in RCMIP, detailing their components and development history. Ultimately, we hope this texts serves as a valuable guide for the difficult task of SCM selection. While other considerations such as accuracy, calibration or usability are important when selecting a model, clarifying which processes a model resolves is a critical first step to assess model suitability for a particular use. Consequently, we offer here a brief summary of the model descriptions presented in section 4, first classifying SCMs in two broad families based on design philosophy, and then summarising the commonalities and differences of the models included in this review. Finally, we conclude with a brief discussion of the limitations of our review.*

*When selecting an SCM, it is important to recognise that different models were developed with different intended applications and design philosophies. Broadly, SCMs can be grouped into what might be termed "specialist" and "generalist" models. Specialist models are developed around clearly defined objectives or processes. Examples include AR5-IR, which provides a minimal framework to estimate warming from CO2 concentrations; EMGC, which explicitly represents natural variability; ESMICON, designed within a system dynamics framework; GREB, whose spatio-temporal resolution and process representation are designed to support a physical understanding of climate and its teaching; OSCAR, which focuses on the carbon cycle and LULCC disturbances; and WASP, with an emphasis on ocean dynamics. In contrast, generalist SCMs can be viewed as models developed without prioritising any particular component or objective beyond providing climate simulations. Note, however, that the design philosophy should be evaluated across the whole model lifetime, as this distinction is likely violated in the short-term (e.g., FaIR was initially developed as an extension of AR5-IR consisting of a small set of equations to produce warming estimates, arguably qualifying as an "specialist" SCM, but has evolved considerably since then). The most representative example of a generalist SCM is MAGICC, as the oldest SCM in this review with an extended history of usage to generate climate projections. Other models in this category arguably include ACC2, CICERO-SCM, Hector, FaIR, MCE, and SCM4OPT. Reflecting their broad scope, several generalist models (ACC2, Hector, MAGICC, SCM4OPT) have been coupled as climate modules within IAMs, while others (FaIR, MAGICC) have been widely used across multiple IPCC assessment reports. This specialist-generalist split can be a useful first step to evaluate which SCM to use, particularly if a given specialist design philosophy aligns with the requirements of the user. However, a more detailed understanding of the processes and methods used by different SCMs is likely still required after this first step, which we offer below.*"

In a similar vein, there are points in the manuscript that explicitly address model advantages, but as a developer/user, I'd also love to see disadvantages/failure modes of these different models. Is there a consistent/rigorous way to define a failure mode of an SCM? If so, why have you chosen not to discuss them?

We could not find any clear, objective way to discuss model disadvantages or model failures by solely looking at model structure and implementations. Model advantages are usually considerably easier to discuss because different schemes are commonly implemented to address certain limitations in previous models. As a result, advantages are usually explicitly stated, while limitations are more difficult to identify. We think model disadvantages or failures are more easily identified when comparing model performance and dynamics, hence better accomplished under the RCMIP efforts. There is currently an ongoing third phase of RCMIP

(https://doi.org/10.5194/egusphere-2025-5775) that will inform about potential model disadvantages in the medium future. We have included a paragraph in the (new) limitation section to explicitly explain this:

" *The discussion of model differences in this review has focused on the processes represented in each SCM and the advantages those representations provide, rather than on potential disadvantages. This was a deliberate choice: we found no consistent or objective method for assessing model drawbacks solely from their technical specifications. Different methodological approaches, most notably RCMIP efforts (Nicholls et al., 2020, 2021; Romero-Prieto et al., 2025) with their systemic evaluation and benchmarking of model outputs, are better suited to address questions of model disadvantages.*"

Minor:

1. The historical overview of the field is comprehensive but ends somewhat abruptly around 2020, and doesn't acknowledge the rapid rise in statistical/machine learning-based approaches. Since that isn't the focus of the paper, it's not a major issue, but a comprehensive review such as this should acknowledge the current state of the field.

Thank you for pointing this omission. We agree that some more discussion around the recent innovations in machine learning is warranted. Towards the end of the historical overview section, we have added the following paragraph:

*In the early 2020s, following the strong demand for climate emulation established in the previous decade and fuelled by the rapid popularisation of artificial intelligence (AI), data-driven emulators rose sharply in prominence (Watson-Parris et al., 2021, 2022; Watt-Meyer et al., 2023; Bassetti et al., 2024). These models typically provide regional rather than global outputs and follow methodological approaches that differ substantially from those of SCMs, using machine learning and AI algorithms. For these reasons, they fall outside the scope of the present review. Readers interested in data-driven climate emulation are referred to the comprehensive review by Tebaldi et al. (2025).*

2. In a similar vein, climate emulators are referenced on line 23 in the introduction, but aren't addressed again. For both this and the previous point consider the question: how would someone coming from the ML/emulator literature connect to your manuscript?

Thank you for raising this point. We agree that the distinction between SCMs and the broader emulator family is important, as it is often neglected in the literature and may lead some readers to confusion. We have updated the text to frame SCMs within the broader emulator family, and mentioned statistical models as the other big family of emulators. The text now reads:

"*Their emulation capability positions SCMs within the broader family of climate emulators, at least when they are operated in an emulator configuration. Although these two terms are sometimes used interchangeably, the ``emulator'' label strictly encompasses a greater number of models than SCMs, as it applies to any model designed to approximate the output of another model. In particular, the emulator label also includes any machine learning and AI approach able to mimic the output of more complex models through statistical learning. The boundary between SCMs and data-driven emulators is not always sharp—some SCMs employ techniques such as*

*impulse-response functions that could be viewed as statistical. Nevertheless, SCMs generally adopt a more mechanistic emulation strategy, grounded in physical reasoning and parametrisation."*

3.  352: What is meant by 'a higher number of layers provides the model with... an enhanced capability to reproduce non-linearities'? Unless I'm misunderstanding Equation 11 (please correct me if this is the case!), this is still a linear model, so what type of non-linearities do you mean?

Thank you for pointing this out. We agree that Eq. 11 still describes a linear model, regardless of the number of layers. The point we wanted to raise here was the enhanced ability to describe complex dynamics in the transient response, which we have now amended the text with:

*"A higher number of layers increases the number of distinct timescales in the system (see $\tau_i$ in Eqs. 12 and 13), and therefore increases the complexity the model is able to display in its transient response. The equilibrium state, however, remains independent from the number of layers, as Eq. 11 reduces to $T_1 = F/\kappa_1$ when $T_1 = T_2 = ... = T_n$."*

4.  366-367: "... they (IRFs) are not derived from fundamental physical principles but through mathematical approximations of the relationships we aim to model." I'm not sure what is meant by this statement, as response functions are, by definition, the exact solution to the type of systems defined by Equations 9-11. You explicitly state this in lines 390-392 as well. The second statement implies they're only approximations to the same extent that 9-11 are approximations as well. Can you be more precise in what you mean by the former statement?

We agree that EBMs (Eq 9-11) and IRFs are both equivalent approximations to the heat dynamics in. The point we wanted to raise here was that the EBM framework is usually derived applying physical arguments to heat transport, while IRFs are a broader mathematical framework that allows an approximation of any non-linear system, independent of physical arguments or the heat transport phenomenon. The paragraph now reads as follows, which hopefully clarifies this point:

*"In the preceding sections, methods were explored that aimed to achieve this emulation through the development of computationally efficient approximations of the Earth system maintaining physical intuition. IRMs (Joos and Bruno, 1996) take a different route; they are not derived from fundamental physical principles but rather, offer a mathematical framework able to approximate the dynamics of any non-linear system through empirical parameter tuning.*

5.  418 (and elsewhere): the term 'complexity' is used with respect to models frequently, but without a rigorous definition. You seem to imply complexity is proportional to the number of physical processes represented, but can you clarify this against e.g. computational complexity?

We agree this clarification is warranted. We have added the following text in the introduction (~L78):

*Throughout this review, "complexity" refers to the conceptual or process-level complexity of a model—i.e., the number and intricacy of physical processes it represents—rather than to other80 notions such as computational complexity.*

6.  Figure 1: I don't follow what 'None' means just from looking at the figure, consider clarifying this in the caption.

We have added the following text to the caption to clarify the meaning of "None": *"No public implementation was found for AR5-IR, so no specific programming language was associated with this model (None)."*

7. Figure 5: Unclear what is meant by 'Earth' from just the figure, consider clarifying this in the figure or caption.

We have added the following text to the caption to clarify the nature of the "Earth" box: *"The "Earth" box in Hector represents long-term carbon storage, including fossil fuels and carbon capture."*

8. You state on 250 that "…SARF should be assumed whenever radiative forcing is mentioned…", this is probably worth highlighting in the caption for Table 2 for readers who are looking primarily at the figures/tables to summarize the review.

Thank you for this very useful comment which has forced us to think more deeply about what models are actually representing in terms of their radiative forcing inputs. In essence, where models are calculating a radiative forcing value derived from concentrations or emissions and passing this onto a temperature module, they are implicitly using ERF, since this is the measure that is most closely related to long-term temperature response (e.g. IPCC AR5 WG1 Chapter 8; AR6 WG1 Chapter 7). However, they commonly use expressions that compute SARF instead. We have therefore changed our discussion of radiative forcing here, and updated Table 2 caption as suggested:

*"However, consideration of these nuances between forcing categories is often inconsistent in SCMs. Models may employing SARF expressions to estimate forcing without any further modifications, effectively treating that quantity as ERF. Generally, unless otherwise stated, ERF should be assumed whenever radiative forcing is mentioned in this document, although the underlying formula may have originally be intended to approximate SARF.*

And in Table 2:

*"Generally, ERF should be assumed, although the underlying mathematical expressions used by the models to compute this quantity may have been originally intended to approximate SARF. More details can be found in section 3.2 and the individual publications."*

**Technical Corrections:**

1. 16: 'system' should be capitalized

Thank you. Now corrected in the text.

2. 37: Typo - 'downscalling'

Thank you. Now corrected in the text.

3. The signposting is very clear throughout, but inconsistent at times. E.g. The GREB signposting (4.6.X) doesn't follow the same format as the other models.

    1. Some places 4.X.1 is labeled model 'specification' and others it's 'description'

Thank you. Now changed EM-GC "*model specification*" to "*model description*" for consistency.

4.  241: Missing a second set of closing parentheses after LULCC? Also should the first letters be capitalized when the acronym is spelled out explicitly?

The parenthesis after LULCC are correct (black carbon is part of the albedo changes). However, while checking the paragraph we did detect a missing parentheses after "for a full list" for minor GHG. We have also capitalised the LULCC acronym components.

5.  342: The 'co2' in the '1pctCO2' experiment is typically capitalized. Consider italicizing experiment names as well to make it clear they're separate from the models.

Thank you. Now corrected in the text. We also italicised 1pctCO2 in L342 and R01 in ~1275

6.  680: Typo - 'continuos'.

Thank you. Now corrected in the text.

7.  833: Same citation listed twice.

Thank you. Now corrected in the text.

8.  941: Units are typically not italicized (should *ppm* be ppm?)

Thank you. Now corrected in the text.

9.  Ensure consistent hyphenation of your acronyms, there's at least one instance of "OML IRM" (no hyphen) on 1019

Thank you. Now corrected in the text to OML-IRM in every instance.

10. 1511: Typo - 'spacially' (not sure if this is a valid alternate spelling, but you spell it 'spatially' on 1127)

Thank you. Now corrected in the text.

11. 1558: Missing word - "... a decision was made (to) exclude..."

Thank you. Now corrected in the text.

12. 1590: Typo - 'reviwed'

Thank you. Now corrected in the text.

**Other changes**

Following a request from the authors, in section 4.9.1 (~L1110) we modified the phrasing regarding MAGICC's vegetation regrowth parameter. The text now reads:

"To better approximate real-world dynamics where land-use changes usually result in persistent alterations to carbon stocks, the MAGICC carbon cycle module uses a1095 regrowth fraction parameter to adjust the turnover time and thereby allow for partial regrowth (Meinshausen et al., 2011a)"

We also changed incorrectly attributed citations to

Meinshausen, M., Wigley, T. M. L., and Raper, S. C. B.: Emulating atmosphere-ocean and carbon cycle models with a simpler model, MAG-2050 ICC6 – Part 2: Applications, Atmospheric Chemistry and Physics, 11, 1457–1471, https://doi.org/10.5194/acp-11-1457-2011, publisher: Copernicus GmbH, 2011c

To the relevant publication:

Meinshausen, M., Raper, S. C. B., and Wigley, T. M. L.: Emulating coupled atmosphere-ocean and carbon cycle models with a simpler model, MAGICC6 – Part 1: Model description and calibration, Atmospheric Chemistry and Physics, 11, 1417–1456, https://doi.org/10.5194/acp-2045 11-1417-2011, 2011a.

---

## Author Comment (AC3)

**Response to reviewers**

Thank you for reading our manuscript and taking the time and effort to leave a community comment to strengthen the manuscript. We have responded to each comment in full and outlined the changes we will make to the manuscript to address your comments in this document. Our responses are in black font in response to review comments in blue, and where we quote new text, this is in italics.

**Reviewer comments**

First: Kudos to the authors for pulling together a great review of existing reduced complexity model structures, components, and other factors. This will be very much appreciated by those of us who work with reduced complexity models.

Thank you for your kind words. The intention of this manuscript, which evolved from a literature review made by the lead author as part of his PhD, was precisely to serve as a guide to newcomers to the field as well as SCM users/developers. Hearing that this objective may have been fulfilled is very encouraging. We respond to each comment in turn here and in the revised manuscript.

Second: I have a couple minor comments:

Minor comment one regards this statement:

"Moreover, ESMs operate at finer scales, benefitting local and regional analysis, although downscalling approaches to generate regional climate emulators from SCMs have also been explored (Beusch et al., 2020; Mitchell, 2003; Mathison et al., 2025)"

I think this sentence, as written, is slightly misleading because most (all?) regional climate emulators depend on ESM data. All three cited approaches - Beusch et al (MESMER), Mitchell (pattern scaling), and Mathison et al. (PRIME) rely on ESM output in order to build their databases. I'd also recommend adding Tebaldi et al. (2022) (STITCHES) to your list of regional climate emulators (and which also relies on ESM data). Perhaps if the sentence was reframed as, "Moreover, ESMs operate at finer scales, benefitting local and regional analysis. Pattern scaling approaches that leverage ESM data to generate regional climate emulators, which can then be coupled with reduced complexity models to provide higher resolution functionality, have also been explored (citations)."

We agree that the statements could be misread as it was initially written. We have rewritten it to make sure we acknowledge these downscaling approaches are employing ESM data, and added a reference to STITCHES as suggested. The text now reads:

*"Moreover, ESMs operate at finer scales, benefitting local and regional analysis, although downscaling approaches that leverage ESM data to generate regional climate emulators have also been explored, partially mitigating this limitation (Beusch et al., 2020; Mitchell, 2003; Mathison et al., 2025; Tebaldi et al., 2022; Sandstad et al., 2025). These regional emulators can then be coupled with SCMs to provide higher resolution functionality."*

As a more general statement (and less relevant to this manuscript), I would like the ESM community to lean into the role of understanding the patterns of responses to forcing by running stylized scenarios (e.g., isolating GHG, aerosol, and land use change forcing components) rather than running scenarios produced by IAMs, which add complexity to comparing different

scenarios as they differ on all 3 forcing types. Then, these various pattern scaling approaches could be coupled to reduced complexity models to project more realistic scenarios as well as important probabilistic assessments.

This is a valid point, although some efforts in this area are underway. We have just submitted a manuscript, RCMIP Phase 3 (https://egusphere.copernicus.org/preprints/2025/egusphere-2025-5775/) that will run idealised experiments including some of the single-forcing experiments that you propose. While the scope of this intercomparison is global (and SCM-focused), we are aware of some efforts to isolate the regional patterns of different forcings, particularly the METEOR model (https://gmd.copernicus.org/articles/18/8269/2025/), which we have also added a citation for. New ESM experiments such as the Regional Aerosol Model Intercomparison Project (RAMIP; https://gmd.copernicus.org/articles/16/4451/2023/) will provide useful regional response data that can be used to calibrate regional single-forcing responses.

Minor comment two is that I would be interested in understanding in what fashion (if at all) box models and IRM approaches differ in terms of behavior. After looking at RCMIP and various other tests, I haven't identified any fundamental differences between IRMs (as a class) and box models (as a class). Perhaps the discussion at lines 390-409 regarding mathematical equivalence is the answer to this question? If that's true, maybe this equivalence could be emphasized even more.

They are indeed equivalent models displaying the same dynamics. We have updated the text (~L421) to make this more explicit:

*"While the n-time-constant temperature IRMs and the n-layer temperature models are mathematically equivalent, and could therefore be fundamentally considered the same model describing the same dynamics, the IRM formulation has the advantage of a simple relationship (Geoffroy et al., 2013a) between the parameters in Eq. 15 and two of the most critical and widely discussed quantities in climate science, the Equilibrium Climate Sensitivity (ECS) and the TCR:"*

Minor additional notes:

Typo: downscaling, not scalling

Thank you, now corrected in the text.

Figure 10 typo: atmopshere should be atmosphere

Thank you, now corrected in the text. While revising this figure I also detected an additional typo, which said "*greenshouse*" instead of "*greenhouse*" in "*concentration of greenhouse gases in atmosphere*". This is also now corrected.